# Glaciological history and structural evolution of the Shackleton Ice Shelf System, East Antarctica, over the past 60 years

Sarah S. Thompson,[1,2*] Bernd Kulessa,[2,3] Adrian Luckman,[2] Jacqueline A. Halpin[4], Jamin S. Greenbaum[5], Tyler Pelle[5], Feras Habbal[6], Jingxue Guo[7], Lenneke M. Jong[8], Jason L. Roberts[8], Bo Sun[6], Donald D. Blankenship[9]

[1]Australian Antarctic Program Partnership, Institute for Marine and Antarctic Studies, University of Tasmania, Hobart, Tasmania, 7001, Australia
[2]School of Biosciences, Geography and Physics, Swansea University, UK
[3]School of Technology, Environments and Design, University of Tasmania, Hobart, TAS, 7001, Australia
[4]Institute for Marine and Antarctic Studies, University of Tasmania, Hobart, Tasmania, 7001, Australia
[5]Scripps Institution of Oceanography, University of California, San Diego, USA
[6]Oden Institute for Computational Engineering and Sciences, University of Texas at Austin, Austin, Texas, USA
[7]Polar Research Institute of China, Shanghai, China
[8]Australian Antarctic Division, Kingston, TAS, Australia
[9]Institute for Geophysics, University of Texas at Austin, Austin, TX, USA

*Correspondence to*: Sarah S Thompson (ss.thompson@utas.edu.au)

**Abstract.** The discovery of Antarctica's deepest subglacial trough beneath the Denman Glacier, combined with high rates of
basal melt at the grounding line, have caused significant concern over its vulnerability to retreat. Recent attention has therefore been focusing on understanding the controls driving Denman Glacier's dynamic evolution. Here we consider the Shackleton system, comprising of the Shackleton Ice Shelf, Denman Glacier and adjacent Scott, Northcliffe, Roscoe and Apfel glaciers, about which almost nothing is known. We widen the context of previously observed dynamic changes in the Denman Glacier to the wider region of the Shackleton system; with a multi-decadal timeframe and an improved biannual temporal frequency
of observations in the last seven years (2015-22). We integrate new satellite observations of ice structure and airborne radar data with changes in ice front position and ice-flow velocities to investigate changes in the system. Over the 60-year period of observation we find significant rift propagation on the Shackleton Ice Shelf and Scott Glacier and notable structural changes in the floating shear margins between the ice shelf and the outlet glaciers, as well as features indicative of ice with elevated salt concentration and brine infiltration in regions of the system. Over the period 2017-2022 we observe a significant increase
in ice flow speed (up to 50%) on the floating part of Scott Glacier, coincident with small scale calving and rift propagation close to the ice front. We do not observe any seasonal variation or significant change in ice flow speed across the rest of the Shackleton system. Given the potential vulnerability of the system to accelerating retreat into the overdeepened, potentially sediment filled bedrock trough, an improved understanding of the glaciological, oceanographic, and geological conditions in the Shackleton system are required to improve the certainty of numerical model predictions and we identify a number of

priorities for future research. With access to these remote coastal regions a major challenge, coordinated internationally collaborative efforts are required to quantify how much the Shackleton region is likely to contribute to sea level rise in the coming centuries.

## 1 Introduction

The East Antarctic Ice Sheet has historically been perceived as the stable sector of Antarctica (Silvano et al., 2016); however, it has now emerged that the subglacial basins of Wilkes Land in East Antarctica have been contributing to sea level rise since the 1980s, with the Aurora subglacial basin contributing 1.9 mm (Rignot et al., 2019). The Shackleton system, fed by the Knox subglacial basin, is thought to be the most direct connection to the western portion of the Aurora subglacial basin (Fig. 1a). The Shackleton system is one of the largest ice shelf systems in East Antarctica and is comprised of a number of outlet glaciers including Denman, Scott, Northcliffe, Roscoe and Apfel. The floating component of the system is comprised of the Shackleton Ice Shelf together with the distinctive tongues of the Denman and Northcliff glaciers, Scott Glacier and Roscoe Glacier, and an area of fast ice to the west of the Denman Glacier tongue (Fig. 1b). The Denman Glacier alone is estimated to hold an equivalent of 1.5 m of sea level rise equivalent ice mass (Morlighem et al., 2020).

Ice thicknesses across the grounded portion of the system range from ~ 400 m inland of Shackleton Ice Shelf to > 4000 m at the Denman Glacier (Morlighem et al., 2020). Bed elevations are also wide ranging with the main outlet glaciers grounded well below sea level and experiencing retrograde slopes (Brancato et al., 2020; Morlighem et al., 2020). Close to the grounding line ice thicknesses range from 300 m to 1400 m, thinning toward the ice front where ice thickness is in the range of $180 - 250$ m with the exception of the front of the Denman Glacier tongue at ~ 450 m thick (Morlighem et al., 2020). Ice velocity data from the region are sparse before the late 2000s, but recent work identified an increase in ice velocity of the Denman Glacier of ~ 16 % since the 1970s (Rignot et al., 2019), with an increase of $11 \pm 5$ % just upstream of the Denman Glacier grounding line between 1972-74 and 1989 and a more recent rate of acceleration of $3 \pm 2$ % between 1989 and 2007-08 (Miles et al., 2021, their Fig. 3c). Analysis of Envisat data indicated that the Denman Glacier was thinning by 0.4 m year$^{-1}$ upstream of its grounding line between 2002 and 2010 (Flament and Rémy, 2012), and a $5.4 \pm 0.3$ km grounding line retreat was detected between 1996 and 2017–2018 (Brancato et al., 2020).

The discovery of the deepest sub-glacial trough in Antarctica (>3500 m below sea level) beneath the Denman Glacier, with a gentle and slightly retrograde bed slope close to the grounding line (Morlighem et al., 2020), has prompted suggestions that the system may be vulnerable to marine ice sheet instability, potentially triggered by high basal melt rates in the ocean cavity just offshore the grounding line (Brancato et al., 2020; Morlighem et al., 2020; Rignot et al., 2019). Meltwater production from basal melt of the Shackleton system (73 Gt year$^{-1}$) between 2003 and 2008 rivalled that from Thwaites (98 Gt year$^{-1}$) (Rignot et al., 2013). Satellite derived basal melt rates between 2010 and 2018 revealed high but localised average basal melt rates of > 50 m year$^{-1}$ close to the Denman grounding line (Liang et al., 2021), on par with basal melt rates in the Bellingshausen and

Amundsen Sea (Adusumilli et al., 2020). Beyond the Denman Glacier grounding zone, there has been much less observation, although the surrounding Shackleton system has so far shown few signs of major dynamic change, with its flow restrained by islands, ice rises, and ice rumples (Stephenson et al., 1989; Young, 1989) (Fig.1). Much of the Shackleton system has an average basal melt rate of between 0 and 1 m year$^{-1}$, with the exception of the Denman Glacier shear margins where refreezing in the order of 0.5 m year$_{-1}$ is indicated, as well as Roscoe Glacier where average basal melt rates away from the grounding line are 2-3 m year$^{-1}$ (Adusumilli et al., 2020; Liang et al., 2021).

The high basal melt rates close to the grounding line in this region of Antarctica have been linked to a warming of up to 0.5 °C over the last 40 years that occurred in the open ocean off East Antarctica, concurrent with an even more pronounced warming of 0.8–2 °C observed over the continental slope (Herraiz-Borreguero and Naveira Garabato, 2022). This Circumpolar Deep Water (CDW) warming is linked to a poleward shift of the Antarctic Circumpolar Current's southern extent onto the Indian Ocean sector of the East Antarctic continental slope, in which the Shackleton system is located. The continental slope warming appeared strongest near ice shelves that are thinning or have retreating grounding lines such as the Denman Glacier (Herraiz-Borreguero and Naveira Garabato, 2022). In addition, recent profiling float data have revealed that the thickest and warmest modified CDW layers in the region have been observed in a deep trough adjacent to the Denman Glacier tongue (van Wijk et al., 2022). Although basal melt is considered the dominant form of melt related mass loss in East Antarctica, the outermost portions of Shackleton Ice Shelf have been observed to experience the most intense surface melt outside of the Antarctic Peninsula (>>200mm w.e. year$^{-1}$) (Trusel et al., 2013). There is also evidence that the Shackleton Ice Shelf may experience an increase in surface melting with an increase in the length of the melt season observed between 2002 and 2011 (Zheng et al., 2018). Circum-Antarctic studies have identified the Shackleton system as experiencing substantial losses due to thinning in recent decades (Greene et al., 2022) and a reported average thickness change of -3.4 m between 2010 and 2017 (Hogg et al., 2021).

Despite increased scientific scrutiny in recent years (Arthur et al., 2020; Brancato et al., 2020; Miles et al., 2021; Morlighem et al., 2020; Rignot et al., 2019; Stokes et al., 2019), existing data and knowledge are still insufficient to predict the future evolution of the Shackleton system with confidence. Aside from incomplete understanding of the dynamic controls on Denman Glacier flow and Shackleton Ice Shelf stability, almost nothing is known about the adjacent Scott, Northcliffe, Roscoe and Apfel glaciers or their shear margins. Here we add to the previously reported dynamic changes in the Denman Glacier over the 60-year period of observation and place them into the wider regional context of the Shackleton system. We do so by also presenting an improved biannual temporal frequency of observations in the last seven years (2015-22) integrating airborne radar data, new satellite observations of ice structure, changes in ice front position and ice-flow velocities, with known geometrical and glaciological constraints. We firstly report on the main structural features (Sections 3.1-3.3) and dynamic changes across the whole system (Sections 3.4-3.6) and then discuss the changes in the system and their possible impact by sub-system region (Sections 4.1-4.4).

## 2 Methods

### 2.1 Structure and feature mapping from optical and synthetic aperture radar (SAR) imagery

Surface structures and features of the Shackleton system were mapped from satellite imagery using standard GIS techniques (following Glasser et al. (2009)). Structural features have been mapped every 6 months from February 2015 to February 2022 using freely available datasets from Landsat 8 OLI and Sentinel 2A and 2B, where cloud cover is <15%, in combination with Sentinel 1A and 1B GRD to improve spatial and temporal coverage. We used available optical and SAR imagery from February and SAR imagery from August of each year to allow us to capture any seasonal variation while maximising the number of

cloud-free optical images available. To include multi-decadal changes in the extent and structure of the whole system, we used several different datasets from three time periods, only choosing the datasets that covered the entire area of interest. These include the declassified ARGON KH-5 images acquired 16$^{th}$ May 1962, Landsat 1 MMS acquired 27$^{th}$ February 1974, Landsat 5 TM acquired between 10$^{th}$ and 12$^{th}$ February 1991 and the MODIS Mosaic of Antarctica (MOA) image map, a composite of 259 swaths of both Aqua and Terra MODIS images acquired between 01 Nov 2008 and 28 Feb 2009 (Scambos et al., 2007).

Datasets were registered to the Sentinel-2 imagery as required.

Mapped features included, where visible: the ice-shelf or floating-glacier edges, rifts, crevasses and crevasse traces and longitudinal surface features following the methodology of Glasser et al. (2009). Interpretation of the optical imagery was performed using multiple band combinations to provide natural colour (Landsat-1 MSS bands 7-4-3, Landsat-5 TM bands 5-

2-1 and Sentinel-2 bands 4-3-2) and for all imagery standard enhancement procedures (contrast stretching and histogram equalisation) were used to improve the contrast across features. The spatial resolution of the data sets varies from 10 m to 150 m and is thus a limitation on the minimum size and accuracy of the features mapped in each data set.

### 2.2 Feature tracking from Sentinel-1

Glacier surface velocities were derived using feature tracking between pairs of synthetic aperture radar (SAR) images acquired

by the Sentinel-1 satellite. Using the standard Gamma software (https://www.gamma-rs.ch/software) and following commonly adopted methods, feature tracking uses cross-correlation to find the displacement of surface features between pairs of images, which are then converted to velocities using the time delay between those images (Luckman et al., 2007). We use image patch sizes of ~ 1 km in ground range and sample at ~ 100 m in range and azimuth. Where the time-delay between images is sufficiently short, and surface change is minimized, trackable surface features include fine-scale coherent phase patterns

(speckle) and the quality of the derived velocity map is maximized. We applied feature tracking to all image pairs of 6 and 12-day repeat-pass period (about 60 per year) and selected the best single velocity map in terms of minimum noise and maximum coverage of high-quality matches for each year to provide mean annual velocity maps, including ice flow direction. We then produced percentage difference maps, scaled between +/- 10% but excluding data where the mean velocities are less than 0.2 m/day as uncertainties in velocity magnitude are around 0.2 m day$^{-1}$ (Benn et al., 2019). This approach allows us to optimize

the quality of the surface strain map derived from the surface velocity. The magnitude of the principal strain rate was derived from the mean velocity maps (2017-2020), strain was calculated across 2 pixels (of 100m) North-South and East-West of the central pixel, no filtering was applied to suppress noise.

### 2.3 Ice penetrating radar

The ice-penetrating radar data presented here were acquired on two survey flights using the Snow Eagle 601 BT-67 aircraft (Cui et al., 2018) flown on 19 and 20 December 2018 (Line locations shown in Fig. 7). The data were acquired using a radar system that is functionally equivalent to the High Capability Airborne Radar Sounder that has been described in the literature (Peters et al., 2007) and used in numerous studies of both grounded (e.g. Young et al., 2011) and floating ice properties and grounding zones (e.g. Greenbaum et al., 2015). The phase coherent radar system transmits a frequency modulated chirped

signal between 52.5 and 67.5 MHz. The images presented in this manuscript reflect a postprocessing sequence that coherently adds 10 raw radar records at a time to increase signal to noise, applies matched filtering to account for the chirped transmit pulse, then incoherently stacks the resulting complex valued radar traces five times to suppress speckle noise. The ice bottom elevation data were computed using horizons picked with a semiautomatic approach involving manual localization above and below horizons of interest (the ice surface and ice bottom interfaces in this instance). A constant ice velocity of 167

m/microsecond was applied to generate each radargram.

### 3 Results

We have identified a number of dynamic and structural features of interest across the Shackleton system which we present in turn, describing changes in each of the following sections by sub-system region.


### 3.1 Ice front positions

Between 1962 and 2022, there is some small-scale variability (1-2 km) in the front position of Roscoe Glacier, but this is not consistent along the length of the ice front and there is no sustained change evident in the front position or the shape of the floating extent of Roscoe Glacier (Fig. 2a). Over the same time period, the ice front position of the Shackleton Ice Shelf

advanced a total of 18 km with no obvious change in the annual rate of advance (Fig. 2a). Calving occurred from the ice front of the Shackleton Ice Shelf, on the side adjacent to Roscoe Glacier in 1991, and a portion of the calved ice has since remained grounded just offshore of it (Fig. 2a). Between 2015 and 2022, the front of the Shackleton Ice Shelf then advanced steadily by ~ 0.3 km year[-1]. An iceberg from a calving event on the Denman Glacier, hypothesized to have occurred in the late 1940s (> 70 km in length; Miles et al., 2021) is visible in 1962, roughly 100 km offshore the ice front (Fig. 2a). The Denman ice front

position retreated in 1984 due to another major calving event (54 km in length; Miles et al., 2021). By 1991 the floating margin was still located 10-15 km south of the 1962 position but has since advanced ~ 63 km (Fig. 2a). Over the tine period 2015-2022, the Denman Glacier's ice front, which includes Northcliff Glacier, advanced at a rate of 1.8 km year[-1] with a uniform

pattern of advance and no seasonal variability in advance rate observed (Fig. 2b). The floating ice front of Scott Glacier has experienced more variability than that of Denman or Shackleton (Fig. 2a). Between 2015 and 2019, the front advanced at a steady rate of ~ 0.75 km year$^{-1}$ (Fig. 3a). Since 2020, small scale calving has been occurring on the side of the Scott Glacier adjacent to Mill Island, where the ice front position now lies ~ 5 km inland of the 2015 front (Fig. 3b-d). In February 2022, the portion of the Scott Glacier ice front adjacent to Chugunov Island was in a similar position to that of 1962 but the whole ice front was ~ 10 km further inland in 2009 (Fig. 2a, 3d). and in April 2022, a section >25 km long calved from the Chugunov Island side of Scott (Fig. 4a). There is no clear margin between Scott Glacier and Apfel Glacier offshore of the Tylor Islands so we cannot comment on a distinct ice front position (Fig. 4a).

### 3.2 Rifts and crevasses

There is little observable change in surface structure on Roscoe Glacier with the exception of a rift opening in the vicinity of the margin with Shackleton Ice Shelf (Fig. 5). In 2022 the rift is 15 km long and 2 km wide at the widest point and extends to within 3.2 km of the ice front, a significant increase in dimensions of 5.3 km and 0.35 km, observable in 2015 when the feature terminated 8.5 km from the ice front (Fig. 5). Two major rift systems dominate on the Shackleton Ice Shelf, both of which extend into the ice shelf (labelled 'R1' and 'R2' in Fig. 1b) from the region of heavily fractured ice shelf ice, ~ 2,300 km$^2$ in size (Fig. 1b), held in place by fast ice and ~ 150 m thinner than the adjacent ice shelf body (Fretwell et al., 2013). System R1 is a maximum of ~ 15 km wide at the margin and extends over 40 km into the ice shelf, narrowing and eventually terminating at a spatially extensive suture zone that originates in the leeside cavity of Masson Island (Fig. 1b & 2a). A subsidiary rift branches off and connects with the ice shelf front (Fig. 2a). The geometry of system R1 has not changed significantly since 1962, although its width increased by ~ 5.3 km between 1962 and 1991 (Fig. 2a) and in 1962, there was no clear connection between the infant subsidiary rift and a front-parallel rift, visible by 1991 (Fig. 2a). System R2 is a maximum of ~ 5 km wide at the margin and extends into the shelf for 16.5 km, before branching into two rifts that trend in opposing directions, ~ 14 km and ~ 21 km in length respectively (Fig. 1b & 2a). System R2 changed more substantially than system R1, branching towards the grounding line and lengthening by 3.8 km between 2015 and 2022. In 1962 the rift is only visible as a crack, opening to a rift 2.3 km wide by 1991 and at the eastern edge, 4.5 km wide by 2017 (Fig. 1b & 2a). Between 1991 and 2015 the branch extending towards the grounding line increased in length from ~ 10 km to ~ 16 km and in width by ~ 1 km at the ice margin (Fig. 1b & 2a). The branch extending towards the ice front increased in length from ~ 13 km to ~ 16 km over the same time-period. Both systems advected with ice flow towards the ice front between 2015 and 2022, with no significant changes in shape (Fig. 2a). The surface of the Denman Glacier tongue is heavily fractured with a combination of crevasses, flow lines and channel-like features (Fig. 1b). However, there is a distinct difference in surface appearance across the width of the tongue (Fig. 2b). The ice originating from Northcliff Glacier on the Shackleton Ice Shelf side of the tongue, has a much higher spatial density of crevasses, while the ice originating from Denman Glacier is dominated by a combination of flowlines, channel-like features, and crevasses (Fig. 2b). A number of small rifts (< 6km long) are evident along the margin, separated by fast ice from Shackleton Ice Shelf and there is no identifiable change on the length or position of these rifts relative to the ice front between

2015 and 2022 (Fig. 2b). The floating portion of Scott Glacier is dominated by a series of rifts striking perpendicular to the flow direction (Fig. 4). The rifts initiate approximately 20 km down glacier of the grounding line (as defined by MEaSURES (Rignot et al., 2017)) and widen to ~ 2.5 km as they flow around the Taylor Islands. Between 2015 and 2022 the up-flow rift S1 widens at a rate of ~ 200 m year$^{-1}$ (Fig. 4a – labelled S1), while the down-flow rift S2 narrows at a rate of ~ 100 m year$^{-1}$ (Fig. 4a – labelled S2). The rift formation, widening and narrowing process is evident from 1962 through to 2009 (Fig. 4b). A rift on Scott Glacier, initiated from the shear margin on the Mill Island side of Scott Glacier, has increased in length toward Chugunov Island by ~ 5 km between February 2021 and June 2022 (Fig. 4c - labelled S3). There is no clear margin between Scott Glacier and Apfel Glacier offshore of the Tylor Islands and no observable rifts on Apfel Glacier inland of the Tylor Islands (Fig. 4a).

### 3.3 Ice thickness

The ICECAP airborne radar lines flown in December 2016 provide a snapshot of information about the thickness and ice structure through the floating ice. The available airborne radar lines do not cover the Roscoe Glacier. The side of the Shackleton Ice Shelf adjacent to Denman Glacier thins from ~300 m thick, closer to the grounding line to ~150 m thick close to the Denman Glacier – Shackleton Ice Shelf shear margin (Fig. 7a), with a smooth, clearly defined ice base (Fig. 7b). A persistent near surface reflector is found in the Shackleton Ice Shelf region in all 6 radar lines, ~ 50 below the main surface reflector, at ~0 m asl in elevation (Fig. 7b). The Denman Glacier tongue consists of two distinct longitudinal sections with very different ice thicknesses, with the portion adjacent to Shackleton Ice Shelf ~ 130-150 m thick and originating from Northcliff Glacier (Fig. 7a). The side of Denman Glacier tongue adjacent to Scott Glacier ranges from ~ 300 m to >500 m thick towards the central flowline (Fig. 7a), with the thickest parts of the tongue following the longitudinal features visible at the surface (Fig. 7a). In all 6 radar lines the Denman-Northcliff Glacier region shows significant surface and basal roughness (Fig. 7a). The Scott Glacier tongue has less variation in thickness across the width but thins from ~ 370 m thick in the vicinity of the first rift to ~ 150 m thick close to the ice front (Fig. 7a). The radar lines which extend beyond the Scott Glacier shear margin, towards the Taylor Islands and Mill Island appears dimmer, and the reflectors muted (Fig. 7b).

### 3.4 Shear margins

Across the whole system, changes have been observed in the shear margins separating the various inlet glaciers and along the main body of the Shackleton Ice Shelf. Between 2015 and 2022, small changes are observable in the floating shear margin between the Shackleton Ice Shelf and Roscoe Glacier. As described in Section 3.2, a rift along the shear margin was observed to be three times the length and 10 times the width to that observed in 2015 (Fig. 5). There is no significant observable change in the floating shear margin between Shackleton Ice Shelf and Northcliff Glacier, visible surface features are advected toward the ice front with ice flow with little change in geometry (Fig. 1b). Over the same time period, small changes are observable in the floating shear margins on both sides of Scott Glacier (Fig. 6). The Scott Glacier shear margin that flows past the Tylor Island appears as a series of small rifts and crevasses, largely perpendicular to ice flow (Fig 6a-b). Over the 7-year period there

has been lengthening of the features into the ice to both sides of the margin, as well as opening of existing features (Fig. 6a-b). The Scott Glacier floating shear margin that abuts the Denman Glacier is more clearly defined and has been widening into Denman Glacier in the vicinity of Chugunov Island (Fig. 6c-d). In 2015 this margin is relatively straight, in line with the floating margin of Denman and ~ 1.3 km wide. Notably, the shear margin appears to bulge progressively into Denman Glacier and is double the width by 2022 as compared with 2015 (Fig. 6c-d).

### 3.5 Mean ice flow speed and strain rate

Mean ice speed derived at annual temporal frequency from Sentinel-1 data varies across the Shackleton System. In 2021, ice speed ranges from ~0.2 m day$^{-1}$ in the area between the grounding line and Masson Island on the Shackleton Ice Shelf to ~5 m day$^{-1}$ on the tongue of Denman Glacier (Fig. 8a). Surface ice speeds observed along Roscoe and Scott glaciers reach 1-2 m day$^{-1}$ and 2-3 m day$^{-1}$, respectively. Inland of the junction with Denman Glacier, Northcliff Glacier ice flow speed is in the range of 1.5 m day$^{-1}$, but once the two glaciers join there is no distinction between flow speeds (Fig. 8a). There are significant gaps in the coverage of Apfel Glacier, but observable values are in the region of 0.4 – 0.6 m day$^{-1}$ (Fig. 8a). The magnitude of the principal strain rate, derived from the mean velocity maps (2017-2020), highlights the shear margins of the Denman Glacier and Roscoe Glacier with the Shackleton Ice Shelf and the shear margins of the Scott Glacier (Fig. 8b), as well as four pinning points across the system (Fig. 8b). Pinning points have previously been identified at the front of the Roscoe Glacier and Shackleton Ice Shelf shear margin (label-a in Fig. 8b) and upstream of rift R2 on the Shackleton Ice Shelf (label-b in Fig. 8b, (Fürst et al., 2015). There is evidence of two additional pinning points, Chugunov Island at the front of the Denman-Scott shear margin (label-c in Fig. 8b) and at the ice margin of Shackleton Ice Shelf (label-d in Fig. 8b). The latter coincides with a local topographic high in ocean bathymetry (Arndt et al., 2013).

### 3.6 Temporal change in ice flow speed

Recent changes in ice speed, derived by differencing the mean speed across the whole system between 2021 and 2018, are confined to the seaward ~ 60 km of the floating tongue of Scott Glacier and the Shackleton Ice Shelf (Fig. 9). On the floating portion of Scott Glacier increases of > 10 % occur across the outer 50 km (Fig. 9). The side of the Shackleton Ice Shelf closer to Denman Glacier, including the fast ice, appears to have decelerated over the same time period, where a change of between -4 % and -6 % is observed (Fig. 9). There is some evidence of deceleration on the side of the ice shelf closer to Roscoe Glacier, although the signal is unclear with values ranging between +/- 5 %. Annual ice speed percentage differences illustrate the increase in ice speed on the Mill Island flank of the ice front of Scott Glacier between 2017 and 2018 (Fig. 10a), which then appears to decelerate between 2018 and 2019, when a 4 % increase in the ice speed of the Chugunov Island flank of the ice front is observed (Fig. 10b). The increase in speed continues through 2019-2020 with a 10% acceleration from the ice front up to 45 km upstream and across the entire ice tongue width (Fig. 10c). The increase extends a further 15 km in the up-flow direction between 2020 and 2021 (Fig. 10d). There is more spatial variability in the annual speed percentage differences across

the Shackleton Ice Shelf. The overall trend between 2018 and 2021 appears to be deceleration but there are small regions of acceleration and much of the variability is within the uncertainty bounds, thus complicating interpretation (Fig, 9).

Ice speed extracted from Sentinel-1 provides a timeseries between 2017-2021, which we extend back to 2002 using MEaSUREs (Mouginot et al., 2012, 2019; Rignot et al., 2011) and ITS_LIVE (Gardner et al., 2018, 2021) in locations where available to highlight variability through time across the system (Fig. 9 and 11). Point locations on Shackleton Ice Shelf vary between ~ 0.2 m day$^{-1}$ at the grounding line (point 3 and 4 in Fig. 9) and ~ 1 m day$^{-1}$ towards the front of the floating ice (point 2 in Fig. 8b), with no consistent temporal trends (Fig. 11a). Denman Glacier exhibits higher speeds, from < 2 m day$^{-1}$ upstream of the grounding line (point 5 in Fig. 9) to ~ 5 m day$^{-1}$ on the floating tongue (point 9 in Fig 9) but speeds remain constant through time at each point location (Fig. 11b). Scott Glacier has a similar spatial pattern with speeds increasing from ~ 1.2 m day$^{-1}$ at the grounding line (point 10 in Fig. 9) to > 4 m day$^{-1}$ close to the floating ice front (point 13 in Fig 9; Fig. 101c). There is no observable change in speed within 10 km of the grounding line of Scott Glacier (points 10 and 11 in Fig. 9; Fig. 11c). However, the downstream 30 km of the floating ice tongue show significant acceleration from January 2020 through to May 2021 (points 12 and 13 in Fig. 9; Fig. 11c). Over the 17-month period, ice speeds increase ~ 30 % to 2.5 m day$^{-1}$ 30 km from the ice front (point 12 in Fig. 9) and ~ 40 % to 3.2 m day$^{-1}$ close to the front (point 13 in Fig. 9; Fig. 11c). Roscoe Glacier has similar ice speed spatial patterns to both Shackleton Ice Shelf and Denman Glacier, with slower speeds of ~ 0.4 m day$^{-1}$ at the grounding line (point 14 in Fig. 9), increasing to ~1.2 m day$^{-1}$ close to the floating ice front (point 16 in Fig. 9) and no significant change in speed through time (Fig. 11d).

## 4 Discussion

Over the ~ 60-year period of observation, the Shackleton system has undergone observable variability in ice velocity and structure, although more frequent satellite observations in recent years have not revealed any distinct seasonal or annual cycles of variability across it. We discuss the observable changes by sub-system region, while considering their possible implications for the whole Shackleton system.

### 4.1 Roscoe Glacier

Roscoe Glacier is the only part of the Shackleton system where we do not observe any significant change in ice front position (Fig. 1b), despite consistent ice flow speeds in the region of 1 m day$^{-1}$ throughout the period of observation (Fig. 8, 11d). There is extensive crevassing visible at the surface (Fig. 1b) which could potentially facilitate high frequency, small scale calving if the features visible at the surface extended to depth. Additionally, the location of Roscoe Glacier at the margin of the system may mean that it experiences slightly different ocean forcing than the rest of the Shackleton System. Average basal melt rates of 2-3 m year over the floating region of the Roscoe Glacier is higher than much of the floating ice of the system away from the grounding zones (Adusumilli et al., 2020). Although no change in flow speed is evident over the period of observation, we do observe structural change at the shear margin with Shackleton Ice Shelf. A rift located along the shear margin both

lengthened and widened since 2015 but has also been advected towards the ice margin with ice flow (Fig. 5). The effect of the feature is difficult to predict as the point at which the rift will intersect the ice front is the location of a pinning point (labelled 'a' in Fig. 8b), as previously identified by Fürst et al. (2015), who noted that the pinning point does not coincide with a topographic rise.

### 4.2 Shackleton Ice Shelf

The front of the Shackleton Ice Shelf is slowly advancing and changes in the geometry of the main surface features are restricted to the two main rift systems (Fig. 2a). The area of fast ice to the Denman Glacier side of the Shackleton Ice Shelf has decelerated by ~ 8 % in the period 2018-2021 (Fig. 9). There is some evidence of deceleration across the rest of the Shackleton Ice but the change in speed varies between + 6 % and -8 % over small areas indicating more significant uncertainty in the overall signal (Fig. 9). In 2021 flow speeds in this area of the ice shelf were in the region of 1 m day$^{-1}$ so that the annual rate of deceleration over the 3-year period would only equate to ~ 10 m year$^{-2}$. It is possible that the deceleration could be linked to reported thinning of the Shackleton Ice Shelf over the period 1997-2021 (Greene et al., 2022) but it contradicts instantaneous changes in velocity of +4 % modelled in response to ice shelf thinning between 1994 and 2012 (Gudmundsson et al., 2019). The lack of significant rift propagation in rift system R1 on the main body of Shackleton Ice Shelf appears directly related to the Masson Island suture zone (Fig 2a). Indeed, there is growing recognition that the softer marine ice present in suture zones inhibits the growth of large-scale fractures, acting to stabilise ice-shelves by reducing local stress intensities (Kulessa et al., 2014; Larour et al., 2021; McGrath et al., 2014). While current observations suggest suture zones promote stability by halting rift propagation, a strong relationship between the thickness of ice mélange and the opening rate of the rifts has been observed, indicating that ice mélange thinning rather than ice shelf thinning can promote rift propagation (Larour et al., 2021). The increased sea water content and warmer temperature of the ice mélange suggests it may be more vulnerable to thinning due to future surface and basal melting than the surrounding meteoric ice, potentially affecting rates of rift opening and propagation (Kulessa et al., 2014, 2019; McGrath et al., 2014). Rift system R2 has undergone more significant change over the period of observation, although the rift tip does not appear to have reached the Masson Island suture zone (Fig. 2a) as yet and may experience a similar behaviour to that of R1 when it does. With prominent suture zones and pinning points as likely agents of stability (Kulessa et al., 2014, 2019), increased ice shelf thinning could lead to a significant calving event due to coincident suture zone thinning and further rift propagation, with unpinning of the ice shelf on the side of the Shackleton Ice Shelf adjacent to Roscoe Glacier ('d' in Fig. 8b). The relatively slow grounded ice flow speeds onshore of the Shackleton Ice shelf suggests such a calving event is unlikely to initiate an increase in ice flow into the ocean, indeed it is considered passive in terms of ice buttressing (Fürst et al., 2016).

The persistent reflector at ~ 0 m asl in the airborne radar data over the Shackleton Ice Shelf (Fig. 7b) could be a result of strong melting and refreezing events observed on ice shelves elsewhere (Kuipers Munneke et al., 2017). These would lead to enhanced firn air depletion and are hypothesised to be a precursor to ice shelf collapse (Kuipers Munneke et al., 2014). Although, over

the period of observation there is reportedly little evidence of an increase in visible surface melt or ponding area on Shackleton Ice Shelf (Arthur et al., 2020), there is some evidence of an increase in the length of the melt season (Zheng et al., 2018). An equally plausible interpretation of the bright reflector could be brine infiltration of the firn layer as this region lies within a zone thought to be susceptible to brine infiltration (Cook et al., 2018). Brine has been detected in firn cores from a number of

Antarctic ice shelves (Heine, 1968; Kovacs et al., 1982; Risk and Hochstein, 1967; Thomas, 1975) and observed as a bright reflector close to sea level in radar data on the McMurdo (Campbell et al., 2017; Grima et al., 2016), Wilkins (Vaughan et al., 1993), Larsen (Smith and Evans, 1972), Brunt (Walford, 1964) and Ross ice shelves (Neal, 1979). Observations indicate brine infiltration may enhance fracture propagation and hydrofracture in ice shelves (Cook et al., 2018; Grima et al., 2016) and has been suggested to have contributed to the disintegration of the Wilkins Ice Shelf in West Antarctica (Scambos et al., 2009).

Although both suggestions are plausible at this location, current observations are insufficient to conclusively identify the most cause of the reflector and therefore the impact of the evolution of the system.

### 4.3 Denman-Northcliff Glacier

What has been referred to as the Denman Glacier tongue is comprised of two distinct ice masses originating from the Northcliff

and Denman Glaciers. The whole of the floating region of the Denman-Northcliff Glacier appears to behave dynamically as a single unit, exhibiting the same ice flow speed (Fig. 8, 9, 10, 11), despite significant differences in ice thickness (Fig. 7) and surface crevasse frequency (Fig 2b). The ice originating from both the Denman and Northcliff Glaciers exhibits significant basal roughness (Fig. 7), which has been cited to significantly influence heat and salt exchange at the ice-ocean interface (Watkins et al., 2021). The acceleration in ice flow speed observed just upstream of the Denman Glacier grounding line

between 1972-4 and 1989 and, to a lesser extent, through to 2008 (Miles et al., 2021, their Fig. 3c) is not observable post 2008 (Fig. 11b) and we do not identify any changes in surface structure to provide insight as to the cause of the acceleration. The ice flow speed of the Denman-Northcliff Glacier has been constantly ~ 5 m day$^{-1}$ close to the ice front over the period 2015-2022, with a uniform flow direction and no observable seasonal variability in flow speed (Fig. 2b, 8a, 11b). While efficient meltwater surface-to-bed connections via moulins and fractures are common in the Greenland Ice Sheet and seasonal changes

in surface meltwater variability have been observed to cause seasonal changes in ice flow speeds (Hoffman et al., 2018; Sundal et al., 2011), there is currently very little evidence for coupling between the surface and basal hydrological systems in Antarctica (Bell et al., 2018). However, season variations in ice flow speed have been observed on an outlet glacier in East Antarctica, coincident with seasonal sea ice break up reducing back stress on the system and the onset of seasonal melt thought to weaken the shear margins (Liang et al., 2019). Our results suggest that seasonal variation in sea ice extent or surface melting

are currently insufficient to have an observable forcing effect on the Shackleton system.

### 4.4 Scott-Apfel Glacier

Scott Glacier has received less attention until now, being thinner and slower than Denman Glacier, with an overall decrease in velocity observed between 1972-4 and 2016-7 (Miles et al., 2021). However, this part of the system is where we observe more

variability (Fig 4, 6, 7, 8, 9, 10, 11). We observe the ice front of Scott Glacier to be in a similar position in 2009, 2002, 1991 and 1962 (Fig. 2a, 3), but the ice front may have experienced retreat inland of this position in the intermediate time periods. However, the ice front position observed in June 2022 (Fig. 4a) is further inland than any previous observations, only marginally so adjacent to Mill Island but by ~ 18 km adjacent to Chugunov Island, with the calving of a 27 km long block in April 2022 (Fig. 4a). Furthermore, the recent > 5 km extension of rift S3 on Scott Glacier leaves only ~ 6 km of heavily

crevassed ice between the S3 rift tip and a small rift that initiates at the ice front at Chugunov Island (Fig. 4a). If rift S3 continued to propagate and the region of ice offshore of the rift calved, the effect of Chugunov Island as a pinning point (labelled in C in Fig. 8b) could be reduced. The observed changes in ice front position of Scott Glacier are coincident with a doubling of ice flow speed at the ice front, from 2 m day$^{-1}$ in January 2020 to 4 m day$^{-1}$ in January 2022 (Fig. 11c). The increase in ice flow speed extends from the ice front to rift S1 but is not observable inland of this feature, suggesting that the

acceleration is being initiated on the floating ice rather than related to any changes at the grounding line. While rift S1 on Scott Glacier has experienced an increase in width in recent years, the rate of change has remained constant at ~200 m year$^{-1}$ between 2015 and 2022 (Fig. 4). We currently do not have enough information to determine causation and therefore to predict future change on Scott Glacier, but it does coincide with instantaneous changes in velocity of +8 % modelled in response to ice shelf thinning between 1994 and 2012 (Gudmundsson et al., 2019). It would be reasonable to expect a continuation of higher flow

speeds or a further increase in ice flow speed on the section of Scott between rift S1 and the ice front if the glacier tongue became unpinned from Chugunov Island. Further investigation is needed to determine the extent to which changes on Scott Glacier tongue impact on the adjacent Denman tongue, as the change in the geometry of the shear margin between Scott Glacier and Denman Glacier (Fig. 6c-d) is coincident in time with the changes in ice front position and flow speed of this section of Scott Glacier.


The muted reflectors in the radar lines that extend past the Scott Glacier shear margin adjacent to Mill Island (Fig. 7) may indicate that the extremely high salt concentration found in the Mill Island ice core (Inoue et al., 2017) extends from Mill Island up-flow towards the grounding line (Fig. 7a). High salt concentrations in ice facilitate mechanical deformation, enhancing grain boundary sliding as well as reducing the melting temperature of ice (De Almeida Ribeiro et al., 2021). If the

muted reflectors identified (Fig. 7b) are in indication of high salt concentration, any changes in atmospheric or ocean forcing could have an enhanced impact on this region of the Shackleton system. No changes in structure or ice flow speed are observed up flow of the large rift to the west of the Taylor Islands (Fig. 4c – labelled 1), and the acceleration does not currently appear to have any connection to the grounded ice (Fig. 9, 11c). Surface meltwater features, reported to be frequent around the Scott and Apfel grounding lines, do not appear to be increasing in area or frequency between 2000 and 2020 (Arthur et al., 2020)

and are unlikely to be contributing to the changes observed on Scott Glacier.

**5 Conclusions**

Over the 60-year period of observation of the Shackleton system we observe significant rift propagation on the Shackleton Ice Shelf and Scott Glacier, and notable structural changes in the floating shear margins between the ice shelf and the outlet glaciers. Over the period 2017-2022 we observe a significant increase in ice flow speed (~ 50 % close to the ice front) on the floating part of Scott Glacier. Over the same time period we do not observe any seasonal variation or significant change in ice flow speed across the rest of the Shackleton system, and there is no observable change in the ice flow speed of the grounded ice or at the grounding line. However, given the likelihood of modified CDW (observed to be 0.8–2 °C warmer) accessing the shelf (Herraiz-Borreguero and Naveira Garabato, 2022; van Wijk et al., 2022), coupled with the observations that the outermost portions of Shackleton Ice Shelf experience the most intense surface melt outside of the Antarctic Peninsula (Trusel et al., 2013), and a lengthening of the melt season (Zheng et al., 2018), the Shackleton system appears vulnerable to changes in both ocean and atmospheric forcing. Indeed, observed grounding line thinning and retreat (Brancato et al., 2020; Flament and Rémy, 2012) and localised ice flow speed change (Miles et al., 2021; Rignot et al., 2019) may indicate that the Shackleton system is already responding to changes in forcing. However, the timescales over which the system responds, and the implications of this response remain challenging to predict.

We still do not have a clear picture of the cause of the 1972-2008 acceleration in ice flow speed of the Denman Glacier or indeed its later stabilisation. A previous calving cycle reconstruction indicated that a calving event at some point in the 2020s is highly likely (Miles er al., 2019) but from our observations we do not find any indication of when this may occur or the mechanism by which it may be initiated. In addition to thinning and induced dynamic changes there are several features of the system that could significantly speed up the response to external forcing. Such features include; (i) thinning suture zones; if the Masson Island suture zone on Shackleton Ice Shelf is thinning in response to ocean warming, the melange at the base of the suture zone will preferentially melt, promoting rift propagation and ice shelf break up (Kulessa et al., 2014; Larour et al., 2021; McGrath et al., 2014); (ii) shear margin weakening; the shear margins across the whole system are likely to weaken with increasing melt and elevated meltwater availability but in the region of the system from Scott Glacier towards Mill Island, potentially high salt concentrations in the ice could further enhance melt and increase deformation; (iii) brine infiltration; possible brine infiltration of the firn layers on Shackleton Ice Shelf could enhance fracture propagation and promote hydrofracture if there is an increase surface meltwater with the observed lengthening melt season; and (iv) subglacial conditions; heat flow anomalies in the region of the Denman Glacier are poorly resolved but recent multivariate analysis of available datasets indicate elevated geothermal heat flow (> 70-80 mW m$^{-2}$) to the west of the Denman region (Wilhelm II Coast) and in the Knox Basin interior (Stål et al., 2021). As observed elsewhere in East Antarctica, the deep trough beneath the Denman Glacier may favour vigorous channelisation of the subglacial meltwater system close to the grounding line (Dow et al., 2020) and freshwater outflow into the sub-ice shelf ocean cavity could locally enhance basal melt rates near the grounding line (Jenkins, 2011; Wei et al., 2020).

The potential vulnerability of the Shackleton system to increasing atmospheric and ocean forcing, the magnitude of potential sea level rise of the Denman Glacier alone (~ 1.5 m) (Morlighem et al., 2020) and the potential link to the Aurora Subglacial Basin (through the Knox Basin) make this region of East Antarctica one of significant importance in improving predictions of sea level rise. Critical to assessing the timing and magnitude of sea level rise contributions are improvements in the measurement of ice cavity bathymetry, subglacial conditions, and the evolution of surface melt water systems, as well as targeted collection of field data to allow better incorporation of features such as suture zones, pinning points and brine infiltration into numerical models of ice shelves.

## Acknowledgements

This project received grant funding from the Australian Government as part of the Antarctic Science Collaboration Initiative program, the AXA Research Council through an AXA Post-Doctoral Fellowship and The National Natural Science Foundation of China grant 41941007. J.S.G. acknowledges supported from NSF OPP-2114454 and NASA grant 80NSSC22K0387. L.M.J. and J.L.R. acknowledge support from Australian Antarctic Division project 4346 and the Antarctic Gateway Partnership (University of Tasmania, Australia). J.G. acknowledges support from the National Natural Science Foundation of China grant 41941007. We thank Gregory Ng for engineering support in the field and with radar data processing.

## Data availability

All satellite imagery used in this work are freely available as follows; Landsat 8 OLI, 5 TM and 1 (all downloaded from https://earthexplorer.usgs.gov/), MODIS Mosaic of Antarctica 2008-2009 (downloaded from https://nsidc.org/data/nsidc335 0593/versions/2), Sentinel 2 A and B (all downloaded from https://scihub.copernicus.eu/dhus/#/home) Sentinel 1A and B GRD (all downloaded from https://scihub.copernicus.eu/dhus/#/home) and ARGON KH-5 (downloaded from https://earthexplorer.usgs.gov/). BedMachine (v2) data is freely available (https://nsidc.org/data/nsidc-0756/versions/2). The radar-based ice bottom elevation data and radargrams presented here will be published to the Australian Antarctic Data Centre (https://data.aad.gov.au/) upon publication.

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

**Figures**

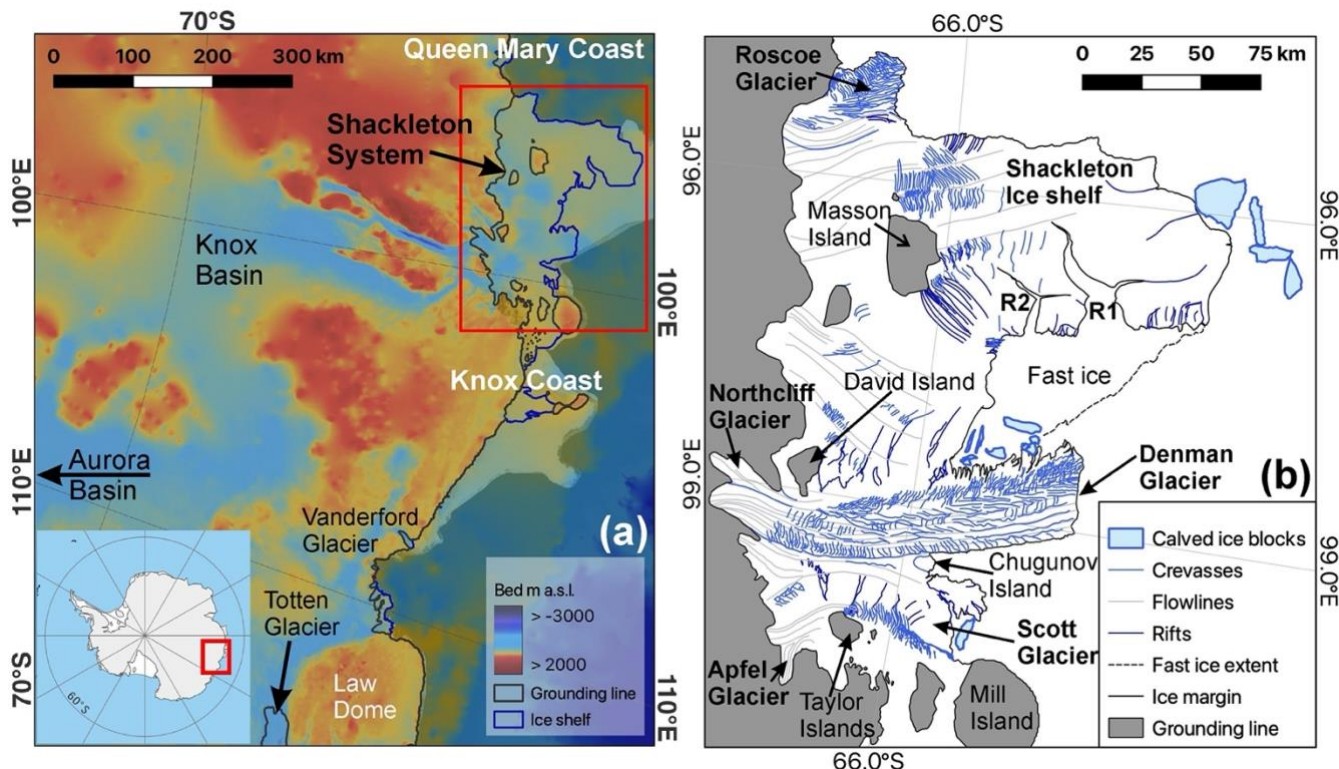


**Figure 1: (a) Area of focus in the regional context of the Aurora and Wilkes subglacial basins, location shown in inset (Background: BedMachine V2 (Morlighem et al., 2020). (b) The Shackleton system overview in February 2021 with the two main rift systems on Shackleton Ice Shelf labelled R1 and R2. All features mapped from Sentinel 2A and 2B imagery acquired 05th – 27th February 2021.**


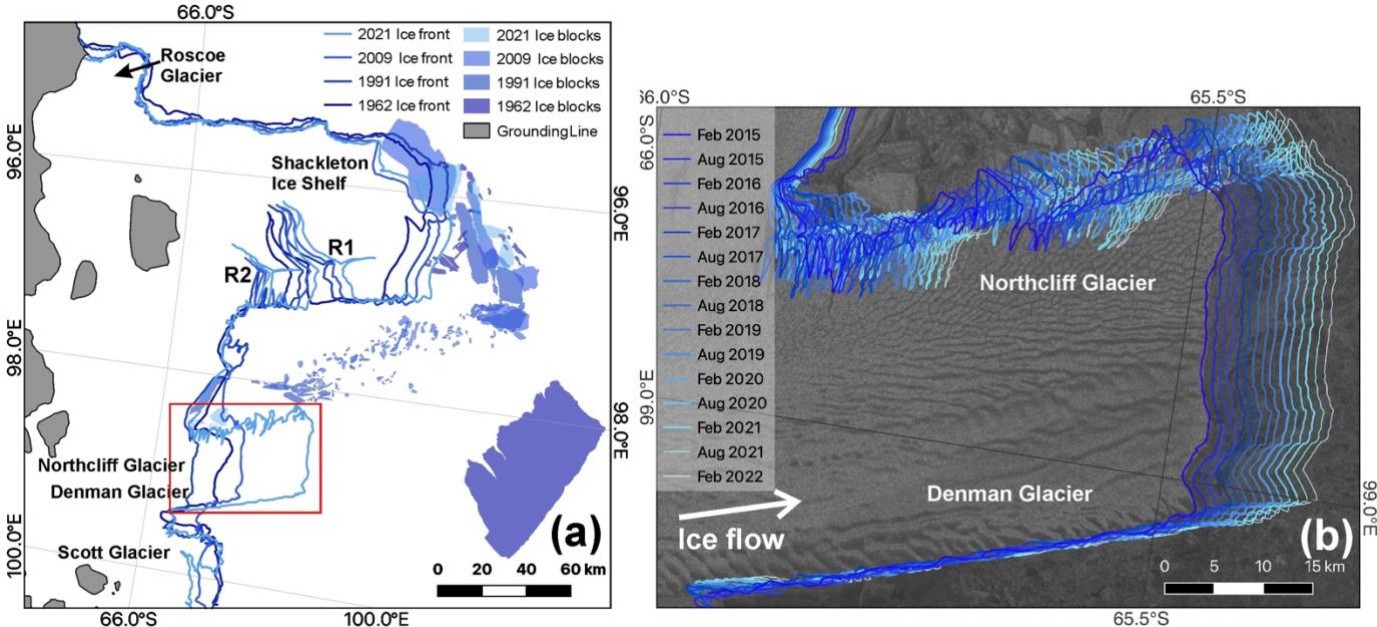

Figure 2: (a) Ice front positions of the Queen Mary and Knox coasts since 1962, including the large iceberg hypothesised to have calved from the Denman tongue in the 1940s. Position and blocks mapped from 16th May 1962 – ARGON KH5, 10th -12th February 1991 – Landsat 5 TM, 1st November – 28th February 2009 – Modis MOA (Scambos et al., 2007) and 5th – 27th February 2021 – Sentinel 2A and 2B. The two main rift systems on the Shackleton Ice Shelf are labelled 1 and 2. (b) Denman Glacier biannual ice front position mapped in February and August from 2015 through 2022 (Background: Sentinel 1a acquired 27th February 2015).


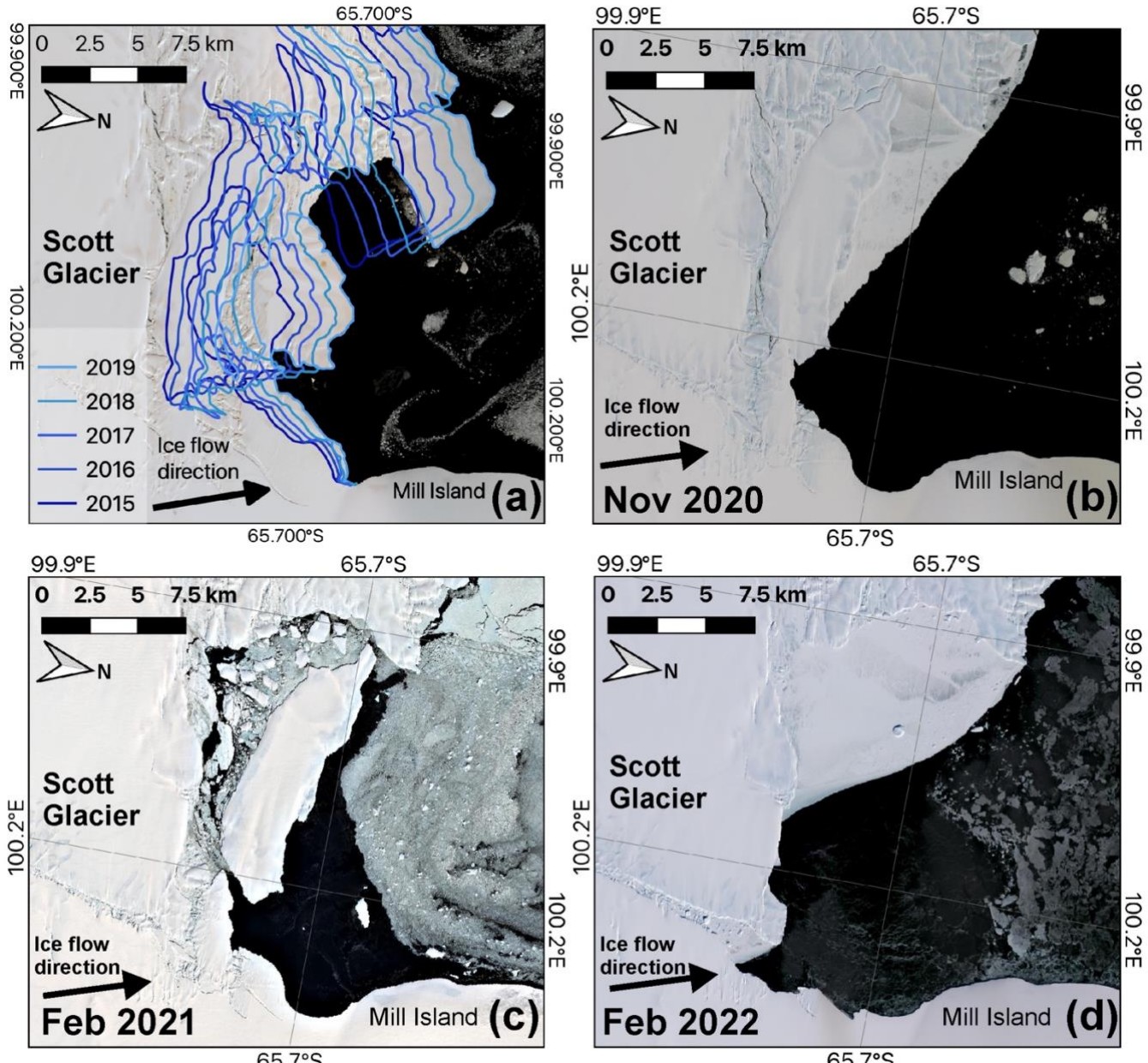

Figure 3: (a) Scott Glacier ice front position between 2015 and 2019 mapped from Landsat 8 OLI acquired on 16th February 2019, 1st February 2018, 24th February 2016 and 7th February 2015 and from Sentinel 2A acquired on 23rd February 2017 (Background: Landsat 8 OLI – 16th February 2019). (b) Scott Glacier ice front 26th November 2020, the central portion of the front has lost some of the blocks held in place by fast ice and an area immediately to the south of Mill Island (Landsat 8 OLI), (c) Scott Glacier ice front 27th February 2021, the fast ice has broken up and a larger block is separated (Sentinel 2B), (d) Scott Glacier ice front 21st February 2022, (Sentinel 2B).

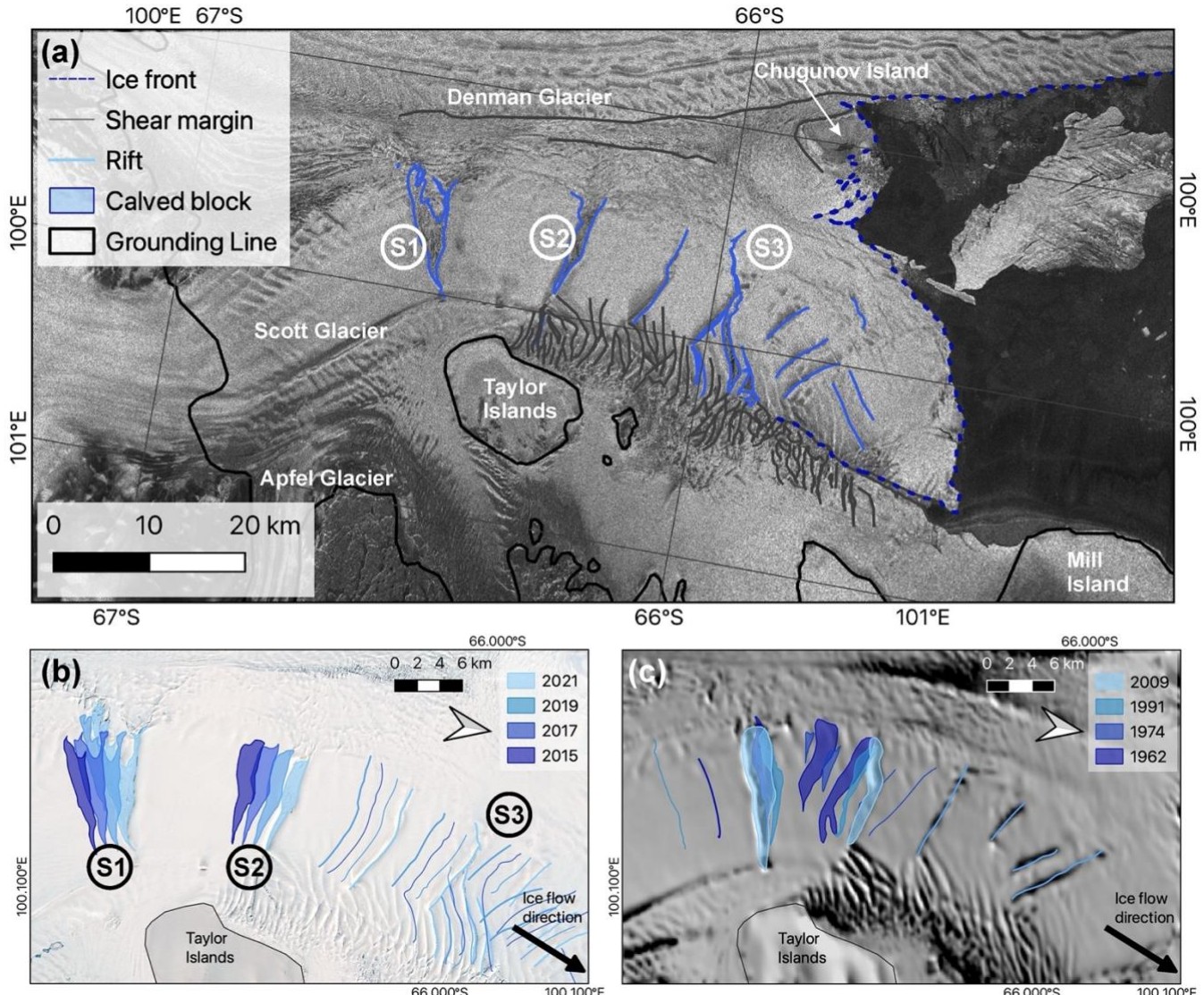

**Figure 4: Evolution of the rifts on Scott Glacier from (a) 2015-2022 mapped from Sentinel 2B (acquired 27th February 2021),**
**Landsat 8 OLI (acquired 16th February 2019 and 25th March 2015) and from Sentinel 2A (acquired 23rd February 2017)**
**(Background: Sentinel 2B acquired 27th February 2021) and (b) 1962-2009 mapped from ARGON KH5 (acquired 16th May 1962),**
**Landsat 1 MMS (acquired 27th February 1974) and Landsat 5 TM acquired (10th -12th February 1991) and from MODIS MOA**
**(acquired 1st November – 28th February 2009) – Modis MOA (Scambos et al., 2007) (Background: MODIS MOA). (c) The floating**
**portion of Scott Glacier in June 2022, highlighting the iceberg that calved from the western front in April 2022 and the rifting across**
**the eastern portion of the front towards Chugunov Island (Background: Sentinel 1A acquired 6th June 2022).**

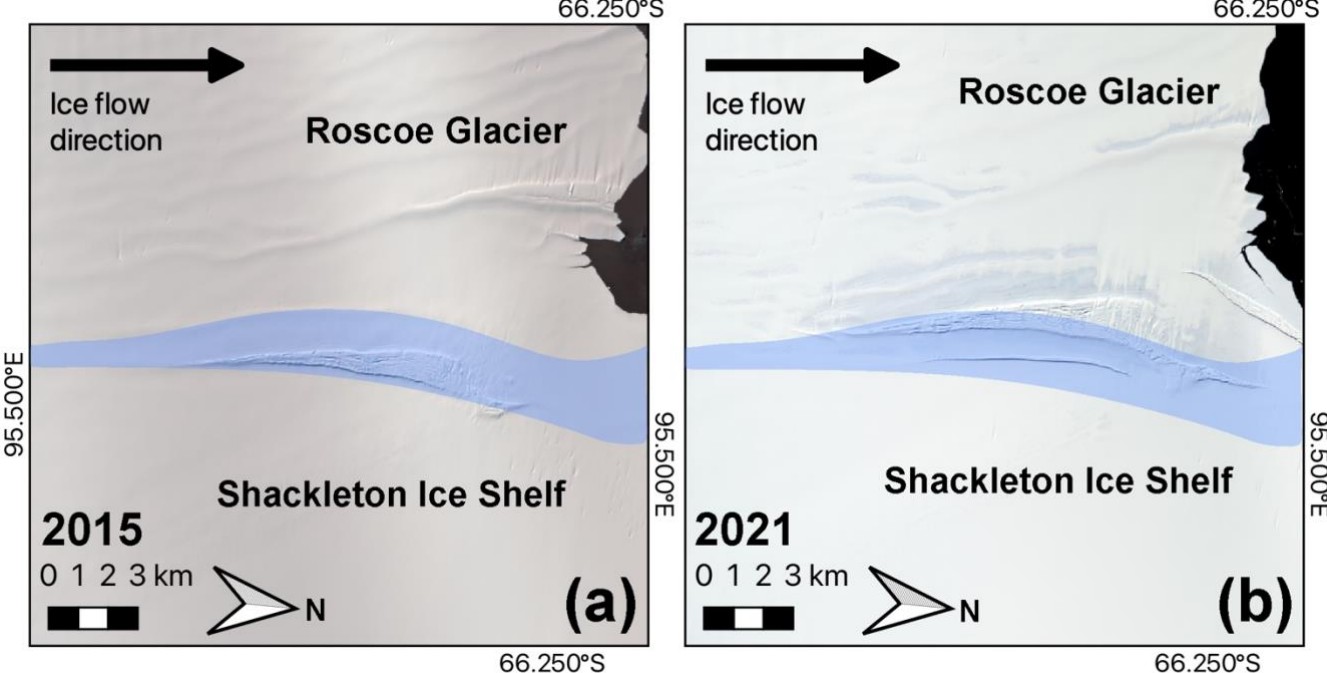

**Figure 5: Rift opening in the vicinity of the shear margin between the Shackleton Ice Shelf and Roscoe Glacier (location marked in blue) between (a) 2015 (Background: Landsat 8 OLI acquired 14th March 2015 and (b) 2021 (Sentinel 2B acquired 13[th] February 2021).**

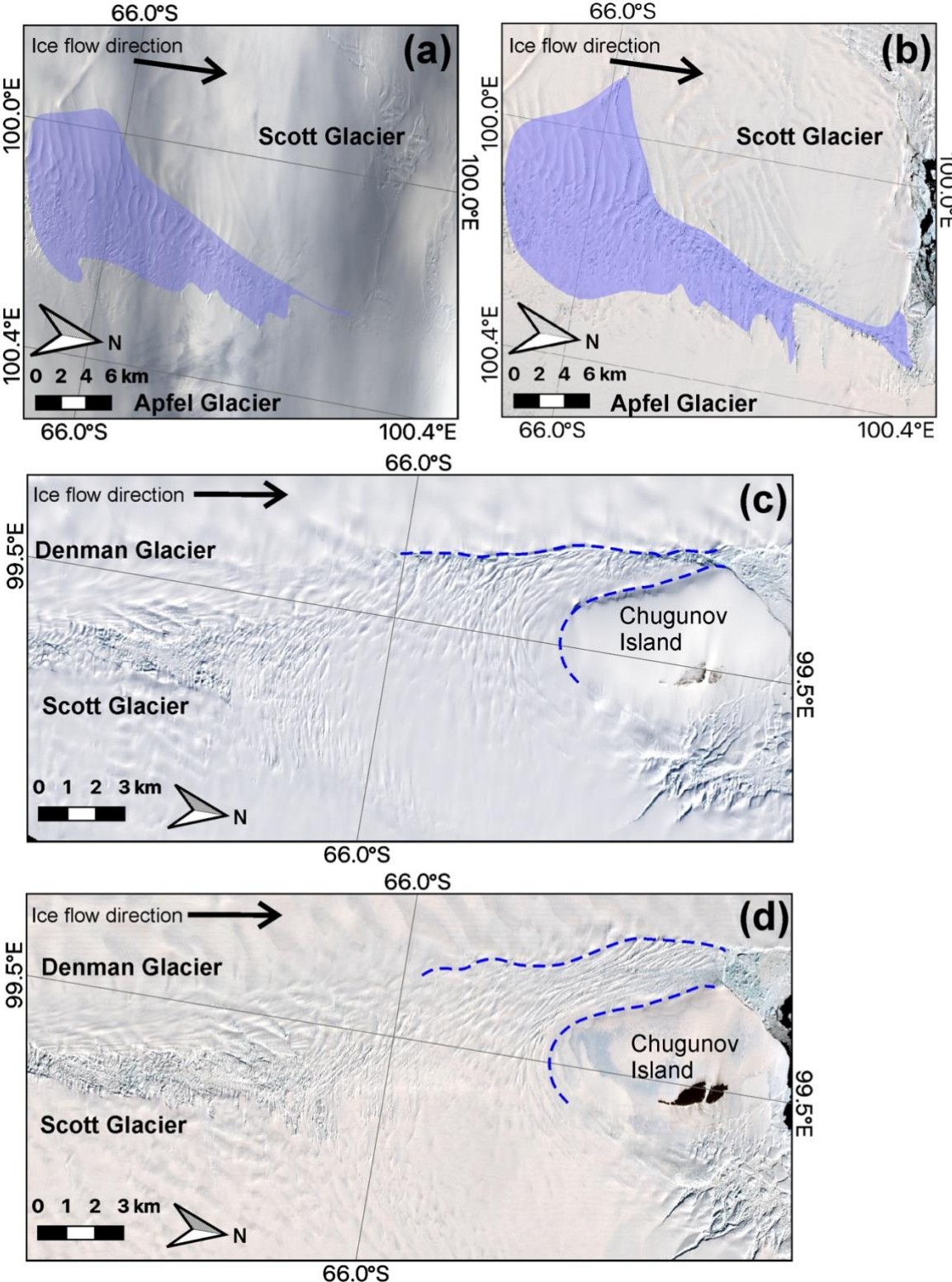

**Figure 6: (a) The 2015 extent of the Scott-Apfel Glacier shear margin highlighted in blue (Background Landsat 8 OLI acquired 25th March). (b) The 2021 extent of the Scott-Apfel Glacier shear margin highlighted in blue (Background: Sentinel 2B 13th February 2021). (c) Scott-Denman Glacier shear margin in 2015, the dashed blue line highlights the position ad shape of the margin. (Background: Landsat 8 OLI acquired February 2015). (d) The Scott-Denman Glacier shear margin in 2021, the dashed blue line highlights the widening of the shear margin into the Denman Glacier (Background: Sentinel 2B acquired 27th February 2021).**


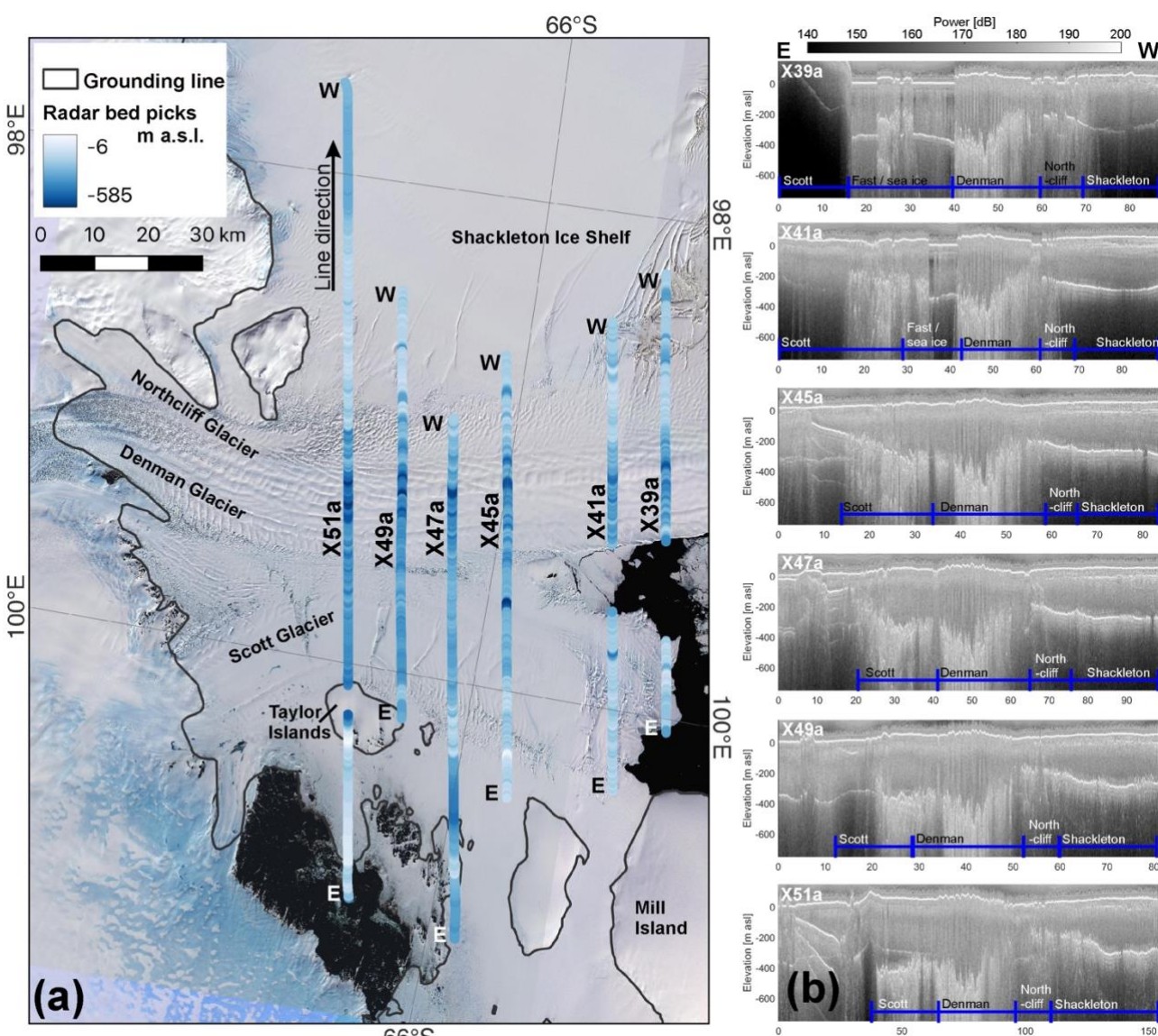

**Figure 7: (a) The location and ice base below sea level of 6 ICECAP radar lines acquired in December 2016 (Background Sentinel 2A acquired 23rd February and 1st March 2017). (b) Annotated ICECAP radar lines (position and east-west line direction shown in a).**

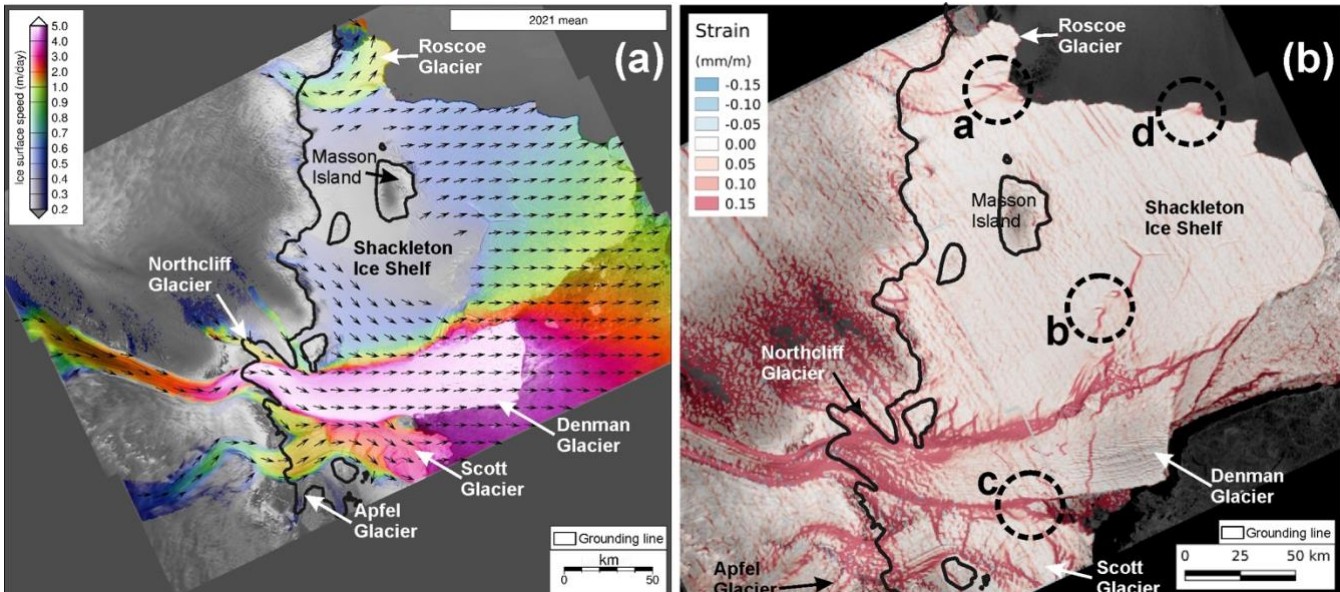

**Figure 8: (a) Mean speed for 2022 with velocity arrows. (b) Magnitude of the principal strain rate of Shackleton system derived from Sentinel-1 derived mean velocity data over the period of observation. N.B. The area of high strain upstream of Northcliff Glacier is likely noise as not filtering was applied to supress noise. The feature down flow of pinning point c on the Denman Tongue is an artefact, we see no evidence of rifting in the remote sensing data in this region (e.g., Fig. 2b, 7a).**

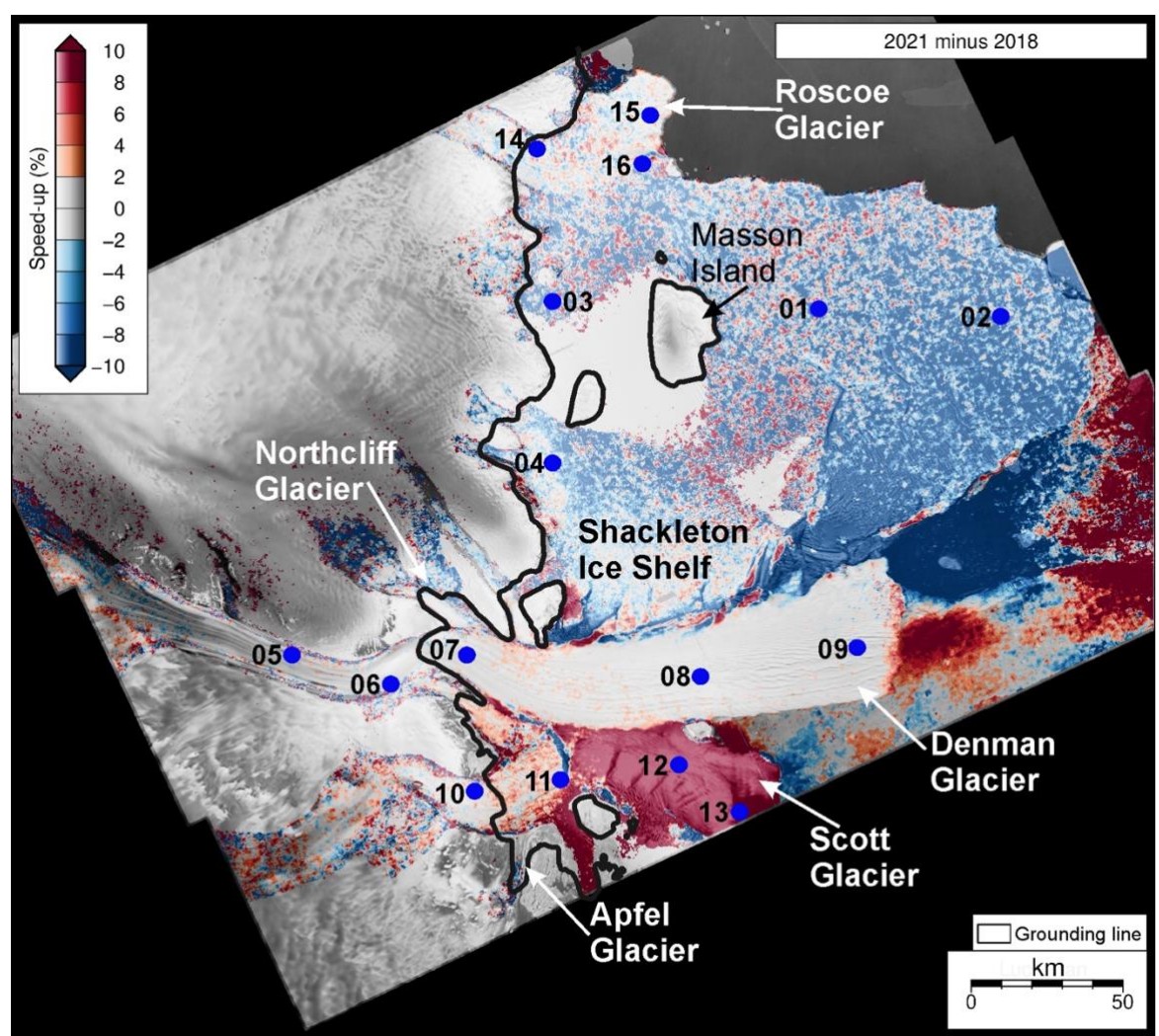

**Figure 9: Percentage difference in mean speed between 2021 and 2018, scaled between +/- 10%, with point locations illustrating the ice speed timeseries in Figure 11.**

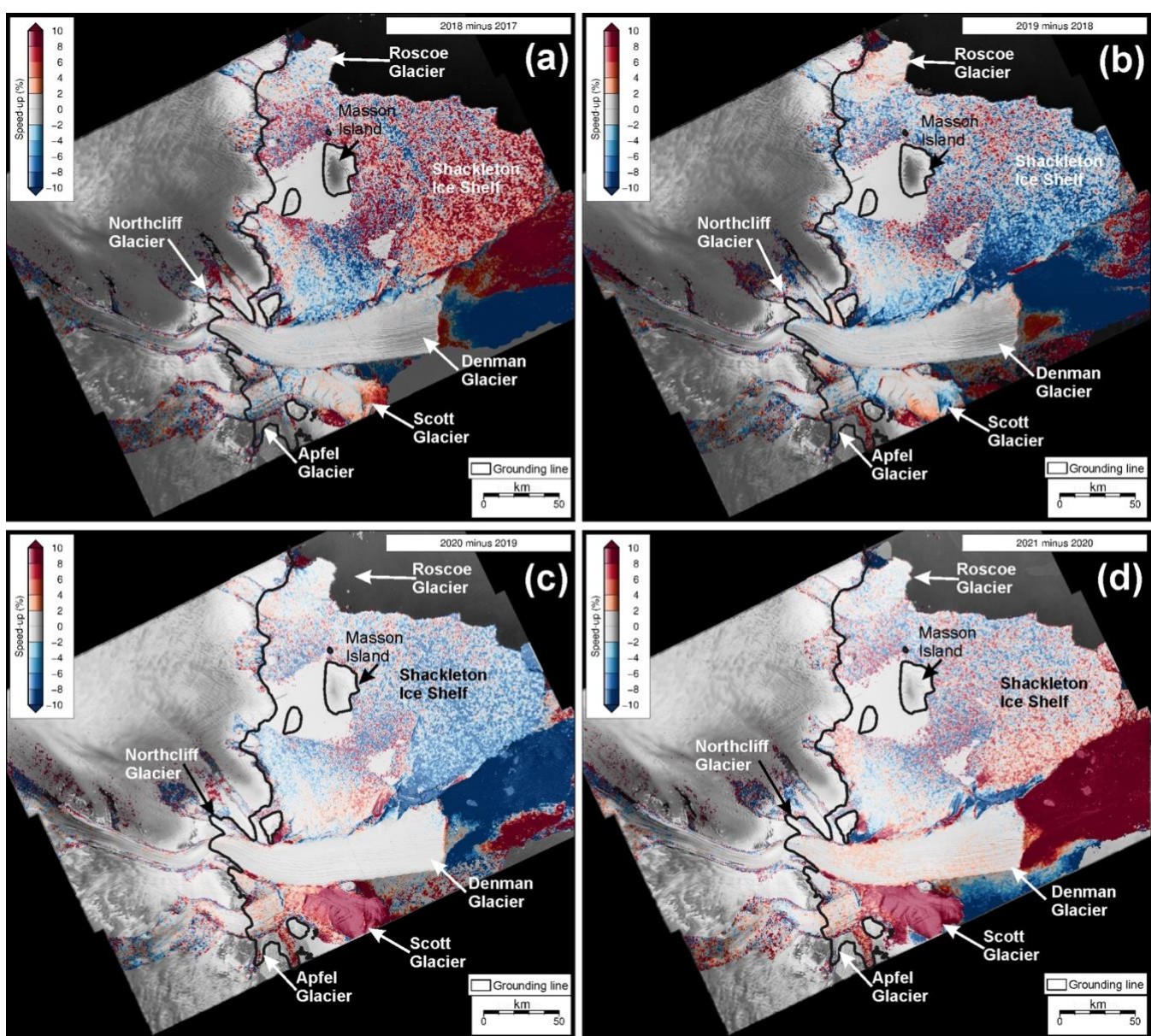

**Figure 10: Percentage difference in mean speed between (a) 2018-17, (b) 2019-18, (c) 2020-19 and (d) 2021-20 scaled between +/-10%.**


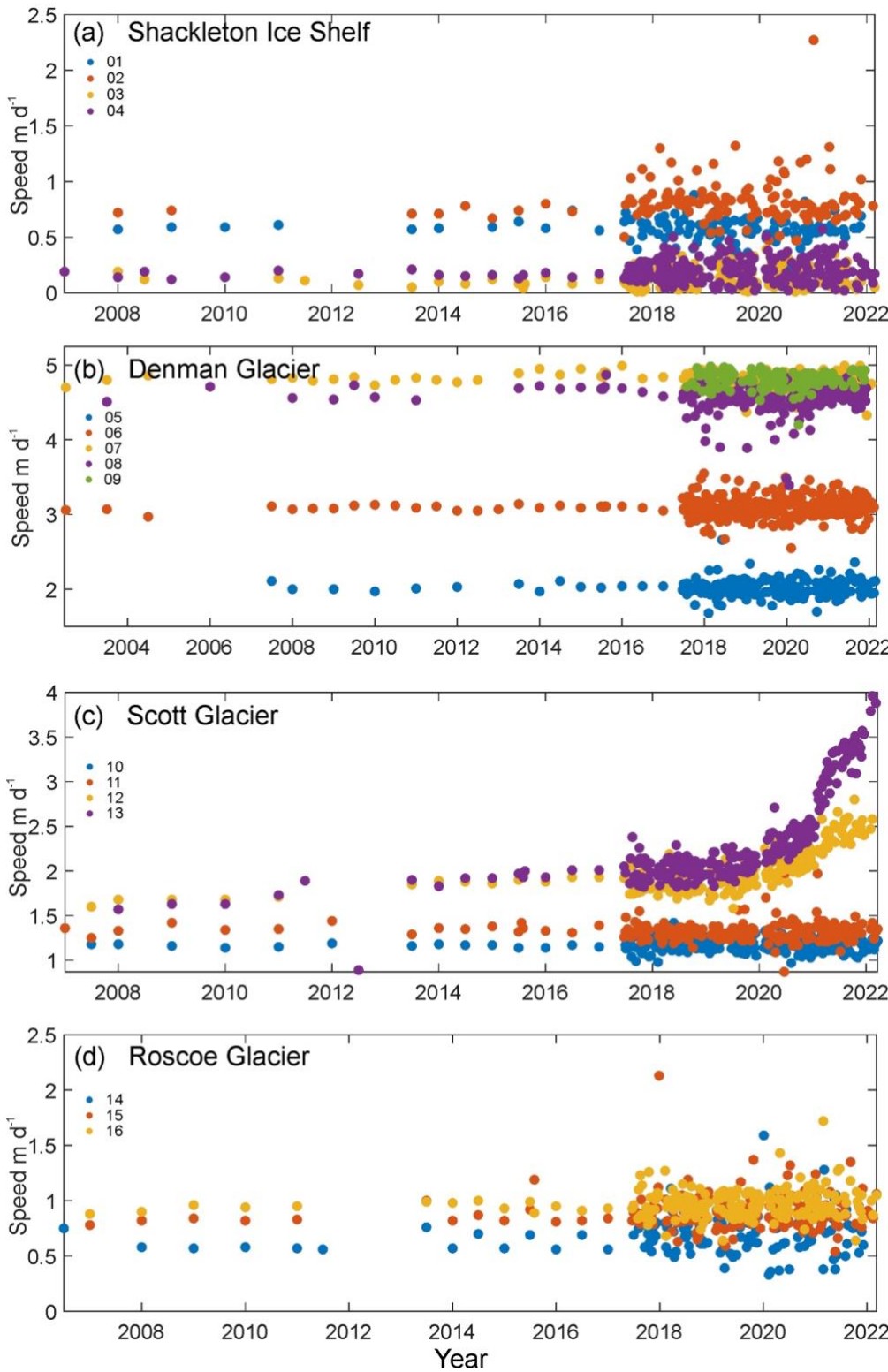

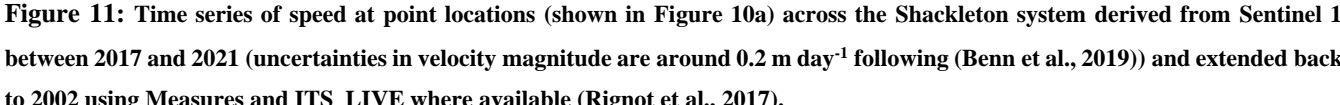

**Figure 11: Time series of speed at point locations (shown in Figure 10a) across the Shackleton system derived from Sentinel 1 between 2017 and 2021 (uncertainties in velocity magnitude are around 0.2 m day$^{-1}$ following (Benn et al., 2019)) and extended back to 2002 using Measures and ITS_LIVE where available (Rignot et al., 2017).**

