# Peer review of "Glaciological history and structural evolution of the Shackleton Ice Shelf System, East Antarctica, over the past 60 years"

_The Cryosphere, 2021_

## Referee Comment (RC1)

**Glaciological setting of the Queen Mary and Knox coasts, East Antarctica, over the past 60 years, and implied dynamic stability of the Shackleton system**
Thompson et al. (2021)

**Summary**:
The manuscript by Dr. S. Thompson and colleagues presents a two-pronged study aimed at (1) reconstructing the 60-year surface feature record and 2014-2021 surface ice velocity field of the Shackleton Ice Shelf System via analysis of satellite imagery, and (2) modeling the future response of the upstream grounded glaciers feeding the Shackleton Ice Shelf System to near-instantaneous disintegration of this ice shelf via a 400-year projection made using the BISICLES ice sheet model. The authors conclude that the Shackleton Ice Shelf System has not changed significantly over the 60-year observational period (aside from a localized acceleration in surface ice velocity near the ice front of Scott Glacier) and that the future upstream glacier response to collapse of the Shackleton Ice Shelf is minimal relative to changes projected across other East Antarctic basins.

I find the extension of the surface-observational record both *spatially* (to the neighboring Scott Glacier, Roscoe Glacier, and greater Shackleton Ice Shelf region) and *temporally* (from 2017-2021) to be the primary strength of this manuscript, as this information is very useful to the ice sheet modeling community. However, I have significant concerns regarding the scope of the paper and the applicability of the numerical modeling work, which make the manuscript difficult to follow. First, I find significant overlap in the analysis of surface features of the Denman Ice Tongue between this manuscript and that of Miles et al. (2021) (e.g. ice front positions, patterns of rifts, and location of pinning points on the Denman Ice tongue, as well as the calving of the large tabular icebergs in the 1940's and in 1984). While the manuscript does properly cite Miles et al. (2021), the repetition of the analyses and findings makes up a significant portion of this study and thus reduces the novelty of the manuscript. The manuscript should be reorganized to have a greater focus on the spatial and temporal extension of the surface observation record.

In addition, the numerical modeling portion of this manuscript is rather disconnected to the scope and findings of the rest of the paper and is not robust enough to support the authors' conclusions. The first half of the manuscript analyzes short-term and fine-scale changes of features on the floating portion of the Shackleton Ice Shelf System; however, the authors then model the 400-year response of the grounded regions of mainly Denman Glacier to near-instantaneous disintegration of this ice shelf. This disconnect between the focus on observing small-scale ice shelf features across the entire Shackleton Ice Shelf System and modeling grounding line retreat and volume loss of Denman Glacier (without mentioning of the response Scott and Roscoe Glacier) makes following the progression of the manuscript very difficult. If modeling is going to be included in this study, it needs to complement the rest of the manuscript (i.e. model how future ice flow responds to changes in the surface features discussed in the first half of the manuscript). Furthermore, from the analysis of a single model simulation, the authors imply that the Queen Mary and Knox coasts are relatively stable and insensitive to reasonable forcing in the next 400 years (see L313, I am also assuming this is the "implied dynamic stability" referenced in the title). I don't believe the authors can claim stability of the system and make a statement about sensitivity without modeling the system's response to realistic forcing

perturbations. Overall, I believe the modeling portion of this manuscript needs to be either redone so that it supports the observational-focus of the paper or separated and made the focus of a secondary manuscript.

It is apparent that the authors have put a lot of effort into the text and figures in the manuscript; however, because of my significant concerns over the scope of the manuscript, its connection to the modelling work, and the key takeaways, I suggest that major revisions (or perhaps a resubmission) are needed before the manuscript can be considered for publication in The-Cryosphere.

**General Comments**:

- **Title**: I don't believe the title accurately describes the presented work. I am unsure what the authors mean by "glaciological setting", I think wording that describes the analysis would be better suited to use in the title. I also think it is a bit misleading to claim that the authors are studying the entire Queen Mary and Knox coasts, when only the Shackleton Ice Shelf system is analyzed. Lastly, I am not sure what the authors mean by "implied dynamic stability". Is this stability over the entire 60-year observational period (which would be inaccurate because grounding line retreat (~5 km, Barancato et al., 2020), floating and grounded ice accelerations (Miles et al., 2021; Rignot et al., 2019), and accelerated ice discharge (Rignot et al. 2019) have been observed over this timeframe), over the 400-year modeling period (which, as stated above, I do not think the authors can claim based on the results presented), or between 2018-2021 (following the ice velocity results)? It is difficult to suggest a new title right now because significant changes to the scope of the manuscript need to be made.

- **Ice sheet model validation**: As the manuscript presents one of the first regional ice sheet models to make future projections of the Shackleton Ice Shelf system, it is critical that it is properly described and validated. It is not enough to only show the mismatch of observed and modeled surface ice velocity in order to validate your ice sheet model, one also needs to know how the modeled and observed grounding line positions and ice discharge values compare. In the initial model solution, there are extensive grounded regions along Denman's ice tongue and floating pockets along Denman's grounded ice stream (figure 11) that are not seen in observations (Barancato et al. 2021; Morlighem et al., 2020). Such errors in the initialization of the model can propagate to the transient solutions, so it is critical to have a well-calibrated model that matches present day observations. The ice sheet model description section (L130-L177) is lacking details that would be needed to ensure that this modelling work reproducible. Some examples of missing methodological descriptions are as follows: which 2D stress balance approximation is used (e.g. SIA, SSA, a combination of both), do the ice stiffness and basal friction coefficient change in time, how is the grounding line tracked (e.g. sub-element parameterization), how is basal melt applied numerically to partially floating elements if using a sub-element grounding line parameterization (e.g. to the entire element, to only the floating part of the element), etc. These details are very important and should be included (perhaps it would be better to give a complete model description as a supplement or appendix). For examples of the types of information needed, one could

refer to the ISMIP6 Antarctica publication (Seroussi et al. 2020) or this recent manuscript submitted to The-Cryosphere Discussions (Castleman et al. 2021).

- **Units of speed**: When referencing speeds of both ice and rift/ice front propagation in the main text and figures, the authors switch between m/day and m/year (see lines 258 and line 270 for examples of each). For ice speed, the convention is m/year, so I think it would be best to abide by this convention so that your results can be easily compared to other values in the literature (change in both the text and in figures). Also, when referencing the unit "year", please stick to either "year" or "a", as both were used in the manuscript.
- **Data availability statement**: Please add a data availability statement at the end of the manuscript, as to abide by The-Cryosphere's data policy. All of the links in sections 2.1 and 2.2 should be moved to the data availability statement. In addition, a link to the BISICLES ice flow model, as well as links to all datasets used in the simulation, should be added to this statement if the modeling portion is to remain in the manuscript.
- **Grammar**: When reading through the manuscript, I noticed a fair amount of spelling and grammar mistakes (especially missing commas, which would help the readability of the text). I tried to point them out as I found them in the specific comments, but it is possible I missed a few!

**Specific Comments**:
- L13-L38: In general, I think the abstract is a bit long and should be condensed. Below, I suggested a few sentences that can be removed and/or shortened.
- L15: change to ``. . . on understanding the controls driving Denman Glacier's dynamic evolution, although . . .''
- L17: Shackleton Ice Shelf (use capitalization because it is a proper name)
- L22-L23: Remove "in response to coupled ocean and atmospheric forcing". Coupled forcing suggests that your ice sheet model is coupled to an atmosphere and/or ocean model, which it is not.
- L31: I make note of this later in the results section, but the authors should not use real years to describe the output of their modeling work because it is not a realistic simulation. Instead of saying "in the third century from now", it would be better to say "in approximately 300 years into the model simulation."
- L31: Please check the computation of the 6 cm of sea level rise, I computed 40 Tt = 40000 Gt / 3600 [Gt/cm] = 11.11 cm sea level rise equivalent ice mass, but it is possible that my math is off! Is this the sea level contribution from just Denman Glacier, or from the entire model domain? I believe this is from the whole domain; however, in the previous sentence, you discuss the grounding line of Denman Glacier, so it is a bit confusing. Please specify.
- L32: I would hesitate to say that 6 cm of global sea level rise equivalent ice volume loss is "small" in comparison to other areas of East Antarctica. First, I don't believe there are any published studies that have run regional transient simulation of the EAIS through 2400, so we cannot compare. Also, 6 cm is on the upper limit of the ISMIP6 projected contribution of the entire Antarctic Ice Sheet to global sea levels by 2100, so this contribution from a single EAIS glacier by 2400 must be fairly significant.

- L32-L34: The sentence "it is clear . . . Shackleton system" can be removed.
- L34: Here you conclude that there is potential vulnerability of the system to accelerating retreat, but further along in the manuscript (L313), you say that the modeled domain is relatively stable and insensitive to reasonable forcing in the next 400 years. These statements conflict and left me confused about the message of the manuscript. Perhaps it would be more consistent with the rest of the manuscript to say that these data are needed to improve model initialization and validation.
- L34: Insert comma after "accelerating retreat".
- L41-L44: These first two sentences can be combined and condensed, which I think would be a bit easier on the reader. Perhaps something like: "It has long been perceived that the East Antarctic Ice Sheet is the stable sector of Antarctica (citations); however, it has now emerged that the Aurora and Wilkes subglacial basins of the EAIS have been contributing to sea level rise since at least the 1980s, with Aurora contributing 1.9 mm and Wilkes contributing 0.6 mm (citations)."
- L45: Insert comma after "WAIS"
- L45 and L47: You are referencing both BedMachine Antarctica (Morlighem et al., 2020) and Bedmap2 (Fretwell et al., 2013) for your values of sea level potential. As BedMachine is the most up-to-date dataset, I would stick to just using the BedMachine citation throughout the manuscript (unless of course you are using the BedMap2 dataset in the paper).
- L52: Change "it is supplied by . . ." to "Major outlet glaciers drain into this ice shelf system, including Denman, Scott, Northcliffe, Roscoe, and Apfel Glaciers."
- L57: Cite Morlighem et al. (2020) instead of Rignot et al. (2019), as the BedMachine publication lists the most updated inventories of glacial ice volume.
- L65: Change "just above" to "just upstream of"
- L73: Adusumilli et al. (2020) show melt rates peaking at approximately 120 m/yr along Denman Glacier's deep grounding zone. Please check the value reported in your manuscript (6 m/yr), I think this might be a typo.
- L82-L85: Please remove "A satisfactory explanation . . . with the nearby Totten Glacier." I think this interrupts the flow of the introduction and does not serve the rest of the paper, as the focus is not to determine where the high melt rates are being forced from.
- L92: What does it mean to put previously observed dynamic changes in the Shackleton system into the wider regional context of the Queen Mary and Knox coasts? The observational and modelling components of this study do not investigate changes beyond the Shackleton Ice Shelf System, so I think that claiming to frame the regional context of the entire Queen Mary and Knox coasts is a bit misleading. As stated above, I think the really exciting science presented here is the extension of the observational record to other sectors of the Shackleton Ice Shelf and to 2021. So I think this sentence should reflect that.
- L95: I do not think we are testing the sensitivity of the domain, as this would require further model runs (such as a control simulation and variance of the ocean forcing).
- L96: Remove "in response to coupled ocean and atmospheric forcing"

- L101-L120: Remove links in the main manuscript and add them into a proper data availability statement at the end of the manuscript (see general comments).
- L107: The "th" on 10th should be a super-script.
- L111: change ";" to ":"
- L100 and L112: I think the sentences would read better if you did not use the parentheses at the end of the sentence. For instance, L100 would read as: " . . . using standard GIS techniques following the methodology of Glasser et al. (2009)." The multiple sets of parentheses is confusing for the reader.
- L120: Insert comma between "methods" and "feature tracking"
- L121: "We use image . . ." Use present tense
- L124: " . . . and the quality of the velocity map is maximized"
- L126: Change "allowed" to "allows"
- L146: Change "horizontal rate of strain tensor" to "horizontal strain rate tensor"
- L151: Change "rate-strain" to "strain-rate"
- L163: Please give references to previous BISICLES studies that have used this initialization method.
- L173: Change to "The single future simulation follows the methodology of Matin et al. (2019)."
- L174: "Under" should be lowercase
- L174: I had assumed that the melt rate was 1000 m/yr across all floating ice, but here you say that the melt rate reaches 1000 m/yr in places. How is the basal melt rate computed? Does the melt rate vary in space and/or with the geometry of the ice shelf?
- L195: These rift-systems along the Shackleton Ice Shelf are really fascinating! It is so interesting that the two rift-systems have almost identical shapes (with system-1 being larger than that of system-2).
- L213: Cite figure 1b here, it shows the high concentration of surface features on the Denman Ice Tongue very well.
- L218: Should this first sentence be citing figure 5 (figure 6 shows the rift on the Shackleton-Roscoe shear margin)?
- L222: I am having trouble figuring out which rift you are describing in this line (the one on the western side of Scott Glacier). Since you are highlighting this particular rift, it would be helpful to highlight it or point it out in figure 5c if possible (perhaps an arrow or pointer next to it so that it is easily identifiable by the reader).
- L229: Replace "some changes" with wording that is a bit more definitive.
- L244: Why did you decide to compute the velocity difference between one year (2019-2020)? It seems like this would not give very interesting results because that is not enough time to for the system to respond to a forcing perturbation (aside from the northern point of Scott Glacier's floating extension, which looks like perhaps it is undergoing a calving event).
- L254: Change m/day to m day$^{-1}$ to be consistent with the rest of the paper (ultimately should be m year$^{-1}$, see general comments)
- L255: "Speeds ~ 10 km either side of the grounding line . . ." confuses me a bit. This sentence makes it sound like you are talking about grounded ice as well (since 10 km on

either side of the grounding line would extend 10 km upstream into grounded ice), but I believe you are talking about points 10 and 11 in figure 8a. Perhaps it would be better to say "Speeds up to __ km downstream of the grounding line show . . ." . I also think it would be helpful to reference the specific points in figure 8a that you are discussing (e.g. in L258, "close to the ice front (point 13 in fig. 8a)").

- L265: The format of ((b) in Fig. 10)(Furst et al., 2015) is a bit crowded. Instead of using double parentheses, change to (label-b in Fig. 10, Furst et al., 2015). Same with L266 and L267.
- L267: Change "rise in the ocean floor" to "local topographic high in ocean bathymetry".
- L269-L283: When describing the model results, I would stray away from using actual years (e.g. ". . . of Denman Glacier occurs after 2150 . . ."), as you are modeling with unrealistic forcing. Instead, I would change this to something like ". . . of Denman Glacier occurs 150 years into the model simulation . . .".
- L276: The dynamic response of the system seems pretty significant, as Denman Glacier retreats more than 100 km upstream and Denman and Scott Glaciers end up connecting around a topographic high.
- L285: This first sentence of the discussion section contradicts existing literature (e.g. Barancato et al. 2020; Miles et al. 2021), which have cited patterns of grounding line retreat and ice velocity change that appear to be ocean induced since the 1970s or so. This study only looked at changes in surface features through the 60 year observational period and velocity changes over the past ~15 years; however, it seems that changes outside of those presented in this paper are occurring over those timescales. As such, I don't think the authors can claim, based on the presented manuscript, that the Queen Mary and Knox coasts have not changed significantly in the last 60 years.
- L291: change "groundling" to "grounding"
- L310-315: I don't think the authors can make this claim based on a single transient model run of the Denman/Shackleton system. The Denman Glacier grounding line is currently retreating under present day forcing conditions (Barancato et al. 2020) and this retreat could be susceptible to the marine ice sheet instability, as the bed upstream of the current grounding line position is retrograde. Your model results show > 100 km of grounding line retreat by 2310 over Denman Glacier, which is significant. However, you did not test the response of the system to realistic forcing over the same timeframe, so we cannot make a statement on the sensitivity of the system. Lastly, the Aurora and Wilkes subglacial basins are not included in the model domain, so this last statement is a speculative conclusion rather than one based on your modeling results and should not be included in the manuscript.
- L317-L319: It is a great addition to include model limitations in the discussion section; however, unless you are running a thermal model, I do not believe that the geothermal heat flux is used by the ice sheet model. In addition, the ocean conditions and bathymetry will not impact your model run because you assume near-instantaneous disintegration of the floating ice shelf. In the modeling results that you presented, I would expect the results to be primarily impacted by mesh resolution (I am assuming you are not using adaptive mesh refinement, so as the grounding line retreats upstream, the size of the

elements will most likely become larger), poorly constrained basal friction and ice stiffness parameters that do not change in time, use of a 2D stress balance approximation instead of a higher order model, etc. It would be helpful to the reader to know exactly how the limitations you listed impact your model (e.g. poorly constrained basal hydrology leads to a poorly constrained basal friction parameter, ice properties impact the ice stiffness parameter, etc.).

- L321-L337: This paragraph lost me a bit. I understand the comparison to Totten Glacier, but I do not think it is appropriate to dive into such a detailed discussion of the subglacial conditions of Queen Mary Land because it does not connect to the rest of the paper. If the authors want to speculate on the subglacial conditions, they need to tie it back to the conclusions of the paper (i.e. its impact on enhanced ice shelf basal melting rates near the grounding line, reducing basal friction at the ice-bed interface, etc.). Without that obvious connection to tie back to the rest of the paper, this paragraph seems out of place and left me confused.
- L340: See previous comment about L285.

**Figure Comments**:
- General comment: units of ice speed should be changed from m/day to m/year to fit the convention of existing literature.
- Figure 1: In panel-a, the color bar should be flipped upside down so that the color-bar is in increasing order. I believe this panel could also be zoomed in to show more detail of the Knox Basin, rather than showing the entirety of the Sabrina and Aurora basin, as these are not the focus of the paper. Some of the text in panel-b is tough to read because of the numerous crevasse-lines (i.e. Scott Glacier, Denman Glacier). In the legend, I believe the last entry should be "grounded ice" rather than "grounding line".
- Figure 2: I think this figure can be removed so that it is combined with figure 1. Perhaps in figure 1, you could include a dotted or dashed line to denote the model domain.
- In Figures 3-7, it is difficult to know where we are on the Shackleton Ice Shelf System. It would be helpful to have a small inset in each figure that shows the entire floating ice shelf and a box that shows the domain of each figure.
- Figure 9: I really like this figure, I found myself going back to it throughout reading the manuscript because it does a great job at illustrating your velocity results. Very well done!
- Figure 10: This figure is blurry for some reason, please update with a clear copy.
- Figure 12: Change from real years to model simulation years and change font color so that it is easy readable. Also, in the caption you say that you disintegrate "all floating ice in the Denman Glacier shelf", which confuses me because in the methods section, you say you apply a ~1000 m/yr melt rate across all floating ice (not just Denman's ice tongue). Please specify if you are applying this melt rate to all floating ice.
- Figure 13: This figure is also blurry and the x-axis label got cut off. In the top panel, it would be useful to use the other y-axis to show the volume loss in units of cm sea level rise equivalent. Also, in the caption, you say that this is discharge from Denman Glacier, but it is not clear over which bounds the authors are integrating the volume loss (i.e. are

the authors using a flux gate along the Denman Glacier grounding line, or it this actually ice volume loss integrated across the entire domain?). It is a bit odd to me that the main goal of this paper was to observe and model the entire Shackleton Ice Shelf System, but in the captions of figures 12 and 13, the description is only about Denman Glacier (what about retreat and acceleration of Scott Glacier?). This description/analysis is inconsistent with the scope of the paper.

**References**:

Adusumilli, S., Fricker, H.A., Medley, B. *et al.* Interannual variations in meltwater input to the Southern Ocean from Antarctic ice shelves. *Nat. Geosci.* **13,** 616–620 (2020). https://doi.org/10.1038/s41561-020-0616-z

Brancato, V., Rignot, E., Milillo, P., Morlighem, M., Mouginot, J., An, L., et al. (2020). Grounding line retreat of Denman Glacier, East Antarctica, measured with COSMO-SkyMed radar interferometry data. *Geophysical Research Letters*, **47**, e2019GL086291. https://doi.org/10.1029/2019GL086291

Castleman, B. A., Schlegel, N.-J., Caron, L., Larour, E., and Khazendar, A.: Derivation of bedrock topography measurement requirements for the reduction of uncertainty in ice sheet model projections of Thwaites Glacier, The Cryosphere Discuss. [preprint], https://doi.org/10.5194/tc-2021-274, in review, 2021.

Fretwell, P., Pritchard, H. D., Vaughan, D. G. *et al*.: Bedmap2: improved ice bed, surface and thickness datasets for Antarctica, The Cryosphere, **7**, 375–393, https://doi.org/10.5194/tc-7-375-2013, 2013.

Miles, B. W. J., Jordan, J. R., Stokes, C. R., Jamieson, S. S. R., Gudmundsson, G. H., and Jenkins, A.: Recent acceleration of Denman Glacier (1972–2017), East Antarctica, driven by grounding line retreat and changes in ice tongue configuration, The Cryosphere, **15**, 663–676, https://doi.org/10.5194/tc-15-663-2021, 2021.

Morlighem, M., Rignot, E., Binder, T. *et al.* Deep glacial troughs and stabilizing ridges unveiled beneath the margins of the Antarctic ice sheet. *Nat. Geosci.* **13,** 132–137 (2020). https://doi.org/10.1038/s41561-019-0510-8

Rignot, E., Mouginot, J., Scheuchl, B., van den Broeke, M., van Wessem, M. J., and Morlighem, M.: Four decades of Antarctic Ice Sheet mass balance from 1979–2017, P. Natl. Acad. Sci. USA, **116**, 1095–1103, https://doi.org/10.1073/pnas.1812883116, 2019.

Seroussi, H., Nowicki, S., Payne, A. J. *et al*.: ISMIP6 Antarctica: a multi-model ensemble of the Antarctic ice sheet evolution over the 21st century, The Cryosphere, **14**, 3033–3070, https://doi.org/10.5194/tc-14-3033-2020, 2020.

---

## Referee Comment (RC2)

This manuscript uses a suite of remote sensing data to map changes in ice extent, structure and velocity across the wider Shackleton system. The manuscript compliments recently published work showing recent grounding line retreat and acceleration of Denman Glacier, but also includes detailed and novel observations across the wider understudied Shackleton system. The authors use these observations to conclude that there has been limited change across the Shackleton system across the observational time period. They also then simulate the response the Shackleton system to a hypothetical loss of floating ice to demonstrate the systems sensitivity to any future ice shelf loss.

Overall, I think the manuscript contains some interesting and novel observations of the Shackleton system that are worthy of publication. In particular I think the 50 year record of rifting evolution across the Shackleton Ice Shelf is a very nice contribution. The background here is while there has been some recent studies focussing on Denman Glacier, we know very little about the recent behaviour of the many other glaciers that feed the wider Shackleton system, so these results are valuable. However, at the moment I think these interesting results are somewhat lost in the manuscript. It seems like quite a jump and a distraction from discussing these detailed annual scale observations, to discussing the response of the Shackleton system to the hypothetical loss of all floating ice 400 years in the future, which then turns into a discussion as to how the deep trough of Denman may favour vigorous channelization of the subglacial meltwater system close to the grounding line. At the moment I am not sure what the main focus of the manuscript is. I think the manuscript would benefit from being more streamlined, with a greater focus on the novel observations. I have included some more detailed comments below:

**Observations**

The authors state that there have been no significant annual variations in ice flow speed across the Shackleton system. I would argue that the use of 'significant' is not appropriate, what is 'significant' variations greater than 50 m yr$^{-1}$, 100 m yr$^{-1}$ etc?. The plots in Figure 9 give a good overview of the longer-term changes in ice flow speed across the region. However, because they are in m/day and the scales are somewhat stretched it is difficult to determine if there has or has not been any annual variations in ice flow speed. A variation of 0.3 m/day equates to around 100m yr, which would be larger than the uncertainty of the velocity products and would be an interesting result. Are there similar scale variations, particularly in the faster flowing sections of the Shackleton system, to my eyes it looks that there could be, but I could be wrong, I really cannot tell from the plot alone?

In Figure 8 the authors plot an ice speed difference map between 2019 and 2020, I think to illustrate the acceleration of parts of the Scott Glacier. I think these plots are useful in giving a broad overview of the changes in ice speed. Could they also do this over a longer time

period, maybe 2000-2010 and 2010-2020 (brackets whatever the availability of velocity data allows)? This would probably be the best visualization of the speed changes over the observational period.

One of the most striking observations is the migration of the shear margin near Chungunov Island over just a few years. This is a somewhat unique observation. The authors have collated this wide range of velocity data, while they have tended to focus of ice speed, could they also focus on the velocity directional data? Is this change in shear margin caused by the whole Denman ice tongue 'wobbling' or a more localised change to do with Scott?

**Modelling**

My expertise lies in remote sensing, so I am not in a position to comment on the methodological details of the modelling. But I did find the description of the modelling experiment carried out to be lacking. In the discussion it is stated that:

'The upper limit scenario of forcing in our BISICLES model runs suggests that noticeable grounding line retreat occurs in the Denman Glacier over the simulated 400-year time period'

But in the methods section there is no mention of an upper or lower limit scenario, nor any mention of the timescales of the simulation. Aside from the basic description of model, there is only a very limited description of simulation. It is essential that the details here are expanded.

In wider point, while I think it could be a useful contribution to repeat the experiment in Martin et al., but with BedMachine, I did feel the modelling appeared as somewhat left field in the manuscript and appears as a bit of a jump in the discussion from the main body of observations. The general tone of the manuscript is that very detailed remote sensing observations have shown limited changes in the floating ice in the Shackleton system... But then the manuscript jumps to.. 'now we simulate the unrealistic loss of all floating ice in the Shackleton system'... The scientific rationale for this is unclear to me? Of course, it is entirely up to the authors, but I would point out that I think the detailed observations have the potential to be a nice contribution alone.

Line 59: I would not describe the acceleration of Denman Glacier since the 1970s as a short-term fluctuation

Line 62: 'the glacier' – please clarify in the text that you are presumably referring to Denman Glacier.

Line 113: Landsat 1 was not mentioned in the above paragraph? Was it used?

Line 118-129: What software was used for the feature tracking?

Line 164: That of Rignot (2011) – Do you mean the MEASURES ice velocity mosaic?

Line 173-178: I think much more detail is needed here (see comment above)

Line 189: Not sure 'remarkable' would be the word I would use here

Line 243-245: Its nice to see the changes between 2019 and 2020, but I think you could also expand to include 2000-2010 and 2010-2020, maybe even 2000-2020

Line 250: Please use the correct citations for the Measures and ITS LIVE velocity products. I think the correct details can be found on each respective website.

Line 267: Is d Chugunov Island?

Line 269: The paragraph is discussing modelling results. Therefore I think it needs a new appropriate subsection, currently it is under 'ice flow speed'.

Line 285: 'Not changed significantly' What is 'significantly'? Observations of Denman have shown that its grounding line has been retreating and that it has been accelerating and losing mass, this is contradictory.

Line 285-302: I think you could be more detailed in this section. One of the key results here is the lack of propagation of the rifting across the Shakleton over the past 50 years, with the exception of some small activity over the past few years. This is a key result. What does this mean and how does it tie into the wider body of literature? In other regions rifting evolution has been key to ice shelf stability? How important could the melange that fills these rifts be (see recent Larour Larsen C paper in PNAS)?, etc..

Line 321-338: I struggle to see the relevance of this section to the manuscript.

Fig 1a: The scale is a little difficult to make out because of the transparency

Fig 10: The high strain rate band going across the Denman ice tongue – presume there is no evidence of any recent rifting in this location?

---

## Author Comment (AC1)

We thank Anonymous Referee #1 for their considered and detailed review of our manuscript. Below we respond to each comment, with the anonymous referee's original comments shown in italic text and our response in blue text.

**Summary:** *The manuscript by Dr. S. Thompson and colleagues presents a two-pronged study aimed at (1) reconstructing the 60-year surface feature record and 2014-2021 surface ice velocity field of the Shackleton Ice Shelf System via analysis of satellite imagery, and (2) modeling the future response of the upstream grounded glaciers feeding the Shackleton Ice Shelf System to near-instantaneous disintegration of this ice shelf via a 400-year projection made using the BISICLES ice sheet model. The authors conclude that the Shackleton Ice Shelf System has not changed significantly over the 60-year observational period (aside from a localized acceleration in surface ice velocity near the ice front of Scott Glacier) and that the future upstream glacier response to collapse of the Shackleton Ice Shelf is minimal relative to changes projected across other East Antarctic basins.*

*I find the extension of the surface-observational record both spatially (to the neighboring Scott Glacier, Roscoe Glacier, and greater Shackleton Ice Shelf region) and temporally (from 2017- 2021) to be the primary strength of this manuscript, as this information is very useful to the ice sheet modeling community. However, I have significant concerns regarding the scope of the paper and the applicability of the numerical modeling work, which make the manuscript difficult to follow. First, I find significant overlap in the analysis of surface features of the Denman Ice Tongue between this manuscript and that of Miles et al. (2021) (e.g. ice front positions, patterns of rifts, and location of pinning points on the Denman Ice tongue, as well as the calving of the large tabular icebergs in the 1940's and in 1984). While the manuscript does properly cite Miles et al. (2021), the repetition of the analyses and findings makes up a significant portion of this study and thus reduces the novelty of the manuscript. The manuscript should be reorganized to have a greater focus on the spatial and temporal extension of the surface observation record.*

We will restructure the manuscript to clarify the novelty of the extension in spatial and temporal observations. Our aim was to provide full context to the observations we report and discuss, rather than simply citing previous work. We will make the distinction requested.

*In addition, the numerical modeling portion of this manuscript is rather disconnected to the scope and findings of the rest of the paper and is not robust enough to support the authors' conclusions. The first half of the manuscript analyzes short-term and fine-scale changes of features on the floating portion of the Shackleton Ice Shelf System; however, the authors then model the 400-year response of the grounded regions of mainly Denman Glacier to near- instantaneous disintegration of this ice shelf. This disconnect between the focus on observing small-scale ice shelf features across the entire Shackleton Ice Shelf System and modeling grounding line retreat and volume loss of Denman Glacier (without mentioning of the response Scott and Roscoe Glacier) makes following the progression of the manuscript very difficult. If modeling is going to be included in this study, it needs to complement the rest of the manuscript (i.e. model how future ice flow responds to changes in the surface features discussed in the first half of the manuscript). Furthermore, from the analysis of a single model simulation, the authors imply that the Queen Mary and Knox coasts are relatively stable and insensitive to reasonable forcing in the next 400 years (see L313, I am also assuming this is the "implied dynamic stability" referenced in the title). I don't believe the authors can claim stability of the system and make a statement about sensitivity without modeling the system's response to realistic forcing perturbations. Overall, I believe the modeling portion of this manuscript needs to be either redone so that it supports the observational-focus of the paper or separated and made the focus of a secondary manuscript.*

*It is apparent that the authors have put a lot of effort into the text and figures in the manuscript; however, because of my significant concerns over the scope of the manuscript, its connection to the modelling work, and the key takeaways, I suggest that major revisions (or perhaps a resubmission) are needed before the manuscript can be considered for publication in The- Cryosphere.*

We thank Anonymous Referee #1 for their detailed comments on the modelling section of the manuscript, which are echoed by those of Anonymous Referee #2. The authorship team have discussed this at length and appreciate the limitations of the modelling approach highlighted here and in the general comments. The logic in the flow of our original manuscript from observation to modelling lies in the following. As explained further under the 'Title' section below, our observations revealed only minor changes of structural changes in the Shackleton Ice Shelf System over the past 60 years. Our inference is therefore one of a rather stable system (not withstanding previous authors' observations of some dynamic variability) and indeed, this is confirmed by our BISCLES modelling experiments in that even upper limit conditions, i.e. complete loss of all floating ice that may buttress the grounded ice, lead to only minor simulate change relative to those elsewhere in Antarctica.

Nonetheless, given the matching comments by both anonymous referees we feel that – subject to advice by the journal editor, the best approach may be for us to separate the observations from the modelling, with the latter forming the focus of a subsequent manuscript. As suggested by the anonymous referee(s) we will therefore remove the modelling aspects from this manuscript and focus on the observations that already make up the majority of the current manuscript. The modelling proportion will then form the basis of a subsequent manuscript as suggested.

**General Comments:**

**Title:** *I don't believe the title accurately describes the presented work. I am unsure what the authors mean by "glaciological setting", I think wording that describes the analysis would be better suited to use in the title. I also think it is a bit misleading to claim that the authors are studying the entire Queen Mary and Knox coasts, when only the Shackleton Ice Shelf system is analyzed. Lastly, I am not sure what the authors mean by "implied dynamic stability". Is this stability over the entire 60-year observational period (which would be inaccurate because grounding line retreat (~5 km, Barancato et al., 2020), floating and grounded ice accelerations (Miles et al., 2021; Rignot et al., 2019), and accelerated ice discharge (Rignot et al. 2019) have been observed over this timeframe), over the 400-year modeling period (which, as stated above, I do not think the authors can claim based on the results presented), or between 2018-2021 (following the ice velocity results)? It is difficult to suggest a new title right now because significant changes to the scope of the manuscript need to be made.*

The inference of an 'implied stability' is based on our observations of little dynamic change over the past 60 years, as well as reduced modelled sensitivity to even extreme events such as complete removal of all floating ice. It is natural for systems such as the Shackleton Ice Shelf to show some dynamic variability although, as explained in the manuscript, our observations and modelling cannot confirm that the variabilities observed by previous work necessarily herald major future change. We recognise that the same time that field data are too sparse to make this conclusion with confidence and am therefore recommending that focused programs of field observation are urgently needed. Either way, we will reflect on the current title and revise it appropriately.

**Ice sheet model validation**: *As the manuscript presents one of the first regional ice sheet models to make future projections of the Shackleton Ice Shelf system, it is critical that it is properly described and validated. It is not enough to only show the mismatch of observed and modeled surface ice velocity in order to validate your ice sheet model, one also needs to know how the modeled and observed grounding line positions and ice discharge values compare. In the initial model solution,*

*there are extensive grounded regions along Denman's ice tongue and floating pockets along Denman's grounded ice stream (figure 11) that are not seen in observations (Barancato et al. 2021; Morlighem et al., 2020). Such errors in the initialization of the model can propagate to the transient solutions, so it is critical to have a well-calibrated model that matches present day observations. The ice sheet model description section (L130-L177) is lacking details that would be needed to ensure that this modelling work reproducible. Some examples of missing methodological descriptions are as follows: which 2D stress balance approximation is used (e.g. SIA, SSA, a combination of both), do the ice stiffness and basal friction coefficient change in time, how is the grounding line tracked (e.g. sub-element parameterization), how is basal melt applied numerically to partially floating elements if using a sub-element grounding line parameterization (e.g. to the entire element, to only the floating part of the element), etc. These details are very important and should be included (perhaps it would be better to give a complete model description as a supplement or appendix). For examples of the types of information needed, one could refer to the ISMIP6 Antarctica publication (Seroussi et al. 2020) or this recent manuscript submitted to The-Cryosphere Discussions (Castleman et al. 2021).*

Agree and we will make this the focus of a separate manuscript.

***Units of speed***: *When referencing speeds of both ice and rift/ice front propagation in the main text and figures, the authors switch between m/day and m/year (see lines 258 and line 270 for examples of each). For ice speed, the convention is m/year, so I think it would be best to abide by this convention so that your results can be easily compared to other values in the literature (change in both the text and in figures). Also, when referencing the unit "year", please stick to either "year" or "a", as both were used in the manuscript.*

May we clarify that m/year is commonly used when the temporal resolution of available data is greater than one year. For this reason, when describing the movement of structural features and ice frontal positions we have used the unit m/year to describe the longer-term trends as the measurements are based on data with annual or multi-annual temporal frequency. When describing the ice speed data from feature tracking, we have deliberately used the unit m/day because we are using much higher temporal resolution data. While we are happy to include annual trends for the data, ice speed does vary on much shorter timescales and by simply changing the ice speed data to m/year we would lose this valuable information.

***Data availability statement***: *Please add a data availability statement at the end of the manuscript, as to abide by The-Cryosphere's data policy. All of the links in sections 2.1 and 2.2 should be moved to the data availability statement. In addition, a link to the BISICLES ice flow model, as well as links to all datasets used in the simulation, should be added to this statement if the modeling portion is to remain in the manuscript.*

We are happy to make this change.

***Grammar***: *When reading through the manuscript, I noticed a fair amount of spelling and grammar mistakes (especially missing commas, which would help the readability of the text). I tried to point them out as I found them in the specific comments, but it is possible I missed a few!*

The whole authorship team will all check the manuscript thoroughly for spelling and grammatical errors.

***Specific Comments***:
*L13-L38: In general, I think the abstract is a bit long and should be condensed. Below, I suggested a few sentences that can be removed and/or shortened.*

*L15: change to ``. . . on understanding the controls driving Denman Glacier's dynamic evolution, although . . .''*

We are happy to make these the changes.

*L17: Shackleton Ice Shelf (use capitalization because it is a proper name)*

We will make the change throughout the manuscript.

*L22-L23: Remove "in response to coupled ocean and atmospheric forcing". Coupled forcing suggests that your ice sheet model is coupled to an atmosphere and/or ocean model, which it is not.*

This section will be removed from the manuscript as outlined in the response to the general comments above.

*L31: I make note of this later in the results section, but the authors should not use real years to describe the output of their modeling work because it is not a realistic simulation. Instead of saying "in the third century from now", it would be better to say "in approximately 300 years into the model simulation."*

This section will be removed from the manuscript as outlined in the response to the general comments above.

*L31: Please check the computation of the 6 cm of sea level rise, I computed 40 Tt = 40000 Gt / 3600 [Gt/cm] = 11.11 cm sea level rise equivalent ice mass, but it is possible that my math is off! Is this the sea level contribution from just Denman Glacier, or from the entire model domain? I believe this is from the whole domain; however, in the previous sentence, you discuss the grounding line of Denman Glacier, so it is a bit confusing. Please specify.*

This section will be removed from the manuscript as outlined above.

*L32: I would hesitate to say that 6 cm of global sea level rise equivalent ice volume loss is "small" in comparison to other areas of East Antarctica. First, I don't believe there are any published studies that have run regional transient simulation of the EAIS through 2400, so we cannot compare. Also, 6 cm is on the upper limit of the ISMIP6 projected contribution of the entire Antarctic Ice Sheet to global sea levels by 2100, so this contribution from a single EAIS glacier by 2400 must be fairly significant.*

This section will be removed from the manuscript as outlined above.

*L32-L34: The sentence "it is clear . . . Shackleton system" can be removed.*

We are happy to remove the sentence.

*L34: Here you conclude that there is potential vulnerability of the system to accelerating retreat, but further along in the manuscript (L313), you say that the modeled domain is relatively stable and insensitive to reasonable forcing in the next 400 years. These statements conflict and left me confused about the message of the manuscript. Perhaps it would be more consistent with the rest of the manuscript to say that these data are needed to improve model initialization and validation.*

We agree and will alter the manuscript to reflect the main point that we don't know enough about the system to be able to accurately model potential vulnerabilities.

*L34: Insert comma after "accelerating retreat".*

Happy to make the change.

*L41-L44: These first two sentences can be combined and condensed, which I think would be a bit easier on the reader. Perhaps something like: "It has long been perceived that the East Antarctic Ice Sheet is the stable sector of Antarctica (citations); however, it has now emerged that the Aurora and Wilkes subglacial basins of the EAIS have been contributing to sea level rise since at least the 1980s, with Aurora contributing 1.9 mm and Wilkes contributing 0.6 mm (citations)."*

We agree that the change increases the clarity and are happy to make the change.

*L45: Insert comma after "WAIS"*

Happy to make the change.

*L45 and L47: You are referencing both BedMachine Antarctica (Morlighem et al., 2020) and Bedmap2 (Fretwell et al., 2013) for your values of sea level potential. As BedMachine is the most up-to-date dataset, I would stick to just using the BedMachine citation throughout the manuscript (unless of course you are using the BedMap2 dataset in the paper).*

Bedmap2 was used in the first draft of the manuscript in the current version we updated to BedMachine. The inclusion was an oversight, and we will remove the Fretwell reference.

*L52: Change "it is supplied by . . ." to "Major outlet glaciers drain into this ice shelf system, including Denman, Scott, Northcliffe, Roscoe, and Apfel Glaciers."*

Happy to make the change.

*L57: Cite Morlighem et al. (2020) instead of Rignot et al. (2019), as the BedMachine publication lists the most updated inventories of glacial ice volume.*

Happy to make the change.

*L65: Change "just above" to "just upstream of"*

Happy to make the change.

*L73: Adusumilli et al. (2020) show melt rates peaking at approximately 120 m/yr along Denman Glacier's deep grounding zone. Please check the value reported in your manuscript (6 m/yr), I think this might be a typo.*

The value > 6 m/yr was identified from Figure 1 in Adusumilli et al. (2020) as there is no mention of Denman Glacier in the manuscript or supplementary materials. Shackleton Ice Shelf is listed in Table 1 of the supplementary materials with a basal melt rate of 1.8 ± 1.9 m/yr for 1994-2018. Since the manuscript was submitted we have identified an additional reference, Liang et al. (2021), who estimate melt rates exceed 50 m/yr near the Denman Glacier grounding line for 2010-2018 and we will update the manuscript to include this reference.
Liang, Q., Zhou, C., and Zheng, L.: Mapping Basal Melt Under the Shackleton Ice Shelf, East Antarctica, From CryoSat-2 Radar Altimetry, IEEE Journal of Selected

Topics in Applied Earth Observations and Remote Sensing, 14, 5091-5099, 10.1109/JSTARS.2021.3077359, 2021.

*L82-L85: Please remove "A satisfactory explanation . . . with the nearby Totten Glacier." I think this interrupts the flow of the introduction and does not serve the rest of the paper, as the focus is not to determine where the high melt rates are being forced from.*

Happy to make the change.

*L92: What does it mean to put previously observed dynamic changes in the Shackleton system into the wider regional context of the Queen Mary and Knox coasts? The observational and modelling components of this study do not investigate changes beyond the Shackleton Ice Shelf System, so I think that claiming to frame the regional context of the entire Queen Mary and Knox coasts is a bit misleading. As stated above, I think the really exciting science presented here is the extension of the observational record to other sectors of the Shackleton Ice Shelf and to 2021. So I think this sentence should reflect that.*

We agree and are happy to make the change.

*L95: I do not think we are testing the sensitivity of the domain, as this would require further model runs (such as a control simulation and variance of the ocean forcing).*

We remove this sentence from the manuscript as the modelling will be the focus of a separate manuscript as discussed in detail above.

*L96: Remove "in response to coupled ocean and atmospheric forcing"*

We are happy to remove this sentence from the manuscript.

*L101-L120: Remove links in the main manuscript and add them into a proper data availability statement at the end of the manuscript (see general comments).*

We are happy to this change to the manuscript.

*L107: The "th" on 10th should be a super-script.*

Happy to make the change.

*L111: change ";" to ":"*

Happy to make the change.

*L100 and L112: I think the sentences would read better if you did not use the parentheses at the end of the sentence. For instance, L100 would read as: " . . . using standard GIS techniques following the methodology of Glasser et al. (2009)." The multiple sets of parentheses is confusing for the reader.*

Happy to make the change.

*L120: Insert comma between "methods" and "feature tracking"*

Happy to make the change.

*L121: "We use image . . ." Use present tense*

Happy to make the change.

*L124: " . . . and the quality of the velocity map is maximized"*

Happy to make the change.

*L126: Change "allowed" to "allows"*

Happy to make the change.

*L146: Change "horizontal rate of strain tensor" to "horizontal strain rate tensor"*
*L151: Change "rate-strain" to "strain-rate"*
*L163: Please give references to previous BISICLES studies that have used this initialization method.*
*L173: Change to "The single future simulation follows the methodology of Matin et al. (2019)."*
*L174: "Under" should be lowercase*
*L174: I had assumed that the melt rate was 1000 m/yr across all floating ice, but here you say that the melt rate reaches 1000 m/yr in places. How is the basal melt rate computed? Does the melt rate vary in space and/or with the geometry of the ice shelf?*

These sections will all be removed from the manuscript as outlined above.

*L195: These rift-systems along the Shackleton Ice Shelf are really fascinating! It is so interesting that the two rift-systems have almost identical shapes (with system-1 being larger than that of system-2).*

Yes indeed, we completely agree.

*L213: Cite figure 1b here, it shows the high concentration of surface features on the Denman Ice Tongue very well.*

Happy to make the change.

*L218: Should this first sentence be citing figure 5 (figure 6 shows the rift on the Shackleton-Roscoe shear margin)?*

Yes, this first sentence should reference figure 5, we thank Anonymous Referee #1 for noticing the mistake and will correct it and check all figure references thoroughly to make sure they are correct.

*L222: I am having trouble figuring out which rift you are describing in this line (the one on the western side of Scott Glacier). Since you are highlighting this particular rift, it would be helpful to highlight it or point it out in figure 5c if possible (perhaps an arrow or pointer next to it so that it is easily identifiable by the reader).*

We agree that it is difficult to distinguish between the different rifts discussed in this paragraph and will ensure that they are all clearly labelled in figure 5c to allow distinction.

*L229: Replace "some changes" with wording that is a bit more definitive.*

The whole sentence currently reads '*Across the whole system there are some changes in the shear margins between the various inlet glaciers and the main body of 230 the Shackleton Ice Shelf.*' We will change the sentence to read 'We observe changes of different magnitude in all of the shear margins of the Shackleton Ice Shelf System over the period of observation.'

*L244: Why did you decide to compute the velocity difference between one year (2019- 2020)? It seems like this would not give very interesting results because that is not enough time to for the system to respond to a forcing perturbation (aside from the northern point of Scott Glacier's floating extension, which looks like perhaps it is undergoing a calving event).*

We report the difference in ice speed between 2019-2020 because in this paragraph we are focusing on the most recent changes that have not previously been reported. We think Anonymous Referee #1 is referring to a perturbation to the flow of ice by, e.g. atmospheric or ocean forcing, or a calving event which are unlikely to show much change on the timescale of one year. We acknowledge that this is different in Greenland where there are strong year on year and even seasonal changes in response to transient ocean warming/cooling. Figure 9 provides the recent higher resolution data in the context of longer-term changes, and we will amend the text to clarify that we are only looking at the most recent changes in this paragraph. Recent changes are discussed in the context of longer-term ice speed variability (shown in Figure 9) in the subsequent paragraph (L249-260).

*L254: Change m/day to m day^{-1} to be consistent with the rest of the paper (ultimately should be m year^{-1}, see general comments)*

We apologise for the inconsistency in unit form and will check the manuscript carefully to maintain m day$^{-1}$ and m year$^{-1}$.

*L255: "Speeds ~ 10 km either side of the grounding line . . ." confuses me a bit. This sentence makes it sound like you are talking about grounded ice as well (since 10 km on either side of the grounding line would extend 10 km upstream into grounded ice), but I believe you are talking about points 10 and 11 in figure 8a. Perhaps it would be better to say "Speeds up to __ km downstream of the grounding line show . . ." . I also think it would be helpful to reference the specific points in figure 8a that you are discussing (e.g. in L258, "close to the ice front (point 13 in fig. 8a)").*
*L265: The format of ((b) in Fig. 10)(Furst et al., 2015) is a bit crowded. Instead of using double parentheses, change to (label-b in Fig. 10, Furst et al., 2015). Same with L266 and L267.*

We agree that the statement is confusing as point number 10 is approximately located on the grounding line (as identified from Measures data). We will change the sentence to read 'There is no observable change in speed >10 km downstream of the grounding line of Scott Glacier (points 10 and 11 in Fig. 8a).'

*L267: Change "rise in the ocean floor" to "local topographic high in ocean bathymetry".*

Happy to make the change.

*L269-L283: When describing the model results, I would stray away from using actual years (e.g. ". . . of Denman Glacier occurs after 2150 . . ."), as you are modeling with unrealistic forcing. Instead, I would change this to something like ". . . of Denman Glacier occurs 150 years into the model simulation . . .".*

*L276: The dynamic response of the system seems pretty significant, as Denman Glacier retreats more than 100 km upstream and Denman and Scott Glaciers end up connecting around a topographic high.*

This section will be removed from the manuscript as outlined above.

*L285: This first sentence of the discussion section contradicts existing literature (e.g. Barancato et al. 2020; Miles et al. 2021), which have cited patterns of grounding line retreat and ice velocity change that appear to be ocean induced since the 1970s or so. This study only looked at changes in surface features through the 60 year observational period and velocity changes over the past ~15 years; however, it seems that changes outside of those presented in this paper are occurring over those timescales. As such, I don't think the authors can claim, based on the presented manuscript, that the Queen Mary and Knox coasts have not changed significantly in the last 60 years.*

Our own reconstructions of longer-term ice flow velocity change (Luckman, unpublished data) very much match those reported previously, e.g. in Miles et al. (2021). We refrained from repeating these matching observations here but perhaps should provide them as a wider framework to place our more recent inferences of (no substantial) ice flow velocity within. Pre-2000 data points are sparse and as we emphasize later in this paragraph (lines 291-292): "*An increase in ice flow speed was observed just upstream of the Denman Glacier groundling line between 1972-4 and 1989 and, to a lesser extent, through to 2008 (Miles et al., 2021, their Fig. 3c). Variability in ice flow* speed then became insignificant through to 2016-17 (Miles et al., 2021, their Fig. 3c), a pattern that has continued since (Fig. 8b, 9b) and accordingly we cannot identify any related change in the structure of the system (Fig. 3)". Whilst an increase in ice flow speed of the Denman Glacier clearly occurred between the 1970s and 2017 detailed examination of the timing of this change (as matched by our own feature-tracking based inferences) reveals that almost all of it happened sometime between 1972 and 1989, with very little change since. We will review this aspect of the manuscript and make appropriate revisions, including additional (own) ice flow velocity data as maybe required.

*L291: change "groundling" to "grounding"*

Thank you pointing the mistake out, happy to make the change.

*L310-315: I don't think the authors can make this claim based on a single transient model run of the Denman/Shackleton system. The Denman Glacier grounding line is currently retreating under present day forcing conditions (Barancato et al. 2020) and this retreat could be susceptible to the marine ice sheet instability, as the bed upstream of the current grounding line position is retrograde. Your model results show > 100 km of grounding line retreat by 2310 over Denman Glacier, which is significant. However, you did not test the response of the system to realistic forcing over the same timeframe, so we cannot make a statement on the sensitivity of the system. Lastly, the Aurora and Wilkes subglacial basins are not included in the model domain, so this last statement is a speculative conclusion rather than one based on your modeling results and should not be included in the manuscript.*
*L317-L319: It is a great addition to include model limitations in the discussion section; however, unless you are running a thermal model, I do not believe that the geothermal heat flux is used by the ice sheet model. In addition, the ocean conditions and bathymetry will not impact your model run because you assume near-instantaneous disintegration of the floating ice shelf. In the modeling results that you presented, I would expect the results to be primarily impacted by mesh resolution (I am assuming you are not using adaptive mesh refinement, so as the grounding line retreats*

*upstream, the size of the elements will most likely become larger), poorly constrained basal friction and ice stiffness parameters that do not change in time, use of a 2D stress balance approximation instead of a higher order model, etc. It would be helpful to the reader to know exactly how the limitations you listed impact your model (e.g. poorly constrained basal hydrology leads to a poorly constrained basal friction parameter, ice properties impact the ice stiffness parameter, etc.).*

These sections will be removed from the manuscript as outlined above.

L321-L337: This paragraph lost me a bit. I understand the comparison to Totten Glacier, but I do not think it is appropriate to dive into such a detailed discussion of the subglacial conditions of Queen Mary Land because it does not connect to the rest of the paper. If the authors want to speculate on the subglacial conditions, they need to tie it back to the conclusions of the paper (i.e. its impact on enhanced ice shelf basal melting rates near the grounding line, reducing basal friction at the ice-bed interface, etc.). Without that obvious connection to tie back to the rest of the paper, this paragraph seems out of place and left me confused.

Both Anonymous Referees were left confused by this paragraph and it is clear that we need to much better clarify its logic connection with our observations reported earlier. Our discussions as a whole are intended to place our observations within the wider framework of the governing atmospheric, oceanographic and subglacial (incl. solid earth) settings, setting the scene for possible future investigations of key properties and processes that we do not understand well at present (of which there are many). We will do this by comprehensively revising the paragraph and the parts of the manuscript that it connects to.

*L340: See previous comment about L285.*

Please see our reply above.

---

## Author Response (AR1)

Response to Anonymous Referee #1

We thank Anonymous Referee #1 for their considered and detailed review of our manuscript. Below we respond to each comment, with the anonymous referee's original comments shown in italic text and our response in blue text.

*Summary: The manuscript by Dr. S. Thompson and colleagues presents a two-pronged study aimed at (1) reconstructing the 60-year surface feature record and 2014-2021 surface ice velocity field of the Shackleton Ice Shelf System via analysis of satellite imagery, and (2) modeling the future response of the upstream grounded glaciers feeding the Shackleton Ice Shelf System to near-instantaneous disintegration of this ice shelf via a 400-year projection made using the BISICLES ice sheet model. The authors conclude that the Shackleton Ice Shelf System has not changed significantly over the 60-year observational period (aside from a localized acceleration in surface ice velocity near the ice front of Scott Glacier) and that the future upstream glacier response to collapse of the Shackleton Ice Shelf is minimal relative to changes projected across other East Antarctic basins.*

*I find the extension of the surface-observational record both spatially (to the neighboring Scott Glacier, Roscoe Glacier, and greater Shackleton Ice Shelf region) and temporally (from 2017- 2021) to be the primary strength of this manuscript, as this information is very useful to the ice sheet modeling community. However, I have significant concerns regarding the scope of the paper and the applicability of the numerical modeling work, which make the manuscript difficult to follow. First, I find significant overlap in the analysis of surface features of the Denman Ice Tongue between this manuscript and that of Miles et al. (2021) (e.g. ice front positions, patterns of rifts, and location of pinning points on the Denman Ice tongue, as well as the calving of the large tabular icebergs in the 1940's and in 1984). While the manuscript does properly cite Miles et al. (2021), the repetition of the analyses and findings makes up a significant portion of this study and thus reduces the novelty of the manuscript. The manuscript should be reorganized to have a greater focus on the spatial and temporal extension of the surface observation record.*

We have restructured the manuscript to clarify the novelty of the extension in spatial and temporal observations and included additional ice penetrating radar data. Our aim was to provide full context to the observations we report and discuss, rather than simply citing previous work. We have made the distinction as requested; specific changes are detailed in response to the comments below.

*In addition, the numerical modeling portion of this manuscript is rather disconnected to the scope and findings of the rest of the paper and is not robust enough to support the authors' conclusions. The first half of the manuscript analyzes short-term and fine-scale changes of features on the floating portion of the Shackleton Ice Shelf System; however, the authors then model the 400-year response of the grounded regions of mainly Denman Glacier to near- instantaneous disintegration of this ice shelf. This disconnect between the focus on observing small-scale ice shelf features across the entire Shackleton Ice Shelf System and modeling grounding line retreat and volume loss of Denman Glacier (without mentioning of the response Scott and Roscoe Glacier) makes following the progression of the manuscript very difficult. If modeling is going to be included in this study, it needs to complement the rest of the manuscript (i.e. model how future ice flow responds to changes in the surface features discussed in the first half of the manuscript). Furthermore, from the analysis of a single model simulation, the authors imply that the Queen Mary and Knox coasts are relatively stable and insensitive to reasonable forcing in the next 400 years (see L313, I am also assuming this is the "implied dynamic stability" referenced in the title). I don't believe the authors can claim stability of the system and make a statement about sensitivity without modeling the system's response to realistic forcing perturbations. Overall, I believe the modeling portion of this manuscript needs to be either redone so that it supports the observational-focus of the paper or separated and made the focus of a secondary manuscript.*

*It is apparent that the authors have put a lot of effort into the text and figures in the manuscript; however, because of my significant concerns over the scope of the manuscript, its connection to the modelling work, and the key takeaways, I suggest that major revisions (or perhaps a resubmission) are needed before the manuscript can be considered for publication in The- Cryosphere.*

We thank Anonymous Referee #1 for their detailed comments on the modelling section of the manuscript, which are echoed by those of Anonymous Referee #2. The authorship team have discussed this at length and appreciate the limitations of the modelling approach highlighted here and in the general comments. The logic in the flow of our original manuscript from observation to modelling lies in the following. As explained further under the 'Title' section below, our observations revealed only minor changes of structural changes in the Shackleton Ice Shelf System over the past 60 years. Our inference is therefore one of a rather stable system (not withstanding previous authors' observations of some dynamic variability) and indeed, this is confirmed by our BISCLES modelling experiments in that even upper limit conditions, i.e. complete loss of all floating ice that may buttress the grounded ice, lead to only minor simulate change relative to those elsewhere in Antarctica.

Nonetheless, given the matching comments by both anonymous referees we feel that – subject to advice by the journal editor, the best approach may be for us to separate the observations from the modelling, with the latter forming the focus of a subsequent manuscript. As suggested by the anonymous referee(s) we have removed the modelling aspects from this manuscript and focused on the observations that already make up the majority of the current manuscript. The modelling proportion will then form the basis of a subsequent manuscript as suggested.

*General Comments:*

*Title: I don't believe the title accurately describes the presented work. I am unsure what the authors mean by "glaciological setting", I think wording that describes the analysis would be better suited to use in the title. I also think it is a bit misleading to claim that the authors are studying the entire Queen Mary and Knox coasts, when only the Shackleton Ice Shelf system is analyzed. Lastly, I am not sure what the authors mean by "implied dynamic stability". Is this stability over the entire 60-year observational period (which would be inaccurate because grounding line retreat (~5 km, Barancato et al., 2020), floating and grounded ice accelerations (Miles et al., 2021; Rignot et al., 2019), and accelerated ice discharge (Rignot et al. 2019) have been observed over this timeframe), over the 400-year modeling period (which, as stated above, I do not think the authors can claim based on the results presented), or between 2018-2021 (following the ice velocity results)? It is difficult to suggest a new title right now because significant changes to the scope of the manuscript need to be made.*

The inference of an 'implied stability' is based on our observations of little dynamic change over the past 60 years, as well as reduced modelled sensitivity to even extreme events such as complete removal of all floating ice. It is natural for systems such as the Shackleton Ice Shelf to show some dynamic variability although, as explained in the manuscript, our observations and modelling cannot confirm that the variabilities observed by previous work necessarily herald major future change. We recognise that the same time that field data are too sparse to make this conclusion with confidence and am therefore recommending that focused programs of field observation are urgently needed. In response this comment and the removal of the

modelling from the manuscript, we have changed the title of the manuscript which now reads

L1-2: Glaciological setting and structural evolution of the Shackleton Ice Shelf System, East Antarctica, over the past 60 years

*Ice sheet model validation: As the manuscript presents one of the first regional ice sheet models to make future projections of the Shackleton Ice Shelf system, it is critical that it is properly described and validated. It is not enough to only show the mismatch of observed and modeled surface ice velocity in order to validate your ice sheet model, one also needs to know how the modeled and observed grounding line positions and ice discharge values compare. In the initial model solution, there are extensive grounded regions along Denman's ice tongue and floating pockets along Denman's grounded ice stream (figure 11) that are not seen in observations (Barancato et al. 2021; Morlighem et al., 2020). Such errors in the initialization of the model can propagate to the transient solutions, so it is critical to have a well-calibrated model that matches present day observations. The ice sheet model description section (L130-L177) is lacking details that would be needed to ensure that this modelling work reproducible. Some examples of missing methodological descriptions are as follows: which 2D stress balance approximation is used (e.g. SIA, SSA, a combination of both), do the ice stiffness and basal friction coefficient change in time, how is the grounding line tracked (e.g. sub-element parameterization), how is basal melt applied numerically to partially floating elements if using a sub-element grounding line parameterization (e.g. to the entire element, to only the floating part of the element), etc. These details are very important and should be included (perhaps it would be better to give a complete model description as a supplement or appendix). For examples of the types of information needed, one could refer to the ISMIP6 Antarctica publication (Seroussi et al. 2020) or this recent manuscript submitted to The-Cryosphere Discussions (Castleman et al. 2021).*

Agree and we will make this the focus of a separate manuscript.

*Units of speed: When referencing speeds of both ice and rift/ice front propagation in the main text and figures, the authors switch between m/day and m/year (see lines 258 and line 270 for examples of each). For ice speed, the convention is m/year, so I think it would be best to abide by this convention so that your results can be easily compared to other values in the literature (change in both the text and in figures). Also, when referencing the unit "year", please stick to either "year" or "a", as both were used in the manuscript.*

May we clarify that m/year is commonly used when the temporal resolution of available data is greater than one year. For this reason, when describing the movement of structural features and ice frontal positions we have used the unit m/year to describe the longer-term trends as the measurements are based on data with annual or multi-annual temporal frequency. When describing the ice speed data from feature tracking, we have deliberately used the unit m/day because we are using much higher temporal resolution data. While we are happy to include annual trends for the data, ice speed does vary on much shorter timescales and by simply changing the ice speed data to m/year we would lose this valuable information.

*Data availability statement: Please add a data availability statement at the end of the manuscript, as to abide by The-Cryosphere's data policy. All of the links in sections 2.1 and 2.2 should be moved to the data availability statement. In addition, a link to the BISICLES ice flow model, as well as links to all datasets used in the simulation, should be added to this statement if the modeling portion is to remain in the manuscript.*

We are happy to make this change. All of the links in sections 2.1 and 2.2 have been removed and included in a data availability statement. Section 2.1, 2.2 and the data availability statement now read

L 91-120:
**2.1 Structure and feature mapping from optical and SAR imagery**
Surface structures and features of the Queen Mary and Knox coasts were mapped from satellite imagery using standard GIS techniques (following Glasser et al. (2009)). Structural features have been mapped every 6 months from February 2015 to February 2022 using freely available datasets from Landsat 8 OLI and Sentinel 2A and 2B, where cloud cover is <15%, in combination with Sentinel 1A and 1B GRD to improve spatial and temporal coverage. To include multi-decadal changes in the extent and structure of the whole system, we used several different datasets from three time periods, only choosing the datasets that covered the entire area of interest. These include the declassified ARGON KH-5 images acquired 16th May 1962, Landsat 1 MMS acquired 27th February 1974, Landsat 5 TM acquired between 10th and 12th February 1991 and the MODIS Mosaic of Antarctica (MOA) image map, a composite of 259 swaths of both Aqua and Terra MODIS images acquired between 01 Nov 2008 and 28 Feb 2009 (Scambos et al., 2007). Datasets were registered to the Sentinel-2 imagery as required.
Mapped features included, where visible: the ice-shelf or floating-glacier edges, rifts, crevasses and crevasse traces and longitudinal surface features following the methodology of Glasser et al. (2009). Interpretation of the optical imagery was performed using multiple band combinations to provide natural colour (Landsat-1 MSS bands 7-4-3, Landsat-5 TM bands 5-2-1 and Sentinel-2 bands 4-3-2) and for all imagery standard enhancement procedures (contrast stretching and histogram equalisation) were used to improve the contrast across features. The spatial resolution of the data sets varies from 10 m to 150 m and is thus a limitation on the minimum size and accuracy of the features mapped in each data set.

**2.2 Feature tracking from Sentinel-1**
Glacier surface velocities were derived using feature tracking between pairs of synthetic aperture radar (SAR) images acquired by the Sentinel-1 satellite. Using the standard Gamma software and following commonly adopted methods, feature tracking uses cross-correlation to find the displacement of surface features between pairs of images, which are then converted to velocities using the time delay between those images (Luckman et al., 2007). We use image patch sizes of ~ 1 km in ground range and sample at ~ 100 m in range and azimuth. Where the time-delay between images is sufficiently short, and surface change is minimized (for instance by very cold temperatures), trackable surface features include fine-scale coherent phase patterns (speckle) and the quality of the derived velocity map is maximized. We applied feature tracking to many image pairs and selected the best velocity map in terms of minimum noise and maximum coverage of high-quality matches for each year to provide mean annual velocity maps, including ice flow direction. We then produced percentage difference maps, scaled between +/- 10% but excluding data where the mean velocities are less than 0.2 m/day as uncertainties in velocity magnitude are around 0.2 m day-1 (Benn et al., 2019). This approach allows us to optimize the quality of the surface strain map derived from the surface velocity.

L 332-337:

Data availability
All satellite imagery used in this work are freely available as follows; Landsat 8 OLI, 5 TM and 1 (all downloaded from https://earthexplorer.usgs.gov/), MODIS Mosaic of Antarctica 2008-2009 (downloaded from https://nsidc.org/data/nsidc-0593/versions/2), Sentinel 2 A and B (all downloaded from https://scihub.copernicus.eu/dhus/#/home)
Sentinel 1A and B GRD (all downloaded from https://scihub.copernicus.eu/dhus/#/home) and ARGON KH-5 (downloaded from https://earthexplorer.usgs.gov/).

*Grammar: When reading through the manuscript, I noticed a fair amount of spelling and grammar mistakes (especially missing commas, which would help the readability of the text). I tried to point them out as I found them in the specific comments, but it is possible I missed a few!*

The whole authorship team have thoroughly checked the manuscript thoroughly for spelling and grammatical errors.

*Specific Comments:*
*L13-L38: In general, I think the abstract is a bit long and should be condensed. Below, I suggested a few sentences that can be removed and/or shortened.*
*L15: change to ``. . . on understanding the controls driving Denman Glacier's dynamic evolution, although . . .''*

We are happy to make these the changes. The sentence now reads

L19-21: Recent attention has therefore been focusing on understanding the controls driving Denman Glacier's dynamic evolution, although knowledge of the wider regional context and timescales over which the future responses may occur remains incomplete.

*L17: Shackleton Ice Shelf (use capitalization because it is a proper name)*

We have made the change throughout the manuscript.

*L22-L23: Remove "in response to coupled ocean and atmospheric forcing". Coupled forcing suggests that your ice sheet model is coupled to an atmosphere and/or ocean model, which it is not.*

This section has been removed from the manuscript as outlined in the response to the general comments above.

*L31: I make note of this later in the results section, but the authors should not use real years to describe the output of their modeling work because it is not a realistic simulation. Instead of saying "in the third century from now", it would be better to say "in approximately 300 years into the model simulation."*

This section has been removed from the manuscript as outlined in the response to the general comments above.

*L31: Please check the computation of the 6 cm of sea level rise, I computed 40 Tt = 40000 Gt / 3600 [Gt/cm] = 11.11 cm sea level rise equivalent ice mass, but it is possible that my math is off! Is this the sea level contribution from just Denman Glacier, or from the entire model domain? I believe this is from the whole domain; however, in the previous sentence, you discuss the grounding line of Denman Glacier, so it is a bit confusing. Please specify.*

This section has been removed from the manuscript as outlined above.

*L32: I would hesitate to say that 6 cm of global sea level rise equivalent ice volume loss is "small" in comparison to other areas of East Antarctica. First, I don't believe there are any published studies that have run regional transient simulation of the EAIS through 2400, so we cannot compare. Also, 6 cm is on the upper limit of the ISMIP6 projected contribution of the entire Antarctic Ice Sheet to global sea levels by 2100, so this contribution from a single EAIS glacier by 2400 must be fairly significant.*

This section has been removed from the manuscript as outlined above.

*L32-L34: The sentence "it is clear . . . Shackleton system" can be removed.*

We have removed the sentence from the manuscript.

*L34: Here you conclude that there is potential vulnerability of the system to accelerating retreat, but further along in the manuscript (L313), you say that the modeled domain is relatively stable and insensitive to reasonable forcing in the next 400 years. These statements conflict and left me confused about the message of the manuscript. Perhaps it would be more consistent with the rest of the manuscript to say that these data are needed to improve model initialization and validation.*

We agree and have altered the manuscript to reflect the main point that we don't know enough about the system to be able to accurately model potential vulnerabilities. The sentences now read

L33-37: Given the potential vulnerability of the system to accelerating retreat into the overdeepened bedrock trough, better data recording the glaciological, oceanographic, and geological conditions in the Shackleton system are required to improve the certainty of numerical model predictions. With access to these remote coastal regions a major challenge, coordinated internationally collaborative efforts are required to quantify how much the Shackleton region is likely to contribute to sea level rise in the coming centuries.

*L34: Insert comma after "accelerating retreat".*

We have made the change.

*L41-L44: These first two sentences can be combined and condensed, which I think would be a bit easier on the reader. Perhaps something like: "It has long been perceived that the East Antarctic Ice Sheet is the stable sector of Antarctica (citations); however, it has now emerged that the Aurora and Wilkes subglacial basins of the EAIS have been contributing to sea level rise since at least the 1980s, with Aurora contributing 1.9 mm and Wilkes contributing 0.6 mm (citations)."*

We agree that the change increases the clarity, and the sentence now reads

L51-53: The East Antarctic Ice Sheet (EAIS) has historically been perceived as the stable sector of Antarctica (Silvano et al., 2016); however, it has now emerged that the Aurora and Wilkes subglacial basins of the EAIS have been contributing to sea level rise since the 1980s, with Aurora contributing 1.9 mm sea level rise and Wilkes 0.6 mm (Rignot et al., 2019).

*L45: Insert comma after "WAIS"*

We have made the change.

*L45 and L47: You are referencing both BedMachine Antarctica (Morlighem et al., 2020) and Bedmap2 (Fretwell et al., 2013) for your values of sea level potential. As BedMachine is the most up-to-date dataset, I would stick to just using the BedMachine citation throughout the manuscript (unless of course you are using the BedMap2 dataset in the paper).*

Bedmap2 was used in the first draft of the manuscript in the current version we updated to BedMachine. The inclusion was an oversight, and we have removed the Fretwell references and replaced with Morlighem.

*L52: Change "it is supplied by . . ." to "Major outlet glaciers drain into this ice shelf system, including Denman, Scott, Northcliffe, Roscoe, and Apfel Glaciers."*

We have made the change and the sentence now reads

L48-49: The Shackleton system flows from a major drainage basin at the EAIS margin, located at the intersection of the Queen Mary and Knox coasts (Fig. 1a), including major outlet glaciers such as Denman, Scott, Northcliffe, Roscoe and Apfel.

*L57: Cite Morlighem et al. (2020) instead of Rignot et al. (2019), as the BedMachine publication lists the most updated inventories of glacial ice volume.*

We have made the change, the sentence now reads

L53-54 The Denman Glacier alone is estimated to hold an equivalent of 1.5 m of sea level rise equivalent ice mass (Morlighem et al., 2020).

*L65: Change "just above" to "just upstream of"*

We have made the change, the sentence now reads

L60-63 Ice velocity data from the region are sparse before the late 2000s, but recent work identified an increase in ice velocity of the Denman Glacier of 16 % since the 1970s (Rignot et al., 2019), with an increase of 11± 5 % just upstream of the Denman grounding line between 1972-74 and 1989 and a more recent decrease of acceleration rate to 3 ± 2 % between 1989 and 2007-08 (Miles et al., 2021, their Fig. 3c).

*L73: Adusumilli et al. (2020) show melt rates peaking at approximately 120 m/yr along Denman Glacier's deep grounding zone. Please check the value reported in your manuscript (6 m/yr), I think this might be a typo.*

The value > 6 m/yr was identified from Figure 1 in Adusumilli et al. (2020) as there is no mention of Denman Glacier in the manuscript or supplementary materials. Shackleton Ice Shelf is listed in Table 1 of the supplementary materials with a basal melt rate of 1.8 ± 1.9 m/yr for 1994-2018. Since the manuscript was submitted we have identified an additional reference, Liang et al. (2021), who estimate melt rates exceed 50 m/yr near the Denman Glacier grounding line for 2010-2018 and we have update the manuscript to include this reference. The sentence now reads:

L68-72: Meltwater production from basal melt of the Shackleton system (73 Gt year-1) between 2003 and 2008 rivalled that from Thwaites (98 Gt year-1) (Rignot et al., 2013). Satellite derived basal melt rates between 2010 and 2018 revealed high but localised melt rates of > 50 m year-1 close to the Denman grounding line (Liang et al., 2021), on par with basal melt rates in the Bellingshausen and Amundsen Sea (Adusumilli et al., 2020).

L415-416: Liang, Q., Zhou, C., and Zheng, L.: Mapping Basal Melt Under the Shackleton Ice Shelf, East Antarctica, From CryoSat-2 Radar Altimetry, IEEE Journal of Selected Topics in Applied Earth Observations and Remote Sensing, 14, 5091-5099, 10.1109/JSTARS.2021.3077359, 2021.

*L82-L85: Please remove "A satisfactory explanation . . . with the nearby Totten Glacier." I think this interrupts the flow of the introduction and does not serve the rest of the paper, as the focus is not to determine where the high melt rates are being forced from.*

We have removed the sentences from the manuscript.

*L92: What does it mean to put previously observed dynamic changes in the Shackleton system into the wider regional context of the Queen Mary and Knox coasts? The observational and modelling components of this study do not investigate changes beyond the Shackleton Ice Shelf System, so I think that claiming to frame the regional context of the entire Queen Mary and Knox coasts is a bit misleading. As stated above, I think the really exciting science presented here is the extension of the observational record to other sectors of the Shackleton Ice Shelf and to 2021. So I think this sentence should reflect that.*

We agree and the sentence now reads

L86-89: Here we place the previously reported dynamic changes in the Denman Glacier into the wider regional context of the Shackleton Ice Shelf system. We do so by presenting an improved biannual temporal frequency of observations in the last eight years (2014-22) integrating airborne radar data, new satellite observations of ice structure, changes in ice front position and ice-flow velocities, with known geometrical and glaciological constraints.

*L95: I do not think we are testing the sensitivity of the domain, as this would require further model runs (such as a control simulation and variance of the ocean forcing).*

We have removed this sentence from the manuscript as the modelling will be the focus of a separate manuscript, discussed in detail above.

*L96: Remove "in response to coupled ocean and atmospheric forcing"*

We have removed this sentence from the manuscript as the modelling will be the focus of a separate manuscript, discussed in detail above.

*L101-L120: Remove links in the main manuscript and add them into a proper data availability statement at the end of the manuscript (see general comments).*

We have removed the links and added the information to the data availability statement at the end of the manuscript as described above.

*L107: The "th" on 10th should be a super-script.*

We have made the change.

*L111: change ";" to ":"*

We have made the change.

*L100 and L112: I think the sentences would read better if you did not use the parentheses at the end of the sentence. For instance, L100 would read as: " . . . using standard GIS techniques following the methodology of Glasser et al. (2009)." The multiple sets of parentheses is confusing for the reader.*

We have made the change and the sentence now reads

L101-102: Mapped features included, where visible: the ice-shelf or floating-glacier edges, rifts, crevasses and crevasse traces and longitudinal surface features following the methodology of Glasser et al. (2009).

*L120: Insert comma between "methods" and "feature tracking"*

We have made the change.

*L121: "We use image . . ." Use present tense*

We have made the change and the sentence now reads

L112-113: We use image patch sizes of ~ 1 km in ground range and sample at ~ 100 m in range and azimuth.

*L124: " . . . and the quality of the velocity map is maximized"*

I think this is in reference to the spelling of maximised which has been changed. Apologies if we have misinterpreted the comment.

*L126: Change "allowed" to "allows"*

We have made the change.

*L146: Change "horizontal rate of strain tensor" to "horizontal strain rate tensor"*
*L151: Change "rate-strain" to "strain-rate"*
*L163: Please give references to previous BISICLES studies that have used this initialization method.*
*L173: Change to "The single future simulation follows the methodology of Matin et al. (2019)."*
*L174: "Under" should be lowercase*
*L174: I had assumed that the melt rate was 1000 m/yr across all floating ice, but here you say that the melt rate reaches 1000 m/yr in places. How is the basal melt rate computed? Does the melt rate vary in space and/or with the geometry of the ice shelf?*

These sections have all been removed from the manuscript as outlined above.

*L195: These rift-systems along the Shackleton Ice Shelf are really fascinating! It is so interesting that the two rift-systems have almost identical shapes (with system-1 being larger than that of system-2).*

Yes indeed, we completely agree.

*L213: Cite figure 1b here, it shows the high concentration of surface features on the Denman Ice Tongue very well.*

We have added the figure citation to the sentence which now reads

L167-168: The surface of the Denman Glacier is heavily featured with a combination of crevasses, flow lines and channel-like features (Fig. 1b).

*L218: Should this first sentence be citing figure 5 (figure 6 shows the rift on the Shackleton-Roscoe shear margin)?*

Yes, this first sentence should reference figure 5 (now 4), we thank Anonymous Referee #1 for noticing the mistake and we have corrected and checked all figure references thoroughly to make sure they are correct.

*L222: I am having trouble figuring out which rift you are describing in this line (the one on the western side of Scott Glacier). Since you are highlighting this particular rift, it would be helpful to highlight it or point it out in figure 5c if possible (perhaps an arrow or pointer next to it so that it is easily identifiable by the reader).*

We agree that it is difficult to distinguish between the different rifts discussed in this paragraph. All rifts have been clearly labelled in what is now figure 4 to allow distinction and the paragraph now reads

L170-174: The floating portion of Scott Glacier is dominated by a series of rifts striking perpendicular to the flow direction (Fig. 4). The rifts initiate approximately 20 km down glacier of the grounding line and widen to ~ 2.5 km as they flow around the Taylor Islands. Between 2015 and 2022 the up-flow (southern) rift widens at a rate of ~ 200 m year-1 (Fig. 4a – labelled 1), while the down-flow rift narrows at a rate of ~ 100 m year-1 (Fig. 4a – labelled 2). The rift formation, widening and narrowing process is evident from 1962 through to 2009 (Fig. 4b). A rift on the eastern side of Scott Glacier, initiated from the eastern Scott shear margin, has increased in length toward Chugunov Island by ~ 5 km between February 2021 and June 2022 (Fig 4c -

labelled 3).

[Figure]

**Figure 4:** Evolution of the rifts on Scott Glaciers from (a) 2015-2021 mapped from Sentinel 2B acquired 27th February 2021, Landsat 8 OLI acquired 16th February 2019 and 25th March 2015 and from Sentinel 2A acquired 23rd February 2017  (Background: Sentinel 2B acquired 27th February 2021) and (b) 1962-2009 mapped from ARGON KH5 acquired 16th May 1962, Landsat 1 MMS acquired 27th February 1974 and Landsat 5 TM acquired 10th -12th February 1991 and from MODIS MOA acquired 1st November – 28th February 2009 – Modis MOA (Scambos et al., 2007) (Background: MODIS MOA). (c) The floating portion of Scott Glacier in June 2022, highlighting the iceberg that calved from the western front in April 2022 and the rifting across the eastern portion of the front towards Chugunov Island (Background: Sentinel 1A acquired 6th June 2022).

*L229: Replace "some changes" with wording that is a bit more definitive.*

We have changed the sentence to read

L181-182: Across the whole system, small-scale changes have been observed in the shear margins separating the various inlet glaciers and along the main body of the Shackleton Ice Shelf.

*L244: Why did you decide to compute the velocity difference between one year (2019- 2020)? It seems like this would not give very interesting results because that is not enough time to for the system to respond to a forcing perturbation (aside from the northern point of Scott Glacier's floating extension, which looks like perhaps it is undergoing a calving event).*

We report the difference in ice speed between 2019-2020 because in this paragraph we are focusing on the most recent changes that have not previously been reported. We think Anonymous Referee #1 is referring to a perturbation to the flow of ice by, e.g. atmospheric or ocean forcing, or a calving event which are unlikely to show much change on the timescale of one year. We acknowledge that this is different in Greenland where there are strong year on year and even seasonal changes in response to transient ocean warming/cooling. We have now changed the difference maps to percentage difference maps, scaled to +/- 10 % as this provided the clearest visualisation of change and included a new Figure 7b showing the longest difference period we can produce 2021-2018 (this is as longer period as the data from Sentinel-1 allows), prior to 2018 the data is much nosier and the resulting difference maps are very messy. We have added another figure, new Figure 8, to show the annual difference maps 2021-2020, 2020-2019, 2019-2018 and 2018-2017. Figure 9 provides the recent higher resolution data in the context of longer-term changes, and we have amended the text to clarify that we are only looking at the most recent changes in this paragraph.

[Figure]

Figure 8: (a) Mean speed for 2022 with velocity arrows and (b) percentage difference in mean speed between 2021 and 2018, scaled between +/- 10%, with point locations illustrating the ice speed timeseries in Figure 10.

[Figure]

Figure 9: Percentage difference in mean speed between (a) 2018-17, (b) 2019-18, (c) 2020-19 and (d) 2021-20 scaled between +/- 10%.

L205-221: Mean ice speed derived at annual temporal frequency from Sentinel-1 data varies across the Shackleton System. In 2021, ice speed ranges from ~0.2 m day-1 in the area between the grounding line and Masson Island on the Shackleton Ice Shelf to ~5 m day-1 on the floating tongue of Denman Glacier (Fig. 8a). Surface ice speeds observed along Roscoe and Scott glaciers reach 1-2 m day-1 and 2-3 m day-1, respectively. Recent changes in ice speed, derived by differencing the mean speed across the whole system between 2021 and 2018, are confined to the seaward ~ 60 km of the floating tongue of Scott Glacier and the Shackleton Ice Shelf (Fig. 8b). On the floating portion of Scott Glacier increases of > 10 % occur across the outer 50 km (Fig. 8b). The western side of the Shackleton ice shelf, including the fast ice to the western side, appears to have decelerated over the same time period, where a change of between -4 % and -6 % is observed (Fig. 8b). There is some evidence of deceleration on the eastern side of the ice shelf, although the signal is unclear with values ranging between +/- 5 %. Annual ice speed percentage differences illustrate the increase in ice speed on the eastern flank of the ice front of Scott Glacier between 2017 and 2018 (Fig. 9a), which then appears to decelerate between 2018 and 2019, when a 4 % increase in the ice speed of the western flank

of the ice front is observed (Fig. 9b). The increase in speed continues through 2019-2020 with a 10% acceleration from the ice front up to 45 km upstream and across the entire ice tongue width (Fig. 9c). The increase extends a further 15 km in the up-flow direction between 2020 and 2021 (Fig. 9d). There is more spatial variability in the annual speed percentage differences across the Shackleton Ice Shelf. The overall trend between 2018 and 2021 appears to be deceleration but there are small regions of acceleration and much of the variability is within the uncertainty bounds, thus complicating interpretation (Fig, 8b).

*L254: Change m/day to m day^{-1} to be consistent with the rest of the paper (ultimately should be m year^{-1}, see general comments)*

We apologise for the inconsistency in unit form and have checked the manuscript carefully ensuring m day$^{-1}$ and m year$^{-1}$ are used throughout.

*L255: "Speeds ~ 10 km either side of the grounding line . . ." confuses me a bit. This sentence makes it sound like you are talking about grounded ice as well (since 10 km on either side of the grounding line would extend 10 km upstream into grounded ice), but I believe you are talking about points 10 and 11 in figure 8a. Perhaps it would be better to say "Speeds up to __ km downstream of the grounding line show . . ." . I also think it would be helpful to reference the specific points in figure 8a that you are discussing (e.g. in L258, "close to the ice front (point 13 in fig. 8a)").*

We agree that the statement is confusing as point number 10 is approximately located on the grounding line (as identified from Measures data). We have changed the section to read

L225-237: Point locations on Shackleton Ice Shelf vary between ~ 0.2 m day$^{-1}$ at the grounding line (point 3 and 4 in Fig. 8b) and ~ 1 m day$^{-1}$ towards the front of the floating ice (point 2 in Fig. 8b), with no consistent temporal trends (Fig. 10a). Denman Glacier exhibits higher speeds, from < 2 m day$^{-1}$ upstream of the grounding line (point 5 in Fig. 8b) to ~ 5 m day$^{-1}$ on the floating tongue (point 9 in Fig 8b) but speeds remain constant through time at each point location (Fig. 10b). Scott Glacier has a similar spatial pattern with speeds increasing from ~ 1.2 m day$^{-1}$ at the grounding line (point 10 in Fig. 8b) to > 4 m day$^{-1}$ close to the floating ice front (point 13 in Fig 8b; Fig. 10c). There is no observable change in speed within 10 km of the grounding line of Scott Glacier (points 10 and 11 in Fig. 8b; Fig. 10c). However, the downstream 30 km of the floating ice tongue show significant acceleration from the beginning of 2020 through to May 2021 (points 12 and 13 in Fig. 8b; Fig. 9c). Over the 17-month period, ice speeds increase ~ 30 % to 2.5 m day$^{-1}$ 30 km from the ice front (point 12 in Fig. 8b) and ~ 40 % to 3.2 m day$^{-1}$ close to the front (point 13 in Fig. 8b; Fig. 10c). Roscoe Glacier has similar ice speed spatial patterns to both Shackleton and Denman, with slower speeds of ~ 0.4 m day$^{-1}$ at the grounding line (point 14 in Fig. 8b), increasing to ~1.2 m day$^{-1}$ close to the floating ice front (point 16 in Fig. 8b) and no significant change in speed through time (Fig. 10d).

*L265: The format of ((b) in Fig. 10)(Furst et al., 2015) is a bit crowded. Instead of using double parentheses, change to (label-b in Fig. 10, Furst et al., 2015). Same with L266 and L267.*

We have changed the labels and parentheses and the section now reads

L240-244: Pinning points have previously been identified at the front of the Roscoe - Shackleton shear margin (label-a in Fig. 10) and upstream of rift 2 on the Shackleton Ice Shelf (label-b in Fig. 11, Fürst et al., 2015). There is evidence of two additional pinning points, Chugunov Island at the front of the Denman-Scott shear margin (label-c in Fig. 11) and at the ice margin of Shackleton Ice Shelf (label-d in Fig. 10). The latter coincides with a local topographic high in ocean bathymetry (Arndt et al., 2013).

*L267: Change "rise in the ocean floor" to "local topographic high in ocean bathymetry".*

We have made the change and the sentence now reads

L244: The latter coincides with a local topographic high in ocean bathymetry (Arndt et al., 2013).

*L269-L283: When describing the model results, I would stray away from using actual years (e.g. ". . . of Denman Glacier occurs after 2150 . . ."), as you are modeling with unrealistic forcing. Instead, I would change this to something like ". . . of Denman Glacier occurs 150 years into the model simulation . . .".*

This section as been removed from the manuscript as outlined above.

*L276: The dynamic response of the system seems pretty significant, as Denman Glacier retreats more than 100 km upstream and Denman and Scott Glaciers end up connecting around a topographic high.*

This section has been removed from the manuscript as outlined above.

*L285: This first sentence of the discussion section contradicts existing literature (e.g. Barancato et al. 2020; Miles et al. 2021), which have cited patterns of grounding line retreat and ice velocity change that appear to be ocean induced since the 1970s or so. This study only looked at changes in surface features through the 60 year observational period and velocity changes over the past ~15 years; however, it seems that changes outside of those presented in this paper are occurring over those timescales. As such, I don't think the authors can claim, based on the presented manuscript, that the Queen Mary and Knox coasts have not changed significantly in the last 60 years.*

Our own reconstructions of longer-term ice flow velocity change (Luckman, unpublished data) very much match those reported previously, e.g. in Miles et al. (2021). We refrained from repeating these matching observations here but have provided them as a wider framework to place our more recent inferences of (no substantial) ice flow velocity within. Pre-2000 data points are sparse and as we emphasise later in this paragraph (L282-286 below). Whilst an increase in ice flow speed of the Denman Glacier clearly occurred between the 1970s and 2017 detailed examination of the timing of this change (as matched by our own feature-tracking based inferences) reveals that almost all of it happened sometime between 1972 and 1989, with very little change since.

L282-286: An increase in ice flow speed was observed just upstream of the Denman Glacier grounding line between 1972-4 and 1989 and, to a lesser extent, through to 2008 (Miles et al., 2021, their Fig. 3c). Variability in ice flow speed then became

insignificant through to 2016-17 (Miles et al., 2021, their Fig. 3c), a pattern that has continued since (Fig. 8b, 10b) and, accordingly, is not associated with change in the surface structure of the system (Fig. 2).

*L291: change "groundling" to "grounding"*

We have made the correction.

*L310-315: I don't think the authors can make this claim based on a single transient model run of the Denman/Shackleton system. The Denman Glacier grounding line is currently retreating under present day forcing conditions (Barancato et al. 2020) and this retreat could be susceptible to the marine ice sheet instability, as the bed upstream of the current grounding line position is retrograde. Your model results show > 100 km of grounding line retreat by 2310 over Denman Glacier, which is significant. However, you did not test the response of the system to realistic forcing over the same timeframe, so we cannot make a statement on the sensitivity of the system. Lastly, the Aurora and Wilkes subglacial basins are not included in the model domain, so this last statement is a speculative conclusion rather than one based on your modeling results and should not be included in the manuscript.*
*L317-L319: It is a great addition to include model limitations in the discussion section; however, unless you are running a thermal model, I do not believe that the geothermal heat flux is used by the ice sheet model. In addition, the ocean conditions and bathymetry will not impact your model run because you assume near-instantaneous disintegration of the floating ice shelf. In the modeling results that you presented, I would expect the results to be primarily impacted by mesh resolution (I am assuming you are not using adaptive mesh refinement, so as the grounding line retreats upstream, the size of the elements will most likely become larger), poorly constrained basal friction and ice stiffness parameters that do not change in time, use of a 2D stress balance approximation instead of a higher order model, etc. It would be helpful to the reader to know exactly how the limitations you listed impact your model (e.g. poorly constrained basal hydrology leads to a poorly constrained basal friction parameter, ice properties impact the ice stiffness parameter, etc.).*

These sections have been removed from the manuscript as outlined above.

*L321-L337: This paragraph lost me a bit. I understand the comparison to Totten Glacier, but I do not think it is appropriate to dive into such a detailed discussion of the subglacial conditions of Queen Mary Land because it does not connect to the rest of the paper. If the authors want to speculate on the subglacial conditions, they need to tie it back to the conclusions of the paper (i.e. its impact on enhanced ice shelf basal melting rates near the grounding line, reducing basal friction at the ice-bed interface, etc.). Without that obvious connection to tie back to the rest of the paper, this paragraph seems out of place and left me confused.*

Both Anonymous Referees were left confused by this paragraph and it is clear that we need to much better clarify its logic connection with our observations reported earlier. Our discussions as a whole are intended to place our observations within the wider framework of the governing atmospheric, oceanographic and subglacial (incl. solid earth) settings, setting the scene for possible future investigations of key properties and processes that we do not understand well at present (of which there are many). We have comprehensively revised the paragraph and the parts of the manuscript that it connects to. The paragraph now reads

L298-309: On the nearby Totten Glacier, inferences from combined geophysical exploration and numerical modelling are consistent with areas of high basal melt rate coinciding with significant grounding line retreat, possibly linked to channelized subglacial meltwater discharge (Dow et al., 2020). By analogy, the deep trough beneath the Denman Glacier is also likely to favour vigorous channelization of the

subglacial meltwater system close to the grounding line. As freshwater outflow into the sub-ice shelf ocean cavity can locally enhance basal melt rates melting near the grounding line (Jenkins, 2011; Wei et al., 2020), the recently observed retreat of Denman Glacier towards the deepest continental trench on Earth could result from complex interplay between the ice, ocean, and subglacial environments. In addition, interaction of Denman Glacier with neighbouring Scott and Northcliff glaciers, as well as with the numerous surface features and pinning points highlighted in this work, could influence how this ice mass responds to future changes in forcing conditions. As this sector of Antarctica holds over 2 m of global sea level potential, it is critical that we continue to both support numerical modelling efforts of this region and monitor short- and long-term changes of the Shackleton system.

L340: See previous comment about L285.

Please see our reply above.

Response to Anonymous Referee #2

We thank Anonymous Referee #2 for their considered review of our manuscript. Below we respond to each comment, with the reviews original comment shown in italic text and our response in blue text.

*Review: This manuscript uses a suite of remote sensing data to map changes in ice extent, structure and velocity across the wider Shackleton system. The manuscript compliments recently published work showing recent grounding line retreat and acceleration of Denman Glacier, but also includes detailed and novel observations across the wider understudied Shackleton system. The authors use these observations to conclude that there has been limited change across the Shackleton system across the observational time period. They also then simulate the response the Shackleton system to a hypothetical loss of floating ice to demonstrate the systems sensitivity to any future ice shelf loss.*

*Overall, I think the manuscript contains some interesting and novel observations of the Shackleton system that are worthy of publication. In particular I think the 50 year record of rifting evolution across the Shackleton Ice Shelf is a very nice contribution. The background here is while there has been some recent studies focussing on Denman Glacier, we know very little about the recent behaviour of the many other glaciers that feed the wider Shackleton system, so these results are valuable. However, at the moment I think these interesting results are somewhat lost in the manuscript. It seems like quite a jump and a distraction from discussing these detailed annual scale observations, to discussing the response of the Shackleton system to the hypothetical loss of all floating ice 400 years in the future, which then turns into a discussion as to how the deep trough of Denman may favour vigorous channelization of the subglacial meltwater system close to the grounding line. At the moment I am not sure what the main focus of the manuscript is. I think the manuscript would benefit from being more streamlined, with a greater focus on the novel observations. I have included some more detailed comments below.*

These comments echo those of Anonymous Referee #1 and have inspired us to remove the modelling part of the manuscript and make it the focus on a follow-up manuscript instead. In making appropriate revisions to the manuscript as a whole this will allow us to re-focus the manuscript on the novel observations, as suggested by the Anonymous Referee #2.

*Observations: The authors state that there have been no significant annual variations in ice flow speed across the Shackleton system. I would argue that the use of 'significant' is not appropriate, what is 'significant' variations greater than 50 m yr⁻¹ 100 m yr⁻¹ etc?.*

In this context we are using significant to mean greater than the uncertainty and, in this case, we do not find that any changes are greater than the uncertainty. We have clarified this where possible throughout the manuscript.

*The plots in Figure 9 give a good overview of the longer-term changes in ice flow speed across the region. However, because they are in m/day and the scales are somewhat stretched it is difficult to determine if there has or has not been any annual variations in ice flow speed. A variation of 0.3 m/day equates to around 100m yr, which would be larger than the uncertainty of the velocity products and would be an interesting result. Are there similar scale variations, particularly in the faster flowing sections of the Shackleton system, to my eyes it looks that there could be, but I could be wrong, I really cannot tell from the plot alone?*

May we refer to our corresponding answer to Anonymous Referee #1.

*May we clarify that m/year is commonly used when the temporal resolution of available data is greater than one year. For this reason, when describing the movement of structural features and ice frontal positions we have used the unit m/year to describe the longer-term trends as the measurements are based on data with annual or multi-annual temporal frequency. When describing the ice speed data from feature tracking, we have deliberately used the unit m/day because we are using much higher temporal resolution data. While we are happy to include annual trends for the data, ice speed does vary on much shorter timescales and by simply changing the ice speed data to m/year we would lose this valuable information.*

In summary, the use of both m/year and m/day is deliberate as it appropriately reflects the temporal resolution of the available data. Respectfully, in our experience it is usually incorrect to state that "0.3 m/day equates to around 100m yr" because our experience with tracking glacier flow speeds over several decades shows that velocities can change on a daily basis.

*In Figure 8 the authors plot an ice speed difference map between 2019 and 2020, I think to illustrate the acceleration of parts of the Scott Glacier. I think these plots are useful in giving a broad overview of the changes in ice speed. Could they also do this over a longer time period, maybe 2000-2010 and 2010-2020 (brackets whatever the availability of velocity data allows)? This would probably be the best visualization of the speed changes over the observational period.*

We have now changed the difference maps to percentage difference maps, scaled to +/- 10 % as this provided the clearest visualisation of change (and is in line with the reporting style of others e.g., Miles et al 2019). We have included a new Figure 8b showing the longest difference period we can produce 2021-2018 which as longer period as the data from Sentinel-1 allows, prior to 2018 the data is much nosier and the resulting difference maps are very messy. We have added another figure, new Figure 9, to show the annual difference maps 2021-2020, 2020-2019, 2019-2018 and 2018-2017.

[Figure]

Figure 8: (a) Mean speed for 2022 with velocity arrows and (b) percentage difference in mean speed between 2021 and 2018, scaled between +/- 10%, with point locations illustrating the ice speed timeseries in Figure 10.

Figure 9: Percentage difference in mean speed between (a) 2018-17, (b) 2019-18, (c) 2020-19 and (d) 2021-20 scaled between +/- 10%.

*One of the most striking observations is the migration of the shear margin near Chungunov Island over just a few years. This is a somewhat unique observation. The authors have collated this wide range of velocity data, while they have tended to focus of ice speed, could they also focus on the velocity directional data? Is this change in shear margin caused by the whole Denman ice tongue 'wobbling' or a more localised change to do with Scott?*

We can see from Figure 3b that Denman tongue does not appear to be 'wobbling' and all of the evidence indicates that changes are confined to Scott, likely a combination of ice front calving close to Mill Island, the rift opening from Chugunov Island towards Mill Island and the speed increases. We know from previous experience plotting directional arrows onto individual ice speed maps provides a much clearer illustration of any changes in velocity direction. We have added ice speed directional arrows to Figure 8a (see response to previous comment), mean ice speed over the whole Shackleton system for the year 2021. We have also made an additional figure that includes mean ice speed and directional speed arrows for 2017, 2018, 2019 and 2020. We have not included this in the manuscript as it does not show any change in flow direction, but we would be happy to include in a supplement if needed.

[Figure]

Figure S1: Mean speed with velocity directional arrows for the years (a) 2017, (b) 2018, (c) 2019 and 2020.

*Modelling: My expertise lies in remote sensing, so I am not in a position to comment on the methodological details of the modelling. But I did find the description of the modelling experiment carried out to be lacking. In the discussion it is stated that:*
*'The upper limit scenario of forcing in our BISICLES model runs suggests that noticeable grounding line retreat occurs in the Denman Glacier over the simulated 400-year time period' But in the methods section there is no mention of an upper or lower limit scenario, nor any mention of the timescales of the simulation. Aside from the basic description of model, there is only a very limited description of simulation. It is essential that the details here are expanded.*

*In wider point, while I think it could be a useful contribution to repeat the experiment in Martin et al., but with BedMachine, I did feel the modelling appeared as somewhat left field in the manuscript and appears as a bit of a jump in the discussion from the main body of observations. The general tone of the manuscript is that very detailed remote sensing observations have shown limited changes in the floating ice in the Shackleton system... But then the manuscript jumps to.. 'now we simulate the unrealistic loss of all floating ice in the Shackleton system'... The scientific rationale for this is unclear to me? Of course, it is entirely up to the authors, but I would point out that I think the detailed observations have the potential to be a nice contribution alone.*

The term upper limit was used to indicate the upper limit of change in the system by removing all of the floating ice, rather than the upper limit of a series of scenarios. May we refer Anonymous Referee #2 to our response to Anonymous Referee #1 for an explanation of the anticipated logic of connection between observations and modelling in our original manuscript.

Following up on these matching anonymous referee comments, and also explained in that response, we have separated the observational and modelling components into two manuscripts, removing the modelling from this manuscript which already largely focused on observations.

*Specific comments:*

*Line 59: I would not describe the acceleration of Denman Glacier since the 1970s as a short- term fluctuation*

Whilst an increase in ice flow speed of the Denman Glacier clearly occurred between the 1970s and 2017 detailed examination of the timing of this change (as matched by our own feature-tracking based inferences) reveals that almost all of it happened sometime between 1972 and 1989, with very little change since. We have removed the 'short-lived description and emphasised the more recent stability in ice flow speed. The paragraph now reads

L56-64: Despite the lack of an embayment and with the exception of localised fluctuations in ice flow, the Shackleton system has so far shown few signs of major dynamic change, with its flow restrained by islands, ice rises, and ice rumples (Stephenson et al., 1989; Young, 1989). Analysis of Envisat data indicated that the Denman Glacier was thinning by 0.4 m year-1 upstream of its grounding line between 2002 and 2010 (Flament and Rémy, 2012), and a 5.4 ± 0.3 km grounding line retreat was detected between 1996 and 2017–2018 (Brancato et al., 2020). Ice velocity data from the region are sparse before the late 2000s, but recent work

identified an increase in ice velocity of the Denman Glacier of 16 % since the 1970s (Rignot et al., 2019), with an increase of 11± 5 %  just upstream of the Denman grounding line between 1972-74 and 1989 and a more recent decrease of acceleration rate to 3 ± 2 % between 1989 and 2007-08 (Miles et al., 2021, their Fig. 3c).

*Line 62: 'the glacier' – please clarify in the text that you are presumably referring to Denman Glacier.*

Yes, we are referring to the Denman Glacier and the sentence now reads

L58-59: Analysis of Envisat data indicated that the Denman Glacier was thinning 0.4 m year-1 upstream of its grounding line between 2002 and 2010 (Flament and Rémy, 2012),

*Line 113: Landsat 1 was not mentioned in the above paragraph? Was it used?*

Landsat 1 MMS acquired 27th February 1974 was used in Figure 5b and has been added to the paragraph describing the data used, which now reads

L97-100: Ice velocity data from the region are sparse before the late 2000s, but recent work identified an increase in ice velocity of the Denman Glacier of 16 % since the 1970s (Rignot et al., 2019), with an increase of 11± 5 % just upstream of the Denman grounding line between 1972-74 and 1989 and a more recent decrease of acceleration rate to 3 ± 2 % between 1989 and 2007-08 (Miles et al., 2021, their Fig. 3c).

*Line 118-129: What software was used for the feature tracking?*

The standard Gamma software was used, this information has been added to the manuscript which now reads

L110-112: Using the standard Gamma software and following commonly adopted methods, feature tracking uses cross-correlation to find the displacement of surface features between pairs of images, which are then converted to velocities using the time delay between those images (Luckman et al., 2007).

*Line 164: That of Rignot (2011) – Do you mean the MEASURES ice velocity mosaic?*

This section has been removed from the manuscript as outlined above.

*Line 173-178: I think much more detail is needed here (see comment above)*

This section has been removed from the manuscript as outlined above.

*Line 189: Not sure 'remarkable' would be the word I would use here*

We have removed remarkable from the sentence which now reads

L143: The floating ice front of Scott Glacier has experienced more variability than that of Denman or Shackleton (Fig. 2a).

Line 243-245: Its nice to see the changes between 2019 and 2020, but I think you could also expand to include 2000-2010 and 2010-2020, maybe even 2000-2020

As outlined above in response to general comments, we have now included 2021-2018 percentage difference map which as longer time period as the data from Sentinel-1 allows, prior to 2018 the data is much nosier and the resulting difference maps are very messy. We have added another figure, new Figure 8, to show the annual difference maps 2021-2020, 2020-2019, 2019-2018 and 2018-2017.

*Line 250: Please use the correct citations for the Measures and ITS LIVE velocity products. I think the correct details can be found on each respective website.*

The correct citations have been included for both and the sentence now reads

L223-225: Ice speed extracted from Sentinel-1 provides a timeseries between 2017-2021, which we extend back to 2002 using MEaSUREs (Mouginot et al., 2012, 2019; Rignot et al., 2011) and ITS_LIVE (Gardner et al., 2018, 2021) in locations where available to highlight variability through time across the system (Fig. 8 and 10).

*Line 267: Is d Chugunov Island?*

C is Chugunov Island and has been relabelled accordingly, the sentence now reads

L242-244: There is evidence of two additional pinning points, Chugunov Island at the front of the Denman-Scott shear margin (label-c in Fig. 11) and at the ice margin of Shackleton Ice Shelf (label-d in Fig. 10). The latter coincides with a local topographic high in ocean bathymetry (Arndt et al., 2013).

*Line 269: The paragraph is discussing modelling results. Therefore, I think it needs a new appropriate subsection, currently it is under 'ice flow speed'.*

This section has been removed from the manuscript as outlined above.

*Line 285: 'Not changed significantly' What is 'significantly'? Observations of Denman have shown that its grounding line has been retreating and that it has been accelerating and losing mass, this is contradictory.*

As explained in our response to Anonymous Referee #1 it is natural for systems such as the Shackleton Ice Shelf to undergo some variability. Within the framework of our regional observations of system structural evolution (and indeed modelling experiments) previous authors' inferences point to relatively minor dynamic changes that do not necessarily point to a system that is on its path to instability. We have tried to clarify the use of significant(ly) where used and changed this sentence to read

L246-247: Over the ~ 60-year period of observation, the Shackleton Ice Shelf system has undergone observable variability in velocity and structure, but there is no sustained longer-term change.

*Line 285-302: I think you could be more detailed in this section. One of the key results here is the lack of propagation of the rifting across the Shakleton over the past 50 years, with the exception of some small activity over the past few years. This is a key result. What does this mean and how does it tie into the wider body of literature? In other regions rifting evolution has been key to ice shelf stability? How important could the melange that fills these rifts be (see recent Larour Larsen C paper in PNAS)?, etc..*

There have indeed been a series of manuscripts that examine the importance of suture zones and their mélange fills on rift propagation, not least those on Larsen C cited here, as led by some of the current authorship team (Kulessa et al., 2014, 2019, both in Nature Communications), that pre-date the Larour et al. manuscript quoted here. We completely agree, rift mélange is indeed critical to ice shelf stability and have added an expanded explanation to the revised manuscript, which now reads,

Ln 264-276: The lack of significant rift propagation on the main body of Shackleton Ice Shelf appears directly related to the Masson Island suture zone. Indeed, there is growing recognition that the softer marine ice present in suture zones inhibits the growth of large-scale fractures, acting to stabilise ice-shelves by reducing local stress intensities (Kulessa et al., 2014; Larour et al., 2021; McGrath et al., 2014). Suture zones are structurally and mechanically heterogeneous and therefore particularly susceptible to micro-crack formation ahead of the rift tip, reducing the stress intensity around the rift tip and potentially causing rifts to be halted by suture zones (Bassis et al., 2007; Kulessa et al., 2019). In addition, stress intensities around the rift tip are likely to be reduced because of sea-water softened suture-zone ice, relative to harder meteoric ice that is colder and contains less liquid seawater (Kulessa et al., 2019). While current observations suggest suture zones promote stability by halting rift propagation, a strong relationship between the thickness of ice mélange and the opening rate of the rifts has been observed, indicating that ice mélange thinning rather than ice shelf thinning can promote rift propagation (Larour et al., 2021). The warmer temperature and increased sea water content of the ice mélange suggests it may be more vulnerable to thinning due to future surface and basal melting than the surrounding meteoric ice, potentially affecting rates of rift opening and propagation (Kulessa et al., 2014; McGrath et al., 2014).

*Line 321-338: I struggle to see the relevance of this section to the manuscript. Fig 1a: The scale is a little difficult to make out because of the transparency*

This paragraph left both anonymous referees confused. We had included it to set the scene for processes that may control the evolution of the ice shelf and for possible future investigations of key properties and processes that we do not understand well at present. We have simplified the paragraph and integrated it more closely with the main points of the manuscript. The section now reads

L298-308: On the nearby Totten Glacier, inferences from combined geophysical exploration and numerical modelling are consistent with areas of high basal melt rate coinciding with significant grounding line retreat, possibly linked to channelized subglacial meltwater discharge (Dow et al., 2020). By analogy, the deep trough beneath the Denman Glacier is also likely to favour vigorous channelization of the subglacial meltwater system close to the grounding line. As freshwater outflow into the sub-ice shelf ocean cavity can locally enhance basal melt rates melting near the grounding line (Jenkins, 2011; Wei et al., 2020), the recently observed retreat of Denman Glacier towards the deepest continental trench on Earth could result from complex interplay between the ice, ocean, and subglacial environments. In addition, interaction of Denman Glacier with neighbouring Scott and Northcliff glaciers, as well as with the numerous surface features and pinning points highlighted in this work, could influence how this ice mass responds to future changes in forcing conditions. As this sector of Antarctica holds over 2 m of global sea level potential, it is critical that we continue to both support numerical modelling efforts of this region and monitor short- and long-term changes of the Shackleton system.

*Fig 10: The high strain rate band going across the Denman ice tongue – presume there is no evidence of any recent rifting in this location?*

No there is not any evidence of rifting in the area, we have examined all of the imagery in detail, but it is an artefact. We have described this clearly in the figure description which now reads

Figure 11: Magnitude of the principal strain rate of the Denman-Scott-Shackleton system derived from Sentinel-1 data. N.B. The feature down flow of pinning point c on the Denman Tongue is an artefact, we see no evidence of rifting in the remote sensing data in this region (e.g., Fig. 2b, 7a).

---

## Referee Report (RR1)

This paper uses 60 years of satellite data and 6 ice penetrating radar transects to highlight changes in the ice structure and ice flow of the Shackleton Ice Shelf region, where results largely focus on the ice shelf, the well-studied Denman Glacier and the lesser analysed Scott Glacier. The results the authors present are important, and for the area outside of Denman Glacier, they are also novel – helping to characterise a poorly understood sector of Antarctica. I commend the authors on their hard work, as it must have taken a long time to find, collect, process and present the large amounts of data collated in this paper. However, I feel some of the hard work (evident in the methodology and figures) is lost within the text, which is quite poorly structured.

The methodology is sound, and I do not propose any major changes to the manuscript, but I think the paper would benefit from more subheadings and a clearer structure.

Below, you will find my main observations, listed as I work through the manuscript:

1) The introductory paragraph reiterates a lot about what we already know about the general state of the East and West Antarctic Ice Sheets. It should be more area specific.
2) The rest of the introduction jumps around a lot. A more systematic study site description, noting glacier length, bed elevation, ice thickness (from BedMachine) would help to set up context for the paper.
3) The ice penetrating radar methods section lacks detail - including important information like radar frequency and the applied ice velocity.
4) Remove general references to the "Queen Mary and Knox coasts", replacing them with the Shackleton Ice System to reflect the focus of your paper (e.g. in the caption for figure 2). You must also refine your abstract and introduction to state which areas of your paper you will focus on. It sounds like you will analyse all glaciers, but you don't even mention the Northcliffe glacier in the text after the introduction!
5) You talk about directions like north, east, south and west (e.g. in section 3.1 and figure 3) but as north isn't at the top of your figures it's not immediately intuitive what direction you're talking about. Can you also add remarks like inland and offshore as well, to quickly clarify directions (where suitable).
6) Refer to full glacier names in the text and figure captions (not simply 'Scott' – like the last word of paragraph 1 in section 3.1, and in the radar text of section 3.2)
7) The results section isn't very systematic. Not every glacier is described in the same way, for example, there is no information provided about the Apfel Glacier even though you say you will discuss the whole Shackleton Ice Shelf System in the paper. It might be easier to break section 3.2 down further into sub-sections titled ice extent, rifts, strain rate etc. This will help readers interested in a specific component of the glacial system.
8) The ice flow section is clear and well written. I found this very helpful.
9) The discussion isn't very well structured either. It begins by down-playing findings and then launching into radar findings, when it would make more sense to briefly outline how you're going to lay out the discussion, then follow that structure – by either discussing features in turn (as you do in the results) or describing changes by area (like the Denman glacier paragraph does). Many of your results, like ice front positions are not fully discussed, which acts to downplay your important results. Also, you fail to discuss some other important observations, like any distinct seasonal change. Why do you think there are no seasonal changes? Is this common in East Antarctica?
10) The discussion section also contains a lot of results based text, which is presented as almost stand-alone text which isn't used to evidence a discussion.

11) The conclusions don't summarise all the results. They should contain more information on your important findings about rift propagation and shear margin changes as these could be explored further in future work (in detailed models for example) – which would use all the data you have so carefully collected and presented. This is much more important to report than your concluding remarks about needing more data, in a very generic sense.

12) The Data Availability statement doesn't mention the radargrams. If they aren't freely available to download online, can they be requested from the data collectors? Also mention the BedMachine database used in Figure 1.

Suggested figure changes are below (note that all figures appear blurry, but I presume this is a result of copying and pasting):

Figure 1
- Why is the area in panel a so zoomed out? I'd focus in more on the area you examine, maybe just extending as far south as Law Dome so you can really see the detail of the bed topography in your study area.
- In panel b could you make the glacier names stand out in bold as these are key points
- In the caption mention that numbers 1 and 2 refer to rift systems.

Figure 2
- Please add glacier names to panel a to make the figure more useful
- Add rift numbers to panel a too, to help the text in section 3.2

Figure 3
- Change caption to say "Scott Glacier".
- Add glacier name to figure.

Figure 4
- Could you label the rifts differently to those already identified as rifts 1 and 2 elsewhere to avoid confusion? Maybe S1, S2 and S3 here, and D1 and D2 previously?
- Does the arrow point to North? If so, it should say N.
- Write out glacier names in full on panel c, so Scott Glacier etc. The same should be done for Figure 7.
- In the caption put the acquisition dates in brackets

Figure 5
- The caption talks about the "Shackleton Roscoe Glacier". I thought it was just the Roscoe Glacier?
- On line 177 you talk about the margin between the Roscoe Glacier and the Shackleton Ice Shelf. Can you mark this margin on the figure?

Figure 6
- Add year of data collection to each panel

Figure 7

- Radar name text is on the wrong side. Move all radar names to the left side of the transects in panel a as you are looking at the transects from the sea, not from inland.
- The radar bed pick key needs a metric (m a.s.l.?)
- Re-title the radar transect names on the right hand column to match those on the map.
- I'd argue that the power colour bar down each radargram is unnecessary.

Figure 8
- Label glaciers again here (and in Figures 9 and 11). You're used to looking at this area so it's clear to you what's what, but most readers won't be.

Additional, minor comments:

Title: Maybe Glaciological history is a better term than setting, which doesn't really mean much.

Line 28: Be more specific. From what speed to what speed?

Line 30: How do you know acceleration in ice flow and calving are linked?

Line 33: Presumably sediment filled trough rather than just bedrock.

Line 48: This paragraph jumps around. Stick with basin introductions first then move onto the floating components. A more detailed site overview is needed too – to include glacier lengths, thicknesses etc (see points above).

Line 56: What does the lack of an embayment mean in terms of 'major dynamic change'? I fail to see the relevance.

Line 57: Refer to figure 1

Line 62: You've just said the data is sparse, so how reliable is this information? Can you note any error margins?

Line 71: I presume you mean average basal melt rates rather than total melt rates.

Line 73. Introduce all acronyms. The information about the Sabrina Coast doesn't seem to have much relevance to your work but I could be reading this incorrectly.

Line 74: But how likely is this given what you say in the next sentence? Probably best not to jump to possibilities here.

Line 79: I think you mean grounded.

Line 87: You don't mention the multi-decadal timeframe you talk about in the abstract.

Line 92: Just talk about the Shackleton system rather than the whole coastal area.

Line 83: Can you briefly state why you chose every 6 months, presumably you have data from February and July/August then? Is that just because of daylight? March to September might allow you to capture more of the seasonal changes? Note that I'm not asking you to change your data analysis period! Just note exactly which months you studied and why you made that call.

Line 94: The datasets need referenced, or you at least need to provide reference to a data availability section of the paper.

Line 110: Reference required for Gamma software

Line 113: How short? Can you give a range?

Line 114: The cold temperatures is an assumption. Delete this.

Line 115: Can you note how many image pairs you used to show off the rigour of your work?

Line 125 (and the rest of the paragraph): More detail required. What frequency did the radar operate at? How was the data topographically corrected? What velocity did you use? Did you apply a firn correction?

Line 135: Do you mean no obvious change in the annual rate of advance?

Line 137: Which front are you talking about?

Line 139: The lack of seasonal change is interesting is it not? So something you should talk about in your discussion?

Line 145: How do you know it's the calving that's causing the retreat and not faster ice flow or melting?

Line 171: How do you define the grounding line?

Line 179: What happened prior to this? Is there a reason why you don't explore data prior to 2015 here?

Line 183: Why don't you talk about any features on the Apfel Glacier?

Line 186: Reference not needed as it's your observations that show this.

Line 188: Why is this remarkable? A statement like that needs to be backed up by other references that say this situation is really unusual. I'd use the word notably, or something similar instead.

Line 191: You should say the radar provides a snapshot of information.

Line 195: Split is the wrong term. The ice doesn't fracture apart.

Line 199: What's consistently noisy? Radar returns? The bed?

Line 232: Use precise dates here rather than 'beginning'.

Line 239: What time span do you analyse here?

Line 241: I think you mean to refer to figure 11.

Line 247: What do you mean by that? Longer-term change in what? Glacial output?

Line 274: This reads like a literature review. How does it relate to your findings? You never talk about melange.

Line 275: Of what ice melange? At your study site?

Line 278 (the paragraph): Contains lots of results again, but you need to discuss what the results mean for your study site and out knowledge of antarctica and processes as a whole.

Line 294: What do you mean by 'connection'?

Line 305: How do your results underpin this hypothesis?

Line 320: It's unclear exactly what data you need more of. Do you need better bed topography data in the region? Or do you want better ocean temperature data? How will a knowledge of local geology help? This statement is rather generic and therefore unhelpful.

---

## Author Response (AR2)

**Response to Anonymous Referee #1**

We thank Anonymous Referee #1 for their considered review of our manuscript. Below we respond to each comment, with the referee's original comment shown in italic text and our response in blue text.

*Summary: This revised manuscript by Thompson and others focusses on some very detailed and important observations of the Shackleton-Denman system. They show that there has been very little structural change over the past 60 years and no obvious interannual variability in ice flow speeds over the past 20 years, with the exception of the floating section of Scott Glacier. I think these are important observations in the context of long-term change in this understudied section of the East Antarctic Ice Sheet, and I recommend publication in The Cryosphere. However, I do have a few relatively minor comments that are mainly focussed on wording surrounding the long-term changes across the Shackleton system that I think can be easily addressed/clarified.*

*Sustained change/long-term vrs short-term/major change – There are several sections in the manuscript that refer to the absence of a sustained long-term major change in the Shackleton system, many of which I have highlighted below. I appreciate there is an element of subjectivity in what exactly constitutes a sustained or major change, but I find it difficult to believe that a 15% acceleration since the 1970s, current grounding line retreat, inland thinning and mass loss does not constitute a sustained major change at Denman. I think this is just a case of re-wording some of these sentences. On some occasions, it is not clear if you are discussing the wider Shackleton system, or the Shackleton Ice Shelf.*

We thank Referee #1 for their considered comments and agree that the language used to describe change is subjective in places. We have re-structured and added to sections of the introduction, results, discussion and conclusion in response to detailed comment from Referee #2 (see below) and in doing so have ensured that we focus on the main the observations of this work and tried to avoid any subjective descriptions of change in the system. Throughout the whole manuscript we now refer to the Shackleton system as the region of observation of which the Shackleton Ice Shelf is a part of.

*Minor comments*

*Line 26: "Over the 60-year period of observation, we find no evidence of either longer-term sustained change or significant annual or subannual variations in ice flow"*
*Long-term change in ice structure or ice speed, or both? I presume you mean ice structure because Denman has accelerated some 15% since the 1970s, but this is not clear because the previous sentence mentions ice velocities. Please clarify*

We have changed this section of the abstract to reflect our key observations, the section now reads

Line 24-33: Over the 60-year period of observation we find significant rift propagation on the Shackleton Ice Shelf and Scott Glacier and notable structural changes in the floating shear margins between the ice shelf and the outlet glaciers, as well as

features indicative of ice with elevated salt concentration and brine infiltration in regions of the system. Over the period 2017-2022 we observe a significant increase in ice flow speed (up to 50%) on the floating part of Scott Glacier, coincident with small scale calving and rift propagation close to the ice front. We do not observe any seasonal variation or significant change in ice flow speed across the rest of the Shackleton system.

*Line 28: "A previously mapped increase in the ice flow speed of the Denman Glacier is not observable beyond 2008". Unless I have missed it, I do not see anything in the main body that suggests previous literature has suggested in increase in flow speed of Denman Glacier post-2008? Or do you mean that other studies have observed an increase in flow speed of Denman pre-2008, but you observe no change post-2008. Please clarify and re-write appropriately.*

We have removed this sentence from the abstract to focus on our key observations.

*Line 72-75: Understand that this study has been published post-submission of this manuscript, but the below paper seems highly relevant here. You may wish to consider adding it to the discussion:*

*Herraiz-Borreguero & Naveira Garabato: Poleward shift of Circumpolar Deep Water threatens the East Antarctic Ice Sheet, Nature Climate Change, 2022*

We thank the review for highlighting this manuscript, we have a discussion of this to the section which now reads

Line 76-83: The high basal melt rates close to the grounding line in this region of Antarctica have been linked to a warming of up to 0.5 °C over the last 40 years that occurred in the open ocean off East Antarctica, concurrent with an even more pronounced warming of 0.8–2 °C observed over the continental slope (Herraiz-Borreguero and Naveira Garabato, 2022). This Circumpolar Deep Water (CDW) warming is linked to a poleward shift of the Antarctic Circumpolar Current's southern extent onto the Indian Ocean sector of the East Antarctic continental slope, in which the Shackleton Ice Shelf is located. The continental slope warming appeared strongest near ice shelves that are thinning or have retreating grounding lines such as the Denman Glacier (Herraiz-Borreguero and Naveira Garabato, 2022).

*Line 135: Its not clear to me what the shelf's "central front" is? I would try and be more specific here and refer to the "Shackleton Ice Shelf's central front". In my mind I immediately thought of Denman when I read "central front".*

We simply mean the front of the Shackleton Ice Shelf and have changed the sentence to clarify this, the sentence now reads

Line 154-155: Over the same time period, the front position of the Shackleton Ice Shelf advanced a total of 18 km with no obvious change in the annual rate of advance (Fig. 2a).

*Line 155: System 1 and system 2. Maybe clarify in the caption of Figure 1, the '1' and '2' in figure 1b refer to these rifts.*

We have added the rifts to the figure caption, now referred to as R1 and R2. The caption now reads

Line 628-630: Figure 1: (a) Area of focus in the regional context of the Aurora and Wilkes subglacial basins, location shown in inset (Background: BedMachine V2 (Morlighem et al., 2020). (b) The Shackleton system overview in February 2021 with the two main rift systems on Shackleton Ice Shelf labelled R1 and R2. All features mapped from Sentinel 2A and 2B imagery acquired 05th – 27th February 2021.

*Line 245: "Over the ~ 60-year period of observation, the Shackleton Ice Shelf system has undergone observable variability in velocity and structure, but there is no sustained longer-term change."*

*I disagree with the wording here. Denman has accelerated by some 15% since the 1970s and its grounding line is retreating. This is by definition a long-term change? Or are you solely referring to the Shackleton Ice Shelf here? To my mind "Shackleton Ice Shelf system" includes Denman, Scott etc, you even mention this in see line 47. I think this sentence needs to be re-worded/clarified.*

We have changed the structure and emphasis of the discussion section, this sentence now reads

Line 283- 286: Over the ~ 60-year period of observation, the Shackleton system has undergone observable variability in ice velocity and structure, although more frequent satellite observations in recent years have not revealed any distinct seasonal or annual cycles of variability across it. We discuss the observable changes by sub-system region, while considering their possible implications for the whole Shackleton system.

*Line 298-309: You may wish to include some discussion of the Herraiz-Borreguero paper here, it seems highly relevant in this paragraph discussion.*

We have changed the emphasis and structure of this section and included the work presented by Herraiz-Borreguero, the section now reads

Over the 60-year period of observation of the Shackleton system we observe significant rift propagation on the Shackleton Ice Shelf and Scott Glacier, and notable structural changes in the floating shear margins between the ice shelf and the outlet glaciers. Over the period 2017-2022 we observe a significant increase in ice flow speed (~ 50 % close to the ice front) on the floating part of Scott Glacier. Over the same time period we do not observe any seasonal variation or significant change in ice flow speed across the rest of the Shackleton system, and there is no observable change in the ice flow speed of the grounded ice or at the grounding line. However, given the likelihood of modified CDW (observed to be 0.8–2 °C warmer) accessing the shelf (Herraiz-Borreguero and Naveira Garabato, 2022), coupled with the observations that the outermost portions of Shackleton Ice Shelf experience the

most intense surface melt outside of the Antarctic Peninsula (Trusel et al., 2013), and a lengthening of the melt season (Zheng et al., 2018), the Shackleton system appears vulnerable to changes in both ocean and atmospheric forcing. Indeed, observed grounding line thinning and retreat (Brancato et al., 2020; Flament and Rémy, 2012) and localised ice flow speed change (Miles et al., 2021; Rignot et al., 2019) may indicate that the Shackleton system is already responding to changes in forcing. However, the timescales over which the system responds, and the implications of this response remain challenging to predict.

*Line 311-314: Again, I disagree with the wording here. Denman is flowing 15% faster than it did in the 1970s, its grounding line is retreating, there is evidence of inland thinning and the catchment is losing mass. I find it difficult to see how this is anything other than a sustained major change? I do think this needs to be re-worded.*

We have re-worded this section to reflect the changes we observe across the system and their possible implications, highlighting that the timescales over which the system responds to changes in forcing remain challenging to predict. The section now reads

Line 407-429: Indeed, observed grounding line thinning and retreat (Brancato et al., 2020; Flament and Rémy, 2012) and localised ice flow speed change (Miles et al., 2021; Rignot et al., 2019) may indicate that the Shackleton system is already responding to changes in forcing. However, the timescales over which the system responds, and the implications of this response remain challenging to predict.

We still do not have a clear picture of the cause of the 1972-2008 acceleration in ice flow speed of the Denman Glacier or indeed its later stabilisation. A previous calving cycle reconstruction indicated that a calving event at some point in the 2020s is highly likely (Miles er al., 2019) but from our observations we do not find any indication of when this may occur or the mechanism by which it may be initiated. In addition to thinning and induced dynamic changes there are several features of the system that could significantly speed up the response to external forcing. Such features include; (i) thinning suture zones; if the Masson Island suture zone on Shackleton Ice Shelf is thinning in response to ocean warming, the melange at the base of the suture zone will preferentially melt, promoting rift propagation and ice shelf break up (Kulessa et al., 2014; Larour et al., 2021; McGrath et al., 2014); (ii) shear margin weakening; the shear margins across the whole system are likely to weaken with increasing melt and elevated meltwater availability but in the region of the system from Scott Glacier towards Mill Island, potentially high salt concentrations in the ice could further enhance melt and increase deformation; (iii) brine infiltration; possible brine infiltration of the firn layers on Shackleton Ice Shelf could enhance fracture propagation and promote hydrofracture if there is an increase surface meltwater with the observed lengthening melt season; and (iv) subglacial conditions; heat flow anomalies in the region of the Denman Glacier are poorly resolved but recent multivariate analysis of available datasets indicate elevated geothermal heat flow (> 70-80 mW m$^{-2}$) to the west of the Denman region (Wilhelm II Coast) and in the Knox Basin interior (Stål et al., 2021). As observed elsewhere in East Antarctica, the deep trough beneath the Denman Glacier may favour vigorous channelisation of the subglacial meltwater system close to the grounding line (Dow et al., 2020) and

freshwater outflow into the sub-ice shelf ocean cavity could locally enhance basal melt rates near the grounding line (Jenkins, 2011; Wei et al., 2020).

*Line 314: Short lived event – but it has been sustained? The glacier is still flowing faster than it did 50 years ago? It also could be argued the other way around, there has been a long period of acceleration, now there is a short-lived pause in acceleration. We do not know.*

We agree with the reviews concerns over the choice of language and have removed this sentence from the section. The first sentence of this paragraph now reads

Line 411-412: We still do not have a clear picture of the cause of the 1972-2008 acceleration in ice flow speed of the Denman Glacier or indeed its later stabilisation

Fig 1a) Moscow University is labelled in the wrong location. It is hard to see some of the labels, For example the blue labelling on the blue background. In general needs tidying up.

The figure are has changed to reflect the area of observation and all relevant features have been clearly labelled, see below

[Figure]

*Fig 2) I am sure it is just an issue with the PDF uploading, but please double check that the writing on the figures is not blurred when the final figures are submitted.*

This was due to the size of the document uploaded and we will ensure all figures are clear in the resubmission.

*Fig. 2b) It would look nicer if the date labels had some form of white/grey backing box. The black writing is hard to see on top of the dark SAR image. (Like you have done on Figure 4 for example)*

We have changed the figure labelling, see below

[Figure]

**Response to Anonymous Referee #2**

We thank Anonymous Referee #2 for their considered and detailed review that has significantly improved the readability of our manuscript. Below we respond to each comment, with the referee's original comment shown in italic text and our response in blue text.

*Summary: This paper uses 60 years of satellite data and 6 ice penetrating radar transects to highlight changes in the ice structure and ice flow of the Shackleton Ice Shelf region, where results largely focus on the ice shelf, the well-studied Denman Glacier and the lesser analysed Scott Glacier. The results the authors present are important, and for the area outside of Denman Glacier, they are also novel – helping to characterise a poorly understood sector of Antarctica. I commend the authors on their hard work, as it must have taken a long time to find, collect, process and present the large amounts of data collated in this paper. However, I feel some of the hard work (evident in the methodology and figures) is lost within the text, which is quite poorly structured. The methodology is sound, and I do not propose any major changes to the manuscript, but I think the paper would benefit from more subheadings and a clearer structure. Below, you will find my main observations, listed as I work through the manuscript:*

*1) The introductory paragraph reiterates a lot about what we already know about the general state of the East and West Antarctic Ice Sheets. It should be more area specific.*

We agree that the paragraph needs to be more specific to the region of observation and have changed the text to reflect this. The paragraph now reads

Line 39-47: The East Antarctic Ice Sheet has historically been perceived as the stable sector of Antarctica (Silvano et al., 2016); however, it has now emerged that

the subglacial basins of Wilkes Land in East Antarctica have been contributing to sea level rise since the 1980s, with the Aurora subglacial basin contributing 1.9 mm (Rignot et al., 2019). The Shackleton system, fed by the Knox subglacial basin, is thought to be the most direct connection to the western portion of the Aurora subglacial basin (Fig. 1a). The Shackleton system is one of the largest ice shelf systems in East Antarctica and is comprised of a number of outlet glaciers including Denman, Scott, Northcliffe, Roscoe and Apfel. The floating component of the system is comprised of the Shackleton Ice Shelf together with the distinctive tongues of Denman-Northcliff, Scott and Roscoe glaciers, and an area of fast ice to the west of the Denman Glacier tongue (Fig. 1b). The Denman Glacier alone is estimated to hold an equivalent of 1.5 m of sea level rise equivalent ice mass (Morlighem et al., 2020).

*2) The rest of the introduction jumps around a lot. A more systematic study site description, noting glacier length, bed elevation, ice thickness (from BedMachine) would help to set up context for the paper.*

We agree that a more systematic approach is needed in the introduction and have added the suggested thickness/elevation descriptions to the second paragraph. The remainder of the introduction has been restructured to provide a more logical structure to describe the previous observations. The remainder of the introduction now reads

Line 49-98: Ice thicknesses across the grounded portion of the system range from ~ 400 m inland of Shackleton Ice Shelf to > 4000 m at the Denman Glacier (Morlighem et al., 2020). Bed elevations are also wide ranging with the main outlet glaciers grounded well below sea level and experiencing retrograde slopes (Brancato et al., 2020; Morlighem et al., 2020). Close to the grounding line ice thicknesses range from 300 m to 1400 m, thinning toward the ice front where ice thickness is in the range of 180 – 250 m with the exception of the front of the Denman Glacier tongue at ~ 450 m thick (Morlighem et al., 2020). Ice velocity data from the region are sparse before the late 2000s, but recent work identified an increase in ice velocity of the Denman Glacier of ~ 16 % since the 1970s (Rignot et al., 2019), with an increase of 11± 5 % just upstream of the Denman Glacier grounding line between 1972-74 and 1989 and a more recent rate of acceleration of 3 ± 2 % between 1989 and 2007-08 (Miles et al., 2021, their Fig. 3c). Analysis of Envisat data indicated that the Denman Glacier was thinning by 0.4 m year-1 upstream of its grounding line between 2002 and 2010 (Flament and Rémy, 2012), and a 5.4 ± 0.3 km grounding line retreat was detected between 1996 and 2017–2018 (Brancato et al., 2020).

The discovery of the deepest sub-glacial trough in Antarctica (>3500 m below sea level) beneath the Denman Glacier, with a gentle and slightly retrograde bed slope close to the grounding line (Morlighem et al., 2020), has prompted suggestions that the system may be vulnerable to marine ice sheet instability, potentially triggered by high basal melt rates in the ocean cavity just offshore the grounding line (Brancato et al., 2020; Morlighem et al., 2020; Rignot et al., 2019). Meltwater production from basal melt of the Shackleton system (73 Gt year$^{-1}$) between 2003 and 2008 rivalled that from Thwaites (98 Gt year$^{-1}$) (Rignot et al., 2013). Satellite derived basal melt rates between 2010 and 2018 revealed high but localised average basal melt rates of > 50 m year$^{-1}$ close to the Denman grounding line (Liang et al., 2021), on par with

basal melt rates in the Bellingshausen and Amundsen Sea (Adusumilli et al., 2020). Beyond the Denman Glacier grounding zone, there has been much less observation, although the surrounding Shackleton system has so far shown few signs of major dynamic change, with its flow restrained by islands, ice rises, and ice rumples (Stephenson et al., 1989; Young, 1989) (Fig.1). Much of the Shackleton system has an average basal melt rate of between 0 and 1 m year$^{-1}$, with the exception of the Denman Glacier shear margins where refreezing in the order of 0.5 m year$^{-1}$ is indicated, as well as Roscoe Glacier where average basal melt rates away from the grounding line are 2-3 m year$^{-1}$ (Adusumilli et al., 2020; Liang et al., 2021).

The high basal melt rates close to the grounding line in this region of Antarctica have been linked to a warming of up to 0.5 °C over the last 40 years that occurred in the open ocean off East Antarctica, concurrent with an even more pronounced warming of 0.8–2 °C observed over the continental slope (Herraiz-Borreguero and Naveira Garabato, 2022). This Circumpolar Deep Water (CDW) warming is linked to a poleward shift of the Antarctic Circumpolar Current's southern extent onto the Indian Ocean sector of the East Antarctic continental slope, in which the Shackleton Ice Shelf is located. The continental slope warming appeared strongest near ice shelves that are thinning or have retreating grounding lines such as the Denman Glacier (Herraiz-Borreguero and Naveira Garabato, 2022). Although basal melt is considered the dominant form of melt related mass loss in East Antarctica, the outermost portions of Shackleton Ice Shelf in have been observed to experience the most intense surface melt outside of the Antarctic Peninsula (>>200mm w.e. year$^{-1}$) (Trusel et al., 2013). There is also evidence that the Shackleton Ice Shelf may experience an increase in surface melting with an increase in the length of the melt season observed between 2002 and 2011 (Zheng et al., 2018). Circum-Antarctic studies have identified the Shackleton Ice Shelf as experiencing substantial losses due to thinning in recent decades (Greene et al., 2022) and a reported average thickness change of -3.4 m between 2010 and 2017 (Hogg et al., 2021).

Despite increased scientific scrutiny in recent years (Arthur et al., 2020; Brancato et al., 2020; Miles et al., 2021; Morlighem et al., 2020; Rignot et al., 2019; Stokes et al., 2019), existing data and knowledge are still insufficient to predict the future evolution of the Shackleton system with confidence. Aside from incomplete understanding of the dynamic controls on Denman Glacier flow and Shackleton Ice Shelf stability, almost nothing is known about the adjacent Scott, Northcliffe, Roscoe and Apfel glaciers or their shear margins. Here we add to the previously reported dynamic changes in the Denman Glacier over the 60-year period of observation and place them into the wider regional context of the Shackleton Ice Shelf system. We do so by also presenting an improved biannual temporal frequency of observations in the last seven years (2015-2022) integrating airborne radar data, new satellite observations of ice structure, changes in ice front position and ice-flow velocities, with known geometrical and glaciological constraints. We firstly report on the main structural features and dynamic changes across the whole system and then discuss the changes in the system and their possible impact by sub-system region.

*3) The ice penetrating radar methods section lacks detail - including important information like radar frequency and the applied ice velocity.*

The radar system transmits a frequency modulated chirped signal between 52.5 and 67.5 MHz and a constant ice velocity of 167 m/microsecond was applied to generate each radargram. We have added this information to the ice penetrating radar methods section, which now reads

Line 132-142: The ice-penetrating radar data presented here were acquired on two survey flights using the Snow Eagle 601 BT-67 aircraft (Cui et al., 2018) flown on 19 and 20 December 2018. The data were acquired using a radar system that is functionally equivalent to the High Capability Airborne Radar Sounder that has been described in the literature (Peters et al., 2007) and used in numerous studies of both grounded (e.g. Young et al., 2011) and floating ice properties and grounding zones (e.g. Greenbaum et al., 2015). The phase coherent radar system transmits a frequency modulated chirped signal between 52.5 and 67.5 MHz. The images presented in this manuscript reflect a postprocessing sequence that coherently adds 10 raw radar records at a time to increase signal to noise, applies matched filtering to account for the chirped transmit pulse, then incoherently stacks the resulting complex valued radar traces five times to suppress speckle noise. The ice bottom elevation data were computed using a semiautomatic approach involving manual localization above and below horizons of interest (the ice surface and ice bottom interfaces in this instance). A constant ice velocity of 167 m/microsecond was applied to generate each radargram.

*4) Remove general references to the "Queen Mary and Knox coasts", replacing them with the Shackleton Ice System to reflect the focus of your paper (e.g. in the caption for figure 2). You must also refine your abstract and introduction to state which areas of your paper you will focus on. It sounds like you will analyse all glaciers, but you don't even mention the Northcliffe glacier in the text after the introduction!*

All references to Queen Mary and Knox Coasts have been removed from the text and the Shackleton system is used to refer to the region of observation (location shown in Figure 1) throughout the manuscript. Throughout the results and discussion, we have included descriptions of all the outlet glaciers where possible. We have also included area descriptions where no change is observable or a glacier is outside of the available data, for example we have no ice penetrating radar data over Roscoe Glacier.

*5) You talk about directions like north, east, south and west (e.g. in section 3.1 and figure 3) but as north isn't at the top of your figures it's not immediately intuitive what direction you're talking about. Can you also add remarks like inland and offshore as well, to quickly clarify directions (where suitable).*

We agree that this is confusing with the projection used in the figures. In all cases we have changed directions to inland / offshore or described locations in relation to other observed features, e.g., adjacent to Mill Island or adjacent to Chugunov Island.

*6) Refer to full glacier names in the text and figure captions (not simply 'Scott' – like the last word of paragraph 1 in section 3.1, and in the radar text of section 3.2)*

We have changed all references in text and figures to reflect the full glacier name.

*7) The results section isn't very systematic. Not every glacier is described in the same way, for example, there is no information provided about the Apfel Glacier even though you say you will discuss the whole Shackleton Ice Shelf System in the paper. It might be easier to break section 3.2 down further into sub-sections titled ice extent, rifts, strain rate etc. This will help readers interested in a specific component of the glacial system.*

We agree that subsections make the results much clearer to follow and have split the results into 6 subsections, (1) Ice front positions, (2) Rifts and crevasses, (3) Ice extent, (4) Shear margins, (5) Mean ice flow speed and strain rate and (6) Temporal change in ice flow speed. In each of the subsections we have described each of the glaciers of the system in turn or stated why this was not possible. The results section now reads

Line 144-278:

[revised manuscript text omitted]

*8) The ice flow section is clear and well written. I found this very helpful.*

Thank you

*9) The discussion isn't very well structured either. It begins by down-playing findings and then launching into radar findings, when it would make more sense to briefly outline how you're going to lay out the discussion, then follow that structure – by either discussing features in turn (as you do in the results) or describing changes by area (like the Denman glacier paragraph does). Many of your results, like ice front positions are not fully discussed, which acts to downplay your important results. Also, you fail to discuss some other important observations, like any distinct seasonal change. Why do you think there are no seasonal changes? Is this common in East Antarctica?*

We agree that the structure of the discussion was hard to follow and have restructured the section by sub-region, discussing all changes and possible implications by sub-region before discussing possible system wide implications and the open questions that full investigation of these implications pose. The discussion section is now separated into 4 subsections, (1) Roscoe Glacier, (2) Shackleton Ice Shelf, (3) Northcliff-Denman Glacier and (4) Scott-Apfel Glacier. We think there are no seasonal changes in ice flow speed because seasonal forcing observed elsewhere in Antarctica and in Greenland, such as seasonal changes in surface meltwater variability and seasonal sea ice break up reducing back stress, are currently insufficient to have an observable forcing effect on the Shackleton system. We have added an explanation of this to section 4.3 of the discussion below. The discussion section now reads

[revised manuscript text omitted]
 discussion section also contains a lot of results based text, which is presented as almost stand-alone text which isn't used to evidence a discussion.*

We have moved any stand-alone results-based text to the results section and only described results to evidence the discussion. Please see the response to point 9 for more detail.

*11) The conclusions don't summarise all the results. They should contain more information on your important findings about rift propagation and shear margin changes as these could be explored further in future work (in detailed models for example) – which would use all the data you have so carefully collected and presented. This is much more important to report than your concluding remarks about needing more data, in a very generic sense.*

We have added to and restructured the conclusions to emphasise all of our results in the context of existing work and presented directed future research priorities. The conclusions now read

Line 394-434: **5 Conclusions**
Over the 60-year period of observation of the Shackleton system we observe significant rift propagation on the Shackleton Ice Shelf and Scott Glacier, and notable structural changes in the floating shear margins between the ice shelf and the outlet glaciers. Over the period 2017-2022 we observe a significant increase in ice flow speed (~ 50 % close to the ice front) on the floating part of Scott Glacier. Over the same time period we do not observe any seasonal variation or significant change in ice flow speed across the rest of the Shackleton system, and there is no observable change in the ice flow speed of the grounded ice or at the grounding line. However, given the likelihood of modified CDW (observed to be 0.8–2 °C warmer) accessing the shelf (Herraiz-Borreguero and Naveira Garabato, 2022), coupled with the observations that the outermost portions of Shackleton Ice Shelf experience the most intense surface melt outside of the Antarctic Peninsula (Trusel et al., 2013), and a lengthening of the melt season (Zheng et al., 2018), the Shackleton system appears vulnerable to changes in both ocean and atmospheric forcing. Indeed, observed grounding line thinning and retreat (Brancato et al., 2020; Flament and Rémy, 2012) and localised ice flow speed change (Miles et al., 2021; Rignot et al., 2019) may indicate that the Shackleton system is already responding to changes in forcing. However, the timescales over which the system responds, and the implications of this response remain challenging to predict.

We still do not have a clear picture of the cause of the 1972-2008 acceleration in ice flow speed of the Denman Glacier or indeed its later stabilisation. A previous calving cycle reconstruction indicated that a calving event at some point in the 2020s is highly likely (Miles er al., 2019) but from our observations we do not find any indication of when this may occur or the mechanism by which it may be initiated. In addition to thinning and induced dynamic changes there are several features of the system that could significantly speed up the response to external forcing. Such features include; (i) thinning suture zones; if the Masson Island suture zone on

Shackleton Ice Shelf is thinning in response to ocean warming, the melange at the base of the suture zone will preferentially melt, promoting rift propagation and ice shelf break up (Kulessa et al., 2014; Larour et al., 2021; McGrath et al., 2014); (ii) shear margin weakening; the shear margins across the whole system are likely to weaken with increasing melt and elevated meltwater availability but in the region of the system from Scott Glacier towards Mill Island, potentially high salt concentrations in the ice could further enhance melt and increase deformation; (iii) brine infiltration; possible brine infiltration of the firn layers on Shackleton Ice Shelf could enhance fracture propagation and promote hydrofracture if there is an increase surface meltwater with the observed lengthening melt season; and (iv) subglacial conditions; heat flow anomalies in the region of the Denman Glacier are poorly resolved but recent multivariate analysis of available datasets indicate elevated geothermal heat flow (> 70-80 mW m-2) to the west of the Denman region (Wilhelm II Coast) and in the Knox Basin interior (Stål et al., 2021). As observed elsewhere in East Antarctica, the deep trough beneath the Denman Glacier may favour vigorous channelisation of the subglacial meltwater system close to the grounding line (Dow et al., 2020) and freshwater outflow into the sub-ice shelf ocean cavity could locally enhance basal melt rates near the grounding line (Jenkins, 2011; Wei et al., 2020).

The potential vulnerability of the Shackleton system to increasing atmospheric and ocean forcing, the magnitude of potential sea level rise of the Denman Glacier alone (~ 1.5 m) (Morlighem et al., 2020) and the potential link to the Aurora Subglacial Basin (through the Knox Basin) make this region of East Antarctica one of significant importance in improving predictions of sea level rise. Critical to assessing the timing and magnitude of sea level rise contributions are improvements in the measurement of ice cavity bathymetry, subglacial conditions, and the evolution of surface melt water systems, as well as targeted collection of field data to allow better incorporation of features such as suture zones, pinning points and brine infiltration into numerical models of ice shelves.

*12) The Data Availability statement doesn't mention the radargrams. If they aren't freely available to download online, can they be requested from the data collectors? Also mention the BedMachine database used in Figure 1.*

The radar-based ice bottom elevation data and radargrams presented here will be published to the Australian Antarctic Data Centre (https://data.aad.gov.au/) upon publication. We have added this statement, as well as BedMachine, to the data availability statement, which now reads

Line 445-452: **Data availability**
All satellite imagery used in this work are freely available as follows; Landsat 8 OLI, 5 TM and 1 (all downloaded from https://earthexplorer.usgs.gov/), MODIS Mosaic of Antarctica 2008-2009 (downloaded from https://nsidc.org/data/nsidc335 0593/versions/2), Sentinel 2 A and B (all downloaded from https://scihub.copernicus.eu/dhus/#/home) Sentinel 1A and B GRD (all downloaded from https://scihub.copernicus.eu/dhus/#/home) and ARGON KH-5 (downloaded from https://earthexplorer.usgs.gov/). BedMachine (v2) data is freely available (https://nsidc.org/data/nsidc-0756/versions/2). The radar-based ice bottom elevation data and radargrams presented here will be published to the Australian Antarctic Data Centre (https://data.aad.gov.au/) upon publication.

*Suggested figure changes are below (note that all figures appear blurry, but I presume this is a result of copying and pasting):*

The quality of the figures was reduced to keep to the file upload size limits, we will make sure all of the figures are of sufficient quality and clarity when uploaded in this revsion.

*Figure 1*
*• Why is the area in panel a so zoomed out? I'd focus in more on the area you examine, maybe just extending as far south as Law Dome so you can really see the detail of the bed topography in your study area.*
We have changed the regional focus of the figure. See below
*• In panel b could you make the glacier names stand out in bold as these are key points*
All of the glacier names are now in bold, see below
*• In the caption mention that numbers 1 and 2 refer to rift systems.*
We have described rifts R1 and R2 in the caption, see below

[Figure]

Figure 1: (a) Area of focus in the regional context of the Aurora and Wilkes subglacial basins, location shown in inset (Background: BedMachine V2 (Morlighem et al., 2020). (b) The Shackleton system overview in February 2021 with the two main rift systems on Shackleton Ice Shelf labelled R1 and R2. All features mapped from Sentinel 2A and 2B imagery acquired 05th – 27th February 2021.

Figure 2
*• Please add glacier names to panel a to make the figure more useful*
We have added the glacier names, see below
*• Add rift numbers to panel a too, to help the text in section 3.2*
We have added the rift number to panel a, see below

[Figure]

Figure 2: (a) Ice front positions of the Queen Mary and Knox coasts since 1962, including the large iceberg hypothesised to have calved from the Denman tongue in the 1940s. Position and blocks mapped from 16th May 1962 – ARGON KH5, 10th - 12th February 1991 – Landsat 5 TM, 1st November – 28th February 2009 – Modis MOA (Scambos et al., 2007) and 5th – 27th February 2021 – Sentinel 2A and 2B. The two main rift systems on the Shackleton Ice Shelf are labelled 1 and 2. (b) Denman Glacier biannual ice front position mapped in February and August from 2015 through 2022 (Background: Sentinel 1a acquired 27th February 2015).

*Figure 3*
• *Change caption to say "Scott Glacier".*
We have changed the caption to Scott Glacier, see below
• *Add glacier name to figure.*
The glacier name has been added to the figure, see below

[Figure]

Figure 3: (a) Scott Glacier ice front position between 2015 and 2019 mapped from Landsat 8 OLI acquired on 16th February 2019, 1st February 2018, 24th February 2016 and 7th February 2015 and from Sentinel 2A acquired on 23rd February 2017 (Background: Landsat 8 OLI – 16th February 2019). (b) Scott Glacier ice front 26th November 2020, the central portion of the front has lost some of the blocks held in place by fast ice and an area immediately to the south of Mill Island (Landsat 8 OLI), (c) Scott Glacier ice front 27th February 2021, the fast ice has broken up and a larger block is separated (Sentinel 2B), (d) Scott Glacier ice front 21st February 2022, (Sentinel 2B).

*Figure 4*
*• Could you label the rifts differently to those already identified as rifts 1 and 2 elsewhere to avoid confusion? Maybe S1, S2 and S3 here, and D1 and D2 previously?*
The rifts have been labelled S1-S3, see below
*• Does the arrow point to North? If so, it should say N.*
The arrow does point north and has been labelled, see below
*• Write out glacier names in full on panel c, so Scott Glacier etc. The same should be done for Figure 7.*

The glacier names have been added in panel c (now a) and to Figure 7, see below
*• In the caption put the acquisition dates in brackets*
Have added the brackets to the acquisition dates, see below

[Figure]

Figure 4: Evolution of the rifts on Scott Glacier from (a) 2015-2022 mapped from
Sentinel 2B (acquired 27th February 2021), Landsat 8 OLI (acquired 16th February
2019 and 25th March 2015) and from Sentinel 2A (acquired 23rd February 2017)
(Background: Sentinel 2B acquired 27th February 2021) and (b) 1962-2009 mapped
from  ARGON KH5 (acquired 16th May 1962), Landsat 1 MMS (acquired 27th
February 1974) and Landsat 5 TM acquired (10th -12th February 1991) and from
MODIS MOA (acquired 1st November – 28th February 2009) – Modis MOA
(Scambos et al., 2007) (Background: MODIS MOA). (c) The floating portion of Scott
Glacier in June 2022, highlighting the iceberg that calved from the western front in
April 2022 and the rifting across the eastern portion of the front towards Chugunov
Island (Background: Sentinel 1A acquired 6th June 2022).

*Figure 5*
*• The caption talks about the "Shackleton Roscoe Glacier". I thought it was just the*
*Roscoe Glacier?*
We were refereeing to the shear margin between the two. We have changed the
caption to say the shear margin between the Shackleton Ice Shelf and Roscoe
Glacier and have marked the shear margin location in figure 5 for clarification. See
below

*• On line 177 you talk about the margin between the Roscoe Glacier and the Shackleton Ice Shelf. Can you mark this margin on the figure?*
This has been marked on figure 5 and added to the caption. See below

[Figure]

Figure 5: Rift opening in the vicinity of the shear margin between the Shackleton Ice Shelf and Roscoe Glacier (location marked in blue) between (a) 2015 (Background: Landsat 8 OLI acquired 14th March 2015 and (b) 2021 (Sentinel 2B acquired 13th February 2021).

*Figure 6*
*• Add year of data collection to each panel*
Have added the year to each panel, see below

[Figure]

Figure 6: (a) The 2015 extent of the Scott-Apfel Glacier shear margin highlighted in blue (Background Landsat 8 OLI acquired 25th March). (b) The 2021 extent of the Scott-Apfel Glacier shear margin highlighted in blue (Background: Sentinel 2B 13th February 2021). (c) Scott-Denman Glacier shear margin in 2015, the dashed blue line highlights the position ad shape of the margin. (Background: Landsat 8 OLI acquired February 2015). (d) The Scott-Denman Glacier shear margin in 2021, the

dashed blue line highlights the widening of the shear margin into the Denman Glacier (Background: Sentinel 2B acquired 27th February 2021).

*Figure 7*
*• Radar name text is on the wrong side. Move all radar names to the left side of the transects in panel a as you are looking at the transects from the sea, not from inland.*
Have moved to radar name to the left side if the transects, see below
*• The radar bed pick key needs a metric (m a.s.l.?)*
Have added m a.s.l. to the key, see below
*• Re-title the radar transect names on the right hand column to match those on the map.*
Have re-titled the transect names to match the map, see below
*• I'd argue that the power colour bar down each radargram is unnecessary.*
Have changed to a single colour bar top of the figure, see below

[Figure]

Figure 7: (a) The location and ice base below sea level of 6 ICECAP radar lines acquired in December 2016 (Background Sentinel 2A acquired 23rd February and 1st March 2017). (b) Annotated ICECAP radar lines (position and east-west line direction shown in a).

*Figure 8*

• *Label glaciers again here (and in Figures 9 and 11). You're used to looking at this area so it's clear to you what's what, but most readers won't be.*
Have labelled all of the glaciers on Figures 8, 9, and 11, see below

[Figure]

Figure 8: (a) Mean speed for 2022 with velocity arrows. (b) Magnitude of the principal strain rate of Shackleton system derived from Sentinel-1 derived mean velocity data over the period of observation. N.B. The feature down flow of pinning point c on the Denman Glacier tongue is an artefact, we see no evidence of rifting in the remote sensing data in this region (e.g., Fig. 2b, 7a).

[Figure]

Figure 9: Percentage difference in mean speed between 2021 and 2018, scaled between +/- 10%, with point locations illustrating the ice speed timeseries in Figure 11.

[Figure]

Figure 10: Percentage difference in mean speed between (a) 2018-17, (b) 2019-18, (c) 2020-19 and (d) 2021-20 scaled between +/- 10%.

[Figure]

Figure 11: Time series of speed at point locations (shown in Figure 10a) across the Denman-Scott-Shackleton system derived from Sentinel 1 between 2017 and 2021 (uncertainties in velocity magnitude are around 0.2 m day-1 following (Benn et al., 2019)) and extended back to 2002 using Measures and ITS_LIVE where available (Rignot et al., 2017).

*Additional, minor comments:*

*Title: Maybe Glaciological history is a better term than setting, which doesn't really mean much.*

We agree that history is more meaningful than setting and have implemented the change, the title now reads

Line 1: Glaciological history and structural evolution of the Shackleton Ice Shelf System, East Antarctica, over the past 60 years

*Line 28: Be more specific. From what speed to what speed?*

The increase is reported at a 15% increase, however we have altered the abstract to better represent our key findings in response to comments by Referee #1 and the sentence has been removed from the abstract.

*Line 30: How do you know acceleration in ice flow and calving are linked?*

We don't know that that the increase in ice flow speed and calving at linked, only that they occur at the same time. We have adjusted the text to reflect this, and the sentence now reads

Line 29-31: Over the period 2017-2022 we observe a significant increase in ice flow speed (up to 50%) on the floating part of Scott Glacier, coincident with small scale calving and rift propagation close to the ice front.

*Line 33: Presumably sediment filled trough rather than just bedrock.*

We agree that it is likely that the trough is sediment filled and have adjusted the text to reflect this, the sentence now reads

Line 33: Given the potential vulnerability of the system to accelerating retreat into the overdeepened, potentially sediment filled bedrock trough, an improved understanding of the glaciological, oceanographic, and geological conditions in the Shackleton system are required to improve the certainty of numerical model predictions and we identify a number of priorities for future research.

*Line 48: This paragraph jumps around. Stick with basin introductions first then move onto the floating components. A more detailed site overview is needed too – to include glacier lengths, thicknesses etc (see points above).*

We agree that the paragraph structure was confusing and lacked basic details. We have added basic basin details (see below) and have fully addressed this comment in response to main comment point 2, please see the above response to point 2 for more detail.

Line 50-69: Ice thicknesses across the grounded portion of the system range from ~ 400 m inland of Shackleton Ice Shelf to > 4000 m at the Denman Glacier (Morlighem et al., 2020). Bed elevations are also wide ranging with the main outlet glaciers grounded well below sea level and experiencing retrograde slopes (Brancato et al., 2020; Morlighem et al., 2020). Close to the grounding line ice thicknesses range from 300 m to 1400 m, thinning toward the ice front where ice thickness is in the range of 180 – 250 m with the exception of the front of the Denman Glacier tongue at

~ 450 m thick (Morlighem et al., 2020). Ice velocity data from the region are sparse before the late 2000s, but recent work identified an increase in ice velocity of the Denman Glacier of ~ 16 % since the 1970s (Rignot et al., 2019), with an increase of 11± 5 % just upstream of the Denman Glacier grounding line between 1972-74 and 1989 and a more recent rate of acceleration of 3 ± 2 % between 1989 and 2007-08 (Miles et al., 2021, their Fig. 3c). Analysis of Envisat data indicated that the Denman Glacier was thinning by 0.4 m year$^{-1}$ upstream of its grounding line between 2002 and 2010 (Flament and Rémy, 2012), and a 5.4 ± 0.3 km grounding line retreat was detected between 1996 and 2017–2018 (Brancato et al., 2020).

*Line 56: What does the lack of an embayment mean in terms of 'major dynamic change'? I fail to see the relevance.*

We have removed this statement from the restructured introduction.

*Line 57: Refer to figure 1*

We have now referenced figure 1 and the sentence reads

Line 69-71:  Beyond the Denman Glacier grounding zone, the has been much less observation but, rest of the Shackleton system has so far shown few signs of major dynamic change, with its flow restrained by islands, ice rises, and ice rumples (Stephenson et al., 1989; Young, 1989) (Fig.1).

*Line 62: You've just said the data is sparse, so how reliable is this information? Can you note any error margins?*

We have added error margins where they are available in the articles we have referenced. The sentence now reads

Line 54-58: Ice velocity data from the region are sparse before the late 2000s, but recent work identified an increase in ice velocity of the Denman Glacier of ~ 16 % since the 1970s (Rignot et al., 2019), with an increase of 11± 5 %  just upstream of the Denman Glacier grounding line between 1972-74 and 1989 and a more recent rate of acceleration of 3 ± 2 % between 1989 and 2007-08 (Miles et al., 2021, their Fig. 3c).

*Line 71: I presume you mean average basal melt rates rather than total melt rates.*

Yes, we do mean average basal melt rates, we have changed the text accordingly and the sentence now reads

Line 67-69: Satellite derived basal melt rates between 2010 and 2018 revealed high but localised average basal melt rates of > 50 m year$^{-1}$ close to the Denman grounding line (Liang et al., 2021), on par with basal melt rates in the Bellingshausen and Amundsen Sea (Adusumilli et al., 2020).

*Line 73. Introduce all acronyms.* The information about the Sabrina Coast doesn't seem to have much relevance to your work but I could be reading this incorrectly.

We have changed this paragraph in response to comment from Referee #1. We have ensured all acronyms have been introduced and removed references to the Sabrina Coast. The paragraph now reads

Line 76-83: The high basal melt rates close to the grounding line in this region of Antarctica have been linked to a warming of up to 0.5 °C over the last 40 years that occurred in the open ocean off East Antarctica, concurrent with an even more pronounced warming of 0.8–2 °C observed over the continental slope (Herraiz-Borreguero and Naveira Garabato, 2022). This Circumpolar Deep Water (CDW) warming is linked to a poleward shift of the Antarctic Circumpolar Current's southern extent onto the Indian Ocean sector of the East Antarctic continental slope, in which the Shackleton Ice Shelf is located. The continental slope warming appeared strongest near ice shelves that are thinning or have retreating grounding lines such as the Denman Glacier (Herraiz-Borreguero and Naveira Garabato, 2022).

*Line 74: But how likely is this given what you say in the next sentence? Probably best not to jump to possibilities here.*

The sentence has been removed.

*Line 79: I think you mean grounded.*

Yes, this was a typo, we do mean grounded and have changed the word in the text.

Line 87: You don't mention the multi-decadal timeframe you talk about in the abstract.

This was an oversight; we have amended to text to reflect this, and the sentence now reads

Line 94-95: Here we add to the previously reported dynamic changes in the Denman Glacier over the 60-year period of observation and place them into the wider regional context of the Shackleton system.

*Line 92: Just talk about the Shackleton system rather than the whole coastal area.*

We have made the change and the sentence now reads

Line 102-103: Surface structures and features of the Shackleton system were mapped from satellite imagery using standard GIS techniques (following Glasser et al. (2009)).

*Line 83: Can you briefly state why you chose every 6 months, presumably you have data from February and July/August then? Is that just because of daylight? March to September might allow you to capture more of the seasonal changes? Note that I'm not asking you to change your data analysis period! Just note exactly which months you studied and why you made that call.*

It was a compromise between capturing as much of the potential seasonal signal as possible while still having enough cloud free optical images to cover the system once

a year. We have added a sentence to the methods to explain this. The sentence reads

Line 105-108: We used available optical and SAR imagery from February and SAR imagery from August of each year to allow us to capture any seasonal variation while maximising the number of cloud-free optical images available

*Line 94: The datasets need referenced, or you at least need to provide reference to a data availability section of the paper.*

We have provided links to all of the datasets used in the data availability section of the paper. The section reads

Line 447-456:  Data availability
All satellite imagery used in this work are freely available as follows; Landsat 8 OLI, 5 TM and 1 (all downloaded from https://earthexplorer.usgs.gov/), MODIS Mosaic of Antarctica 2008-2009 (downloaded from https://nsidc.org/data/nsidc335 0593/versions/2), Sentinel 2 A and B (all downloaded from https://scihub.copernicus.eu/dhus/#/home) Sentinel 1A and B GRD (all downloaded from https://scihub.copernicus.eu/dhus/#/home) and ARGON KH-5 (downloaded from https://earthexplorer.usgs.gov/). BedMachine (v2) data is freely available (https://nsidc.org/data/nsidc-0756/versions/2). The radar-based ice bottom elevation data and radargrams presented here will be published to the Australian Antarctic Data Centre (https://data.aad.gov.au/) upon publication.

*Line 110: Reference required for Gamma software*

There isn't a reference as such, but we have added to website, the sentence now reads

Line 122- 124: Using the standard Gamma software (https://www.gamma-rs.ch/software) and following commonly adopted methods, feature tracking uses cross-correlation to find the displacement of surface features between pairs of images, which are then converted to velocities using the time delay between those images (Luckman et al., 2007).

*Line 113: How short? Can you give a range?*

The original statement was intended to explain that shorter delays lead to better coherence and where delays are sufficiently short (and no melt etc.) the surface features tracking uses include speckle. While there is not a simple answer to this, we have changed the following sentence to include the possible repeat times for clarification, the sentence now reads

Line 127-128: We applied feature tracking to all image pairs of 6 and 12-day repeat-pass period (about 60 per year) and selected the best single velocity map

*Line 114: The cold temperatures is an assumption. Delete this.*

Happy to delete, the sentence now reads

Line 125-126: Where the time-delay between images is sufficiently short, and surface change is minimized, trackable surface features include fine-scale coherent phase patterns (speckle) and the quality of the derived velocity map is maximized.

*Line 115: Can you note how many image pairs you used to show off the rigour of your work?*

We used roughly 60 images per year and have added this to the text, the sentence now reads

Line 127-128: We applied feature tracking to all image pairs of 6 and 12-day repeat-pass period (about 60 per year) and selected the best single velocity map.

Line 125 (and the rest of the paragraph): More detail required. What frequency did the radar operate at? How was the data topographically corrected? What velocity did you use? Did you apply a firn correction?

We have added the required detail to the section, please see the response to point 3 above.

No firn correction was needed for the qualitative analysis in the present study. The data providers prefer releasing data without a firn correction so that data compilers (e.g. Bedmap, Bedmachine) may apply a consistent approach.

*Line 135: Do you mean no obvious change in the annual rate of advance?*

Yes, we do mean the annual rate of advance, we have clarified this in the text and the sentence now reads

Line 154-155: Over the same time period, the Shackleton Ice Shelf's central front advanced a total of 18 km with no obvious change in the annual rate of advance (Fig. 2a).

*Line 137: Which front are you talking about?*

We are referring to the front of the Shackleton Ice Shelf itself, have clarified this in the text and the sentence now reads

Line157-158: Between 2015 and 2022, the front of the Shackleton Ice Shelf then advanced steadily by ~ 0.3 km year$^{-1}$ while the Denman Glacier's ice front advanced at a rate of 1.8 km year-1 over the same time frame, with a uniform pattern of advance and no seasonal variability in advance rate observed (Fig. 2b).

*Line 139: The lack of seasonal change is interesting is it not? So something you should talk about in your discussion?*

We agree that this is interesting, we think there are no seasonal changes in ice flow speed because seasonal forcing observed elsewhere in Antarctica and in Greenland, such as seasonal changes in surface meltwater variability and seasonal sea ice

break up reducing back stress, are currently insufficient to have an observable forcing effect on the Shackleton system. We have added an explanation of this to section 4.3 of the discussion (detailed in response to point 9 above). The section now reads

Line 351-360: The ice flow speed of the Denman-Northcliff Glacier has been constantly ~ 5 m day-1 close to the ice front over the period 2015-2022, with a uniform flow direction and no observable seasonal variability in flow speed (Fig. 2b, 8a, 11b). While efficient meltwater surface-to-bed connections via moulins and fractures are common in the Greenland Ice Sheet and seasonal changes in surface meltwater variability have been observed to cause seasonal changes in ice flow speeds (Hoffman et al., 2018; Sundal et al., 2011), there is currently very little evidence for coupling between the surface and basal hydrological systems in Antarctica (Bell et al., 2018). However, season variations in ice flow speed have been observed on an outlet glacier in East Antarctica, coincident with seasonal sea ice break up reducing back stress on the system and the onset of seasonal melt thought to weaken the shear margins (Liang et al., 2019). Our results suggest that seasonal variation in sea ice extent or surface melting are currently insufficient to have an observable forcing effect on the Shackleton system.

*Line 145: How do you know it's the calving that's causing the retreat and not faster ice flow or melting?*

We don't know that melting is not contributing but we have no observations of melting here. We do see some indication of small-scale calving events in the form of icebergs adjacent to the ice front and have changed the sentence to report this rather than suggest causation. The sentence now reads

Line 165-166: Since 2020, small scale calving has been occurring on the side of the Scott Glacier adjacent to Mill Island, where the ice front position now lies ~ 5 km inland of the 2015 front (Fig. 3b-d).

*Line 171: How do you define the grounding line?*

The grounding line we use is the same as in the figures, defined by MEaSURES and referenced to Rignot et al., 2017. We have included this in the text and the sentence now reads

Line 198-199: The rifts initiates approximately 20 km down glacier of the grounding line (as defined by MEaSURES (Rignot et al., 2017)) and widen to ~ 2.5 km as they flow around the Taylor Islands.

*Line 179: What happened prior to this? Is there a reason why you don't explore data prior to 2015 here?*

We do describe earlier rift evolution in the following sentence, but we only have sufficient temporal resolution to measure the annual rates of change presented here from 2015 onwards.

Line 183: Why don't you talk about any features on the Apfel Glacier?

We did not discuss Apfel Glacier in this section because we do not observe any significant features or change in this region. We have added a sentence to clarify this which reads

Line 203-205: There is no clear margin between Scott Glacier and Apfel Glacier offshore of the Tylor Islands and no observable rifts on Apfel Glacier inland of the Tylor Islands (Fig. 4a).

*Line 186: Reference not needed as it's your observations that show this.*

We have removed the reference and the sentence now reads

Line 232-233: The Scott Glacier floating shear margin that abuts the Denman Glacier is more clearly defined and has been widening into Denman Glacier in the vicinity of Chugunov Island (Fig. 6c-d).

*Line 188: Why is this remarkable? A statement like that needs to be backed up by other references that say this situation is really unusual. I'd use the word notably, or something similar instead.*

We agree and are happy to make the change. The sentence now reads

Line 234-235: Notably, the shear margin appears to bulge progressively into Denman Glacier and is double the width by 2022 as compared with 2015 (Fig 6c-d).

*Line 191: You should say the radar provides a snapshot of information.*

We agree and are happy to make the change, the sentence now reads

Line 208-209: The ICECAP airborne radar lines flown in December 2016 provide a snapshot of information about the thickness and ice structure through the floating ice.

*Line 195: Split is the wrong term. The ice doesn't fracture apart.*

We agree and have changed the text accordingly, the sentence now reads

Line 213-214: The Denman tongue appears to consist of two distinct longitudinal sections with very different ice thicknesses,

*Line 199: What's consistently noisy? Radar returns? The bed?*

We have removed this sentence.

*Line 232: Use precise dates here rather than 'beginning'.*

Happy to make the change, the text now reads

Line 276-277: However, the downstream 30 km of the floating ice tongue show significant acceleration from the January 2020 through to May 2021.

*Line 239: What time span do you analyse here?* – Emailed Adrian

The magnitude of principle strain rate is derived from the mean velocity maps from 2017-2020, we have added this to the text and the sentence now reads

Line 244-245: The magnitude of the principal strain rate, derived from the mean velocity maps (2017-2020), highlights the shear margins of the Denman Glacier and Roscoe Glacier with the Shackleton Ice Shelf and the shear margins of the Scott Glacier (Fig. 8b), as well as four pinning points across the system (Fig. 8b).

*Line 241: I think you mean to refer to figure 11.*

Yes, we do mean Figure 11 and have corrected the reference.

*Line 247: What do you mean by that? Longer-term change in what? Glacial output?*

This sentence has been removed in the restructuring of the discussion in response to the main points above.

*Line 274: This reads like a literature review. How does it relate to your findings? You never talk about melange.*

The detail was included in response to a previous review request. We have shortened the section and discussed the relevance to our findings, the paragraph now reads

Line 311-326: The lack of significant rift propagation in rift system R1 on the main body of Shackleton Ice Shelf appears directly related to the Masson Island suture zone (Fig 2a). Indeed, there is growing recognition that the softer marine ice present in suture zones inhibits the growth of large-scale fractures, acting to stabilise ice-shelves by reducing local stress intensities (Kulessa et al., 2014; Larour et al., 2021; McGrath et al., 2014). While current observations suggest suture zones promote stability by halting rift propagation, a strong relationship between the thickness of ice mélange found in suture zones, and the opening rate of the rifts has been observed, indicating that ice mélange thinning rather than ice shelf thinning can promote rift propagation (Larour et al., 2021). The increased sea water content and warmer temperature of the ice mélange suggests it may be more vulnerable to thinning due to future surface and basal melting than the surrounding meteoric ice, potentially affecting rates of rift opening and propagation (Kulessa et al., 2014, 2019; McGrath et al., 2014). Rift system R2 has undergone more significant change over the period of observation, although the rift tip does not appear to have reached the Masson Island suture zone (Fig. 2a) as yet and may experience a similar behaviour to that of R1 when it does. With prominent suture zones and pinning points as likely agents of stability (Kulessa et al., 2014, 2019), increased ice shelf thinning could lead to a significant calving event due to coincident suture zone thinning and further rift propagation, with unpinning of the ice shelf on the side of the Shackleton Ice Shelf adjacent to Roscoe Glacier ('d' in Fig. 8b).  The relatively slow grounded ice flow

speeds onshore of the Shackleton Ice shelf suggests such a calving event is unlikely to initiate an increase in ice flow into the ocean, indeed it is considered passive in terms of ice buttressing (Fürst et al., 2016).

*Line 275: Of what ice melange? At your study site?*

We are referring to the ice melange found in suture zones and have clarified this in the text, the sentence now reads

Line 314-316: While current observations suggest suture zones promote stability by halting rift propagation, a strong relationship between the thickness of ice mélange found in suture zones, and the opening rate of the rifts has been observed, indicating that ice mélange thinning rather than ice shelf thinning can promote rift propagation (Larour et al., 2021).

*Line 278 (the paragraph): Contains lots of results again, but you need to discuss what the results mean for your study site and out knowledge of antarctica and processes as a whole.*

This section has been re-written in response to the main points. Please see the detailed response to main point 9 for details.

*Line 294: What do you mean by 'connection'?*

This section has been re-written in response to the main points. The sentence now reads

Line 372-374: The observed changes in ice front position of Scott Glacier are coincident with a doubling of ice flow speed at the ice front, from 2 m day$^{-1}$ in January 2020 to 4 m day$^{-1}$ in January 2022 (Fig. 11c).  The increase in ice flow speed extends from the ice front to rift S1 but is not observable inland of this feature.

*Line 305: How do your results underpin this hypothesis?*

This section has been rewritten in response to the main points above. Please see the response to main point 10 for more details.

*Line 320: It's unclear exactly what data you need more of. Do you need better bed topography data in the region? Or do you want better ocean temperature data? How will a knowledge of local geology help? This statement is rather generic and therefore unhelpful.*

We have rewritten the section to provide specific suggestions for future priorities, the section now reads

Line 431-437: The potential vulnerability of the Shackleton system to increasing atmospheric and ocean forcing, the magnitude of potential sea level rise of the Denman Glacier alone (~ 1.5 m) (Morlighem et al., 2020) and the potential link to the Aurora Subglacial Basin (through the Knox Basin) make this region of East

Antarctica one of significant importance in improving predictions of sea level rise. Critical to assessing the timing and magnitude of sea level rise contributions are improvements in the measurement of ice cavity bathymetry, subglacial conditions, and the evolution of surface melt water systems, as well as targeted collection of field data to allow better incorporation of features such as suture zones, pinning points and brine infiltration into numerical models of ice shelves.

---

## Author Response (AR3)

*L222 and where you include (Herraiz-Borreguero and Naveira Garaboto, 2022): might it be appropriate to also include (van Wijk et al., 2022) here? I appreciate this source was not available at the time of submission, but can perhaps be included in the final revised version?*

Thank you for drawing our attention to work, it is very relevant. We've added it into the manuscript introduction and included the reference in our conclusion.

L76-85 - The high basal melt rates close to the grounding line in this region of Antarctica have been linked to a warming of up to 0.5 °C over the last 40 years that occurred in the open ocean off East Antarctica, concurrent with an even more pronounced warming of 0.8–2 °C observed over the continental slope (Herraiz-Borreguero and Naveira Garabato, 2022). This Circumpolar Deep Water (CDW) warming is linked to a poleward shift of the Antarctic Circumpolar Current's southern extent onto the Indian Ocean sector of the East Antarctic continental slope, in which the Shackleton system is located. The continental slope warming appeared strongest near ice shelves that are thinning or have retreating grounding lines such as the Denman Glacier (Herraiz-Borreguero and Naveira Garabato, 2022). In addition, recent profiling float data have revealed that the thickest and warmest modified CDW layers in the region have been observed in a deep trough adjacent to the Denman Glacier tongue (van Wijk et al., 2022).

*L226 remove 'in' before 'have been observed'*

Removed

L82-4 - Although basal melt is considered the dominant form of melt related mass loss in East Antarctica, the outermost portions of Shackleton Ice Shelf have been observed to experience the most intense surface melt outside of the Antarctic Peninsula (>>200mm w.e. year-1) (Trusel et al., 2013).

*L238 remove 'Ice Shelf' here?*

Removed

L95-6 – Here we add to the previously reported dynamic changes in the Denman Glacier over the 60-year period of observation and place them into the wider regional context of the Shackleton system.

*L241-242 maybe you can explicitly refer to the section numbers where you report on each of these elements of your analysis?*

We have added the relevant sections to each of the elements and the sentence now reads

L98-100 - We firstly report on the main structural features (Sections 3.1-3.3) and dynamic changes across the whole system (Sections 3.4-3.6) and then discuss the changes in the system and their possible impact by sub-system region (Sections 4.1-4.4).

*L383 'initiate' instead of 'initiates'?*

Changed

L199-200 – The rifts initiate approximately 20 km down glacier of the grounding line (as defined by MEaSURES (Rignot et al., 2017)) and widen to ~ 2.5 km as they flow around the Taylor Islands.

*L416 should Section 3.3 be titled 'ice thickness' instead of 'ice extent'?*

Changed

L208 – 3.3 Ice thickness

*Section 2.2 Please add some further details on how you calculated the strain rates. Did you apply filters to suppress the noise, and do you trust results for the extensive area of high strain rates upstream of the Northcliff Glacier grounding line?*

We did not apply filtering to supress noise and it is likely that the high strain feature upstream of Northcliff Glacier is noise. Strain was calculated across 2 pixels (of 100m) N-S and E-W of the central pixel. We have included these details and added the following to Section 2.2 and the caption of Figure 8.

L135-7 - The magnitude of the principal strain rate was derived from the mean velocity maps (2017-2020), strain was calculated across 2 pixels (of 100m) North-South and East-West of the central pixel, no filtering was applied to suppress noise.

Figure 8: (a) Mean speed for 2022 with velocity arrows. (b) Magnitude of the principal strain rate of Shackleton system derived from Sentinel-1 derived mean velocity data over the period of observation. N.B. The area of high strain upstream of Northcliff Glacier is likely noise as not filtering was applied to supress noise. The feature down flow of pinning point c on the Denman Tongue is an artefact, we see no evidence of rifting in the remote sensing data in this region (e.g., Fig. 2b, 7a).

*Section 2.3 Refer to Figure 7 for the geographical context and spatial coverage of the flightlines.*

We have referred to figure 7 in section 2.3, the opening sentence of which now reads

L135-136 - The ice-penetrating radar data presented here were acquired on two survey flights using the Snow Eagle 601 BT-67 aircraft (Cui et al., 2018) flown on 19 and 20 December 2018 (Line locations shown in Fig. 7).

*Fig 7 needs labels a), b),…*

Added

[Figure]

Figure 7: (a) The location and ice base below sea level of 6 ICECAP radar lines acquired in December 2016 (Background Sentinel 2A acquired 23rd February and 1st March 2017). (b) Annotated ICECAP radar lines (position and east-west line direction shown in a).

*L434 refer to section 3.2 instead of 3.3*

Changed

L226-7 – As described in Section 3.2, a rift along the shear margin was observed to be three times the length and 10 times the width to that observed in 2015 (Fig. 5).

*Figs 8, 9 and 10 Can you add the grounding line and ice front for context?*

We have added the grounding line to figures 8, 9 and 10 (see below) but have not added the ice front position to the plots as none of them represent a single point in time. They are either based on an annual mean or represent change over several years. We did try picking a single ice front position, but this made the figures confusing as the Denman Glacier particularly has a high annual flow speed and therefore significant rate of annual ice front advance.

[Figure]

Figure 8: (a) Mean speed for 2022 with velocity arrows. (b) Magnitude of the principal strain rate of Shackleton system derived from Sentinel-1 derived mean velocity data over the period of observation. N.B. The area of high strain upstream of Northcliff Glacier is likely noise as not filtering was applied to supress noise. The feature down flow of pinning point c on the Denman Tongue is an artefact, we see no evidence of rifting in the remote sensing data in this region (e.g., Fig. 2b, 7a).

[Figure]

Figure 9: Percentage difference in mean speed between 2021 and 2018, scaled between +/- 10%, with point locations illustrating the ice speed timeseries in Figure 11.

[Figure]

Figure 10: Percentage difference in mean speed between (a) 2018-17, (b) 2019-18, (c) 2020-19 and (d) 2021-20 scaled between +/- 10%.

*L573 capitals for Ice Shelf*

Changed

L256 - 60 km of the floating tongue of Scott Glacier and the Shackleton Ice Shelf (Fig. 9).

*L734 check units: rate of deceleration should be m/yr2/yr*

Changed

L308-9 - In 2021 flow speeds in this area of the ice shelf were in the region of 1 m day$^{-1}$ so that the annual rate of deceleration over the 3-year period would only equate to ~ 10 m year$^2$.

*L775 remove 'most'?*

Removed

L341-2 - Although both suggestions are plausible at this location, current observations are insufficient to conclusively identify the cause of the reflector and therefore the impact of the evolution of the system.

*L800 'experienced' instead of 'experience'*

Changed

L367-8 - We observe the ice front of Scott Glacier to be in a similar position in 2009, 2002, 1991 and 1962 (Fig. 2a, 3), but the ice front may have experienced retreat inland of this position in the intermediate time periods.

*L828 'an' instead of 'in'*

Changed

L391-2 - If the muted reflectors identified (Fig. 7b) are in indication of high salt concentration, any changes in atmospheric or ocean forcing could have an enhanced impact on this region of the Shackleton system.

References

van Wijk, E. M., Rintoul, S. R., Wallace, L. O., Ribeiro, N., & Herraiz-Borreguero, L. (2022). Vulnerability of Denman Glacier to ocean heat flux revealed by profiling float observations. Geophysical Research Letters, 49, e2022GL100460. https://doi.org/10.1029/2022GL100460